# ADVERSARIAL VISUAL CONTRASTIVE DECODING FOR MITIGATING HALLUCINATIONS IN LARGE VISION-LANGUAGE MODELS

## ABSTRACT

Large Vision-Language Models (LVLMs) have achieved remarkable progress in various multimodal AI tasks but are prone to generating hallucinations—outputs that are plausible-sounding yet factually incorrect or ungrounded in the visual input. This phenomenon, particularly the generation of non-existent objects or misdescribed object attributes, severely undermines model reliability. This paper introduces Adversarial Visual Contrastive Decoding (AVCD), a novel inference methodology designed to mitigate hallucinations in LVLMs. AVCD refines the existing Visual Contrastive Decoding (VCD) framework by replacing its use of random noise with adversarial perturbations. These adversarial images are specifically engineered to perturb the vision encoder's features to decrease cosine similarity with the original image. Our analysis reveals that unlike random noise, these adversarial perturbations are directional; they actively steer the model toward hallucinatory states rather than simply degrading visual features. This creates a more potent and informative contrastive signal, enabling a more effective suppression of hallucinatory content. Experiments on standard benchmarks indicate that the proposed AVCD method achieves notable performance improvements over VCD and other baseline techniques. This work underscores the potential of leveraging adversarial principles not merely for identifying model vulnerabilities but as a constructive tool in enhancing the faithfulness and reliability of LVLM outputs.

## 1 INTRODUCTION

Large Vision-Language Models (LVLMs) have become increasingly prominent, achieving remarkable progress in diverse multimodal AI domains such as image captioning and visual question answering (Liu et al., 2023; Bai et al., 2023). However, similar to text-only Large Language Models (LLMs), LVLMs are prone to the phenomenon of *hallucination*—generating information that does not actually exist as if it were factual (Rohrbach et al., 2018). In LVLMs, hallucinations frequently manifest as descriptions of nonexistent objects, misrepresented object attributes (e.g., color, quantity, location), or plausible yet image-irrelevant text (Huang et al., 2024). These hallucinations significantly degrade the reliability and accuracy of the model's outputs. This is a critical concern, particularly in applications where precision and safety are paramount, because hallucinations can pose serious risks (Chen et al., 2024a). Consequently, there is a growing need for research aimed at mitigating LVLM hallucinations and building more trustworthy AI systems.

One recently proposed approach to reduce LVLM hallucinations is Visual Contrastive Decoding (VCD) (Leng et al., 2024). VCD is a straightforward, training-free technique that suppresses hallucinations by contrasting the output distributions from an original visual input and a distorted version of that input, leveraging the premise that hallucinations become more evident under distorted conditions. The distorted input is typically generated by adding Gaussian noise to obscure visual features, prompting the model to rely excessively on statistical biases and language priors, thus amplifying hallucinations. However, this approach is founded on a suboptimal premise. Its reliance on undirected Gaussian noise merely obscures visual features at random, failing to expose the specific semantic pathways that lead to hallucinations. This yields a weak and diffuse contrastive signal, limiting the effectiveness of VCD. Several subsequent attempts to mitigate these issues (Wang et al., 2024; Favero et al., 2024) still depend on randomness, failing to fully address VCD's limitations.

In this work, we propose a significant improvement upon VCD—*Adversarial Visual Contrastive Decoding (AVCD)*—which modifies how hallucination-inducing distorted images are generated. AVCD replaces random Gaussian noise with adversarial perturbations to create clearer and more informative negative examples. These adversarial images are created by perturbing the original input, causing its visual features (e.g., obtained via a vision encoder like CLIP (Radford et al., 2021)) to deviate substantially from their original semantic content. AVCD thus leverages adversarial examples—inputs designed to mislead models (Szegedy et al., 2013)—as precise negative signals. Traditionally considered a model vulnerability, adversarial examples are repurposed by AVCD as powerful tools to pinpoint and suppress hallucination pathways through contrastive decoding.

Unlike random noise, which indiscriminately degrades visual features toward an informationless state, adversarial perturbations explicitly target specific semantic directions. Consequently, AVCD yields a significantly stronger and more discriminative contrastive signal, effectively identifying and suppressing hallucinations. We also provide a detailed discussion of related works in Appendix C. In summary, our main contributions are as follows:

- We propose Adversarial Visual Contrastive Decoding (AVCD), a novel inference methodology that replaces the random noise in the contrastive decoding stage of VCD with adversarial perturbations to suppress hallucinations.
- We demonstrate through detailed theoretical and empirical analysis that adversarial perturbations actively steer the model toward hallucinatory states to create a more potent and informative contrastive signal than random noise.
- We demonstrate the effectiveness of AVCD across diverse benchmarks, achieving significant reductions in hallucination rates compared to baseline methods.

## 2 METHOD

This section details the proposed AVCD approach for mitigating hallucinations. We first introduce necessary preliminaries and then describe how AVCD generates adversarially perturbed images.

### 2.1 PRELIMINARIES

**Large Vision-Language Models** An LVLM generates output text tokens $\mathbf{y} = \{y_1, \ldots, y_K\}$ based on an input image and an input text prompt. The input image is encoded into tokens $\mathbf{v} = \{v_1, \ldots, v_N\}$, and the text prompt into tokens $\mathbf{x} = \{x_1, \ldots, x_M\}$. The probability of each output token is autoregressively modeled using a softmax function $S(\cdot)$ over the logits $\ell(\cdot)$ as:

$$p(y_k|\mathbf{v}, \mathbf{x}, \mathbf{y}_{<k}) = S(\ell(y_k|\mathbf{v}, \mathbf{x}, \mathbf{y}_{<k})),$$

LVLMs typically comprise three components:

- **Vision Encoder** $(E_V)$**:** Processes the input image data $(I)$ to extract an image feature representation $\mathbf{f}_I = \{\mathrm{f}_I^1, \ldots, \mathrm{f}_I^N\} = E_V(I)$.
- **Projector Module** $(P)$**:** Maps the visual features $\mathbf{f}_I$ into image tokens $\mathbf{v} = P(\mathbf{f}_I)$ that are compatible with the language model's embedding space.
- **LLM:** Generates the output text $\mathbf{y}$ from the sequence of image and text tokens, $\mathbf{v}$ and $\mathbf{x}$.

For example, the LLaVA model (Liu et al., 2023) combines a pre-trained Vicuna LLM (Chiang et al., 2023) with a CLIP vision encoder (Radford et al., 2021), and uses a linear projector to map visual features into the LLM's embedding space. AVCD can be applied on top of any such LVLM architecture at inference time, without modifying these components or requiring additional training.

**Visual Contrastive Decoding** VCD (Leng et al., 2024) is a training-free strategy to mitigate hallucinations in LVLMs. It contrasts the model's outputs from an original image against those from a distorted version of it. The decoding distribution is adjusted as follows:

$$p(y \mid \mathbf{v}, \mathbf{x}) = S\big((1 + \alpha)\,\ell(y \mid \mathbf{v}, \mathbf{x}) - \alpha\,\ell(y \mid \mathbf{v}', \mathbf{x})\big).$$

Here, $\mathbf{v}$ and $\mathbf{v}'$ are the image tokens from the original and the distorted image, respectively. The hyperparameter $\alpha$ controls the strength of the contrast. By subtracting the logits from the distorted

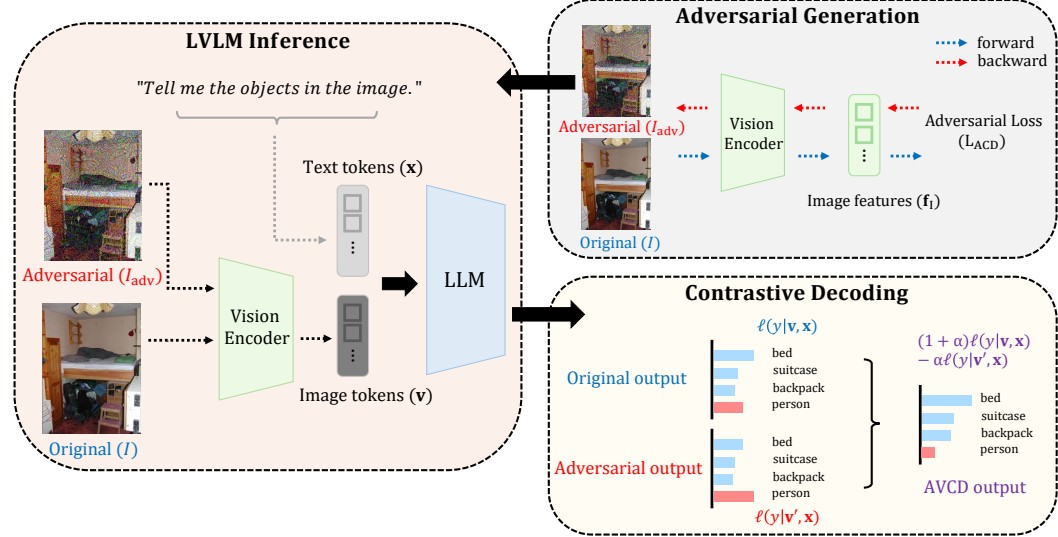

Figure 1: Overview of Adversarial Visual Contrastive Decoding (AVCD). An adversarial image is generated from the original to maximize feature dissimilarity. The LVLM processes both the original and adversarial images. AVCD then contrasts the two resulting outputs, which suppresses hallucinations (e.g., the "person") that are more likely in the adversarial path.

path, VCD suppresses hallucinated content, which tends to be amplified under noisy conditions. However, the effectiveness of VCD is limited by its use of undirected noise (e.g., Gaussian noise). Such distortions do not strategically target model weaknesses and are thus suboptimal for exposing hallucination tendencies. This limitation motivates the search for more effective distortions.

**Adversarial Examples**  Adversarial examples are inputs created by adding a perturbation, $\delta$, to an original input $I$ ($I_{\text{adv}} = I + \delta$). This perturbation is intentionally designed to cause a model to produce erroneous outputs (Szegedy et al., 2013). A classic method for generating such perturbations is the Fast Gradient Sign Method (FGSM) (Goodfellow et al., 2014), which aligns the perturbation with the sign of the model's loss gradient—effectively pushing the input in a direction that maximizes the model's loss:

$$\delta = \epsilon \cdot \text{sign}(\nabla_I L(I)),$$

where $\epsilon$ is a hyperparameter that controls the magnitude of the perturbation. While adversarial examples are typically treated as a model vulnerability, our work reframes them as a constructive tool. We generate adversarial examples not to cause failure, but to serve as potent negative guides in a contrastive decoding framework. This approach leverages attack techniques for a constructive purpose: to actively suppress hallucinations, rather than simply as something to defend against.

## 2.2 Adversarial Visual Contrastive Decoding

Adversarial Visual Contrastive Decoding (AVCD) modifies the contrastive decoding process by replacing the randomly noised image with an adversarial image, $I_{\text{adv}}$. This adversarial image is specifically designed to generate features that are highly dissimilar to the original image's features, thereby acting as a more effective "hard negative" to guide the decoding process and mitigate hallucinations.

**Adversarial Objective**  To achieve this, we define an adversarial loss, $L_{\text{adv}}$, as the cosine similarity between the vision encoder features of the original image $I$ and the adversarial image $I_{\text{adv}}$:

$$L_{\text{adv}}(I, I_{\text{adv}}) = \sum_{j=1}^{N} \frac{\mathbf{f}_I^j \cdot \mathbf{f}_{I_{\text{adv}}}^j}{||\mathbf{f}_I^j|| \cdot ||\mathbf{f}_{I_{\text{adv}}}^j||}.$$

We select cosine similarity over other metrics because cosine similarity isolates the angular relationship between feature vectors—a proxy for semantic meaning. By minimizing this loss, we force the feature representation of $I_{\text{adv}}$ to diverge as much as possible from that of $I$ in the feature space. This ensures that $I_{\text{adv}}$ is not just a noisy variant but a semantically misleading negative example.

**Adversarial Image Generation**    To generate this adversarial image, we employ the Projected Gradient Descent (PGD) attack (Madry et al., 2017), which can be seen as an iterative variant of FGSM for finding more effective perturbations. We employ PGD as a generic optimization framework to maximize our custom feature-level adversarial loss, distinguishing our approach from attacks tied to standard classification objectives. This versatility is crucial for our method, as it allows us to directly optimize our custom feature cosine similarity loss. Furthermore, its iterative nature ensures the perturbation is strong enough to effectively reveal vulnerabilities.

We initialize the process with the original image and iteratively update it to minimize $L_{\mathrm{adv}}$. At each step $t$, we adjust the image in the direction opposite to the gradient of the loss (decrease in similarity). The update rule is as follows:

$$I_{\mathrm{adv}}^{t+1} = \Pi_{\mathcal{B}_\epsilon(I)}(I_{\mathrm{adv}}^t - \eta \cdot \mathrm{sign}(\nabla_{I_{\mathrm{adv}}^t} L_{\mathrm{adv}}(I, I_{\mathrm{adv}}^t))),$$

where $\eta$ is the step size. After each update, the perturbed image is projected back (denoted by $\Pi$) onto $L_\infty$-norm ball $\mathcal{B}_\epsilon(I)$ around the original image $I$. This projection ensures the perturbation $\delta = I_{\mathrm{adv}} - I$ is not changed beyond a certain magnitude ($||\delta||_\infty \leq \epsilon$). We repeat this process for a number of steps $T_{\mathrm{adv}}$ to obtain the final adversarial image $I_{\mathrm{adv}}$.

A key distinction of our approach lies in how we utilize the perturbation budget $\epsilon$. In traditional adversarial attacks, a core constraint is imperceptibility, which necessitates a very small $\epsilon$ to ensure the attack is not detectable by a human observer. However, in AVCD, the adversarial image $I_{\mathrm{adv}}$ is a purely internal and intermediate product used only to generate a contrastive signal; it is never exposed to the end-user. This liberates our method from the imperceptibility constraint, allowing us to use a significantly larger budget (e.g., $\epsilon = 64$ or $256$) than traditional attacks. This larger budget, controlled via the projection step, is crucial for creating a more potent and semantically divergent "hard negative" essential for effective contrastive decoding.

**Decoding with Adversarial Contrast**    During inference, we perform contrastive decoding using the original image $I$ and the adversarial image $I_{\mathrm{adv}}$. The decoding distribution is computed by contrasting the logits from the two inputs, as in standard VCD:

$$p(y \mid \mathbf{v}, \mathbf{x}) = S\big((1+\alpha)\,\ell(y \mid \mathbf{v}, \mathbf{x}) - \alpha\,\ell(y \mid \mathbf{v}'_{I_{\mathrm{adv}}}, \mathbf{x})\big).$$

Intuitively, since $I_{\mathrm{adv}}$ is optimized to activate features that differ from the ground truth, it is more likely to produce hallucination-related tokens. By subtracting the logits from this adversarial path, AVCD can more effectively identify and suppress hallucinated content compared to VCD, which uses random noise. Figure 1 provides a visual summary of the entire AVCD procedure detailed above, from adversarial image generation to the final contrastive decoding. Pseudocode for AVCD is provided in Appendix A.

## 3    ANALYSIS

This section analyzes why AVCD is effective by examining how adversarial perturbations influence the model's behavior. We quantify the extent to which adversarial perturbation amplifies hallucinations compared to Gaussian noise. We then conduct a theoretical analysis to investigate the effectiveness of the amplified hallucinations. Following this, we perform analyses at the encoder-level and LLM-level to validate our theoretical findings and reveal the directional impact of adversarial perturbations.

**Analysis Setting**    We conduct our analysis under a standardized setting similar to prior work (Jiang et al., 2024b; Yang et al., 2025a; Kang et al., 2025). In particular, we use a general image description prompt (e.g., "Please describe this image in detail.") to elicit model outputs for a set of evaluation images (1,000 randomly selected MS COCO (Lin et al., 2014) validation data). To focus our analysis on how image distortion drives hallucination, we examine only tokens corresponding to actual objects in the undistorted image. We utilize the LLaVA-1.5 7B model (Liu et al., 2023) and generate captions for each image using greedy decoding. For consistency, if an object in the image is described by multiple tokens (e.g., an object consisting of two tokens), we consider only the first token of that object for our analysis.

## 3.1 HALLUCINATION AMPLIFICATION EFFECT OF ADVERSARIAL PERTURBATION

We empirically show that adversarial perturbations more effectively amplify LVLM hallucinatory tendencies than random noise. We quantify this effect by measuring the increase in the model's prediction uncertainty for the next token. Specifically, we compute the difference in entropy between the next token probability distributions generated from the distorted image versus the original one:

$$\text{Ent}(p(\cdot|\mathbf{v}', \mathbf{x}, \mathbf{y}_{<k})) - \text{Ent}(p(\cdot|\mathbf{v}, \mathbf{x}, \mathbf{y}_{<k})).$$

Here, $\text{Ent}(\cdot)$ is the entropy function, and $p(\cdot|...)$ is the next-token probability distribution. To isolate the impact of the visual distortion, we keep the input text prompt $\mathbf{x}$ and the response $\mathbf{y}_{<k}$ identical.

As Figure 2 illustrates, adversarial perturbation consistently produces a greater increase in entropy than Gaussian noise across various distortion strengths ($\epsilon$). This suggests that adversarial perturbations are more effective at reducing the model's confidence in its factual visual grounding, a common precursor to hallucination. Furthermore, we corroborate these findings by analyzing the shift in the output probability distribution, arriving at similar conclusions (as detailed in Appendix D). Taken together, these results provide strong empirical evidence that adversarial perturbations lead to greater prediction uncertainty and a more significant deviation in output distributions from the

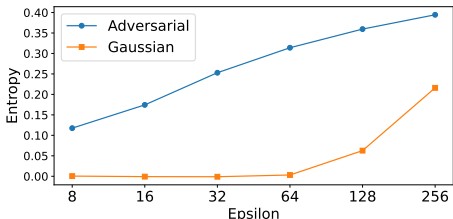

Figure 2: Entropy increase for next token probability distribution as a function of distortion strength ($\epsilon$). Adversarial perturbation leads to a greater increase in entropy.

original. This heightened amplification of hallucinatory signal is hypothesized to provide a stronger contrastive signal for AVCD's decoding step, leading to more effective hallucination suppression.

## 3.2 THEORETICAL ANALYSIS ON ADVERSARIAL PERTURBATION

In the preceding analysis, we established that adversarial perturbations are more adept than Gaussian noise at amplifying hallucinatory tendencies in LVLMs. However, it remains unclear whether these induced hallucinations are directionally different—i.e., whether adversarial perturbations push the model's outputs in a way that is conducive to contrastive decoding. To investigate this, we conduct a theoretical analysis of the directional impact of adversarial perturbations.

We demonstrate that an adversarially perturbed image can become more aligned in the embedding space with text describing arbitrary, non-present objects than with its own ground-truth objects. Let $\mathbf{f}(\cdot)$ be a trained image encoder and $\mathbf{g}(\cdot)$ a text encoder (e.g., CLIP) jointly trained to align their embeddings. Let $x$ represent the ground-truth text describing objects in image $I$, and let $x'$ be a random text describing objects not present in $I$. For well-trained system, we can posit the following.

**Proposition 1.** *The embedding of an image $I$ is closer in cosine similarity to the embedding of the text $x$ describing its true objects than to that of any text $x'$ describing random objects.*

$$\cos(\mathbf{f}(I), \mathbf{g}(x)) > \cos(\mathbf{f}(I), \mathbf{g}(x')). \tag{1}$$

Here, $\cos(\cdot)$ denotes cosine similarity. To formalize the directional impact of minimizing a similarity-based loss like $L_{\text{adv}}$, we establish Theorem 1, which proves the existence of a perturbation $\delta$ that inverts this similarity relationship with high probability.

**Theorem 1.** *Given an image $I$, its ground-truth text $x$ describing objects, and an arbitrary random text $x' \neq x$ describing random objects (chosen independently), then for a sufficiently large perturbation budget $\epsilon$, an untargeted adversarial perturbation $\delta$ with $||\delta|| \leq \epsilon$ satisfies the following:*

$$\mathbb{P}\left[\cos(\mathbf{f}(I + \delta), \mathbf{g}(x)) < \cos(\mathbf{f}(I + \delta), \mathbf{g}(x'))\right] \geq 1 - 2\exp(-c \cdot r) \tag{2}$$

*where $r$ is the rank (effective dimension) of the image embedding subspace and $c > 0$ is a constant determined by the concentration of measure.*

This theoretical result proves that adversarial perturbation does not merely increase uncertainty. Instead, it actively induces hallucinations by steering the image embedding away from its corresponding text ($x$) and towards the semantic direction of an unrelated text ($x'$). This provides a formal basis for the phenomenon of directional steering. The proof of Theorem 1 is provided in Appendix B.

To empirically verify this, we conduct a two-pronged investigation. First, at the encoder level, we directly test the core inequality of the theorem by analyzing whether the similarity of a distorted image embedding to its ground-truth text $x$ falls below its similarity to a random text $x'$. Second, to observe the downstream effects, we analyze the model at the LLM level by comparing its visual attention patterns when conditioned on ground-truth versus random captions. This comparison provides insight into how the embedding-level shift manifests in the LVLM's inference process.

### 3.3 EMPIRICAL VERIFICATION AT THE ENCODER LEVEL

We begin empirical validation at the encoder level by analyzing the change in image-text similarity. To construct text inputs, we define the ground-truth (GT) text, $x$, by extracting ground-truth objects from its caption and placing them into a template sentence: "There are [object1], [object2],...". For the random text, $x'$, we use a corresponding object list from a different, randomly selected image.

**Relative Similarity Change** For this analysis, we utilize the vision encoder from LLaVA (Liu et al., 2023), which is the standard CLIP (Radford et al., 2021) vision encoder ($\mathbf{f}$). For the text side, we use the original CLIP text encoder $\mathbf{g}$, which is not part of LLaVA's generative process but enables a direct comparison in the shared CLIP embedding space. We then measure the cosine similarity of a distorted image embedding, $\mathbf{f}(I + \delta)$, against both the GT text embedding $\mathbf{g}(x)$ and the random text embedding $\mathbf{g}(x')$. For notational simplicity, we use $\delta$ to represent a general distortion. To quantify the directional steering effect, we analyze the relative similarity change:

$$\frac{\cos(\mathbf{f}(I + \delta), \mathbf{g}(x)) - \cos(\mathbf{f}(I + \delta), \mathbf{g}(x'))}{\cos(\mathbf{f}(I + \delta), \mathbf{g}(x))} \times 100.$$

A positive value indicates the distorted image is more similar to the GT text, while a negative value indicates it is more similar to random text,, confirming the inversion predicted by Theorem 1.

**Analysis of Similarity Change** Figure 3 shows the relative similarity change as a function of distortion magnitude. While both types of distortions reduce similarity as expected, their effects are qualitatively different. For images with Gaussian noise, the value diminishes but remains positive, indicating the image's semantic identity is weakened but preserved. In contrast, for images with adversarial perturbations, the score crosses zero and becomes negative as $\epsilon$ increases. This result provides empirical proof for Theorem 1, confirming that the adversarial perturbation is not just degradative but di-

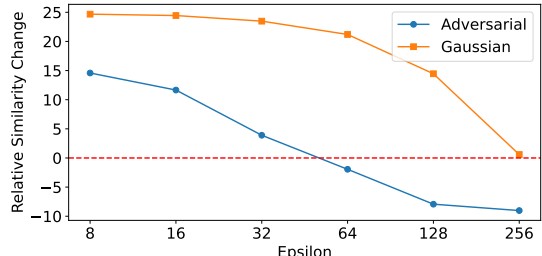

Figure 3: Relative similarity change as a function of distortion strength ($\epsilon$). Adversarial perturbation causes a much sharper decline in feature similarity, uniquely pushing it below zero.

rectional—actively steering the image embedding away from its ground-truth semantics and towards those of unrelated objects, thereby creating the conditions for directional hallucinations.

### 3.4 EMPIRICAL VERIFICATION AT THE LLM LEVEL

To understand how the embedding-level shift translates to the LVLM's generative process, we probe its internal grounding mechanisms using the Visual Attention Ratio (VAR). This metric quantifies how much a generated object token attends to the visual tokens derived from the input image.

**Visual Attention Ratio** For a given object token $y_k$ in the generated sequence and the sequence of $N$ visual tokens $\mathbf{v}$, we calculate VAR as the sum of attention weights from $y_k$ to all visual tokens. If $A_h^l(y_k, v_j)$ represents the normalized attention weight assigned to visual token $v_j$ by object token $y_k$ in attention head $h$ in layer $l$, then VAR is expressed as:

$$\text{VAR}_h^l(y_k, \mathbf{v}) = \sum_{j=1}^{N} A_h^l(y_k, v_j)$$

VAR serves as a direct measure of a token's visual grounding. A diminished VAR value suggests it is likely a hallucinatory output, stemming from insufficient visual evidence.

**Analysis of VAR**  To compare the impact of different distortions, we introduce the Difference Sum VAR (DSVAR), which measures the total change in VAR between a distorted image ($\mathbf{v}'$) and the original image ($\mathbf{v}$). As layer 0 can exhibit distinct behavior (Jiang et al., 2024b), we calculate DSVAR by summing the VAR changes across all heads and all subsequent layers ($l \geq 1$):

$$\text{DSVAR}(y_k, \mathbf{v}, \mathbf{v}') = \sum_{l=1} \sum_h \text{VAR}_h^l(y_k, \mathbf{v}') - \text{VAR}_h^l(y_k, \mathbf{v}).$$

DSVAR quantitatively compares how adversarial and Gaussian distortions impact the model's visual grounding for specific tokens. At the LLM level, we hypothesize that while any major distortion weakens visual grounding for ground-truth (GT) captions, only adversarial perturbation creates a misleading alignment with random captions. To test this, we measure the average DSVAR for object tokens, using prefixes from both GT and random captions. The results are illustrated in Figure 4.

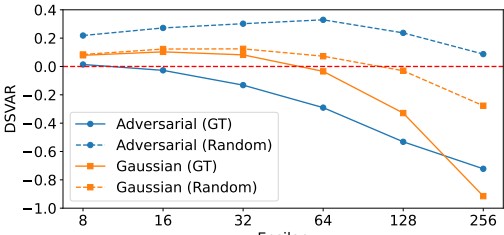

Our results confirm these hypotheses. For GT captions, both distortion types lead to a negative DSVAR. For random captions, however, their effects diverge. As predicted, Gaussian noise yields a near-zero or negative DSVAR. In stark contrast, adversarial perturbation induces a consistently positive DSVAR. This result is critical evidence for directional steering at the LLM level. While both distortion types degrade visual grounding for correct objects, only adversarial perturbation successfully creates spurious grounding for incorrect, random objects. It does not merely break the model's connection to the image; instead, it actively forges a false one. This provides a clear mechanism for how directional hallucinations are induced and, crucially, how they can be suppressed through contrastive decoding.

Figure 4: DSVAR score as a function of distortion strength ($\epsilon$). For ground-truth (GT) captions, adversarial perturbation leads to a more significant reduction in DSVAR than Gaussian noise.

In summary, our analyses reveal why adversarial perturbation is superior for contrastive decoding. Unlike Gaussian noise, we demonstrate that adversarial perturbation is uniquely directional. This directional steering is proven theoretically (Theorem 1) and confirmed empirically at both the encoder and LLM levels, where it inverts semantic similarity and uniquely induces a positive DSVAR for random captions. Ultimately, this strong and directed signal provides a more potent contrast for hallucination suppression than the non-directional degradation from random noise.

## 4 EXPERIMENT

### 4.1 EXPERIMENT SETTINGS

**Models and Baselines**  We conduct experiments on representative LVLMs: LLaVA (Liu et al., 2023) and Qwen-VL (Bai et al., 2023). For each model, we evaluate the performance of our proposed AVCD against several baselines. These include the model's default decoding performance and four contrastive decoding methods: Visual Contrastive Decoding (VCD) (Leng et al., 2024), Instructive Contrastive Decoding (ICD) (Wang et al., 2024), Multi-Modal Mutual Information Decoding (M3ID) (Favero et al., 2024), and Attentional Vision Calibration (AVISC) (Woo et al., 2024).

**Datasets and Hyperparameters**  We evaluate the methods on three standard hallucination benchmarks: POPE (Li et al., 2023), CHAIR (Rohrbach et al., 2018), and MME (Fu et al., 2023). POPE adopts a yes/no question-answering format to probe object hallucination. For CHAIR, we quantify how often the model generates hallucinated objects during image captioning. MME comprises multiple sub-tasks in LVLMs through yes/no questions.

To minimize the tuning burden and ensure a consistent comparison across all experiments, we fix the contrastive decoding hyperparameter $\alpha$ to 0.5. For distortion strength, following VCD's setup (Leng et al., 2024), we use $\epsilon = 64$ for CHAIR/MME and $\epsilon = 256$ for POPE, which correspond to diffusion noise steps 500 and 999 in VCD. For our method-specific parameters, we tune only the number of attack steps, evaluating a computationally efficient set of $T_{\text{adv}} \in \{1, 2\}$ in contrast to the commonly used 10-step PGD. The step size $\eta$ is determined formulaically by $\epsilon$ and $T_{\text{adv}}$. All non-deterministic experiments are run five times. Further details are in Appendix E.

Table 1: Evaluation on the POPE VQA hallucination benchmark, which comprises three subsets: Random, Popular, and Adversarial. The best results in each setting are bolded.

| Model | Method | Random | | Popular | | Adversarial | |
|---|---|---|---|---|---|---|---|
| | | F1 score ↑ | Accuracy ↑ | F1 score ↑ | Accuracy ↑ | F1 score ↑ | Accuracy ↑ |
| LLaVA | Sampling | $84.8 \pm 0.2$ | $86.2 \pm 0.1$ | $83.9 \pm 0.3$ | $85.3 \pm 0.3$ | $80.9 \pm 0.2$ | $81.9 \pm 0.2$ |
| | Greedy | $85.8 \pm 0.0$ | $87.2 \pm 0.0$ | $84.9 \pm 0.0$ | $86.3 \pm 0.0$ | $82.5 \pm 0.0$ | $83.6 \pm 0.0$ |
| | VCD | $87.9 \pm 0.1$ | $88.4 \pm 0.1$ | $86.9 \pm 0.2$ | $87.3 \pm 0.2$ | $82.5 \pm 0.2$ | $82.2 \pm 0.2$ |
| | ICD | $85.3 \pm 0.0$ | $86.8 \pm 0.0$ | $84.5 \pm 0.0$ | $86.0 \pm 0.0$ | $82.1 \pm 0.0$ | $83.4 \pm 0.0$ |
| | M3ID | $85.8 \pm 0.0$ | $87.2 \pm 0.0$ | $84.9 \pm 0.0$ | $86.2 \pm 0.0$ | $82.6 \pm 0.0$ | $\mathbf{83.7 \pm 0.0}$ |
| | AVISC | $88.9 \pm 0.0$ | $88.6 \pm 0.0$ | $86.4 \pm 0.0$ | $85.6 \pm 0.0$ | $80.5 \pm 0.0$ | $78.0 \pm 0.0$ |
| | AVCD | $\mathbf{89.1 \pm 0.2}$ | $\mathbf{89.6 \pm 0.2}$ | $\mathbf{87.2 \pm 0.1}$ | $\mathbf{87.5 \pm 0.1}$ | $\mathbf{83.2 \pm 0.1}$ | $82.8 \pm 0.1$ |
| Qwen-VL | Sampling | $83.7 \pm 0.1$ | $85.6 \pm 0.1$ | $83.3 \pm 0.3$ | $85.2 \pm 0.2$ | $81.7 \pm 0.5$ | $83.5 \pm 0.4$ |
| | Greedy | $83.9 \pm 0.0$ | $85.9 \pm 0.0$ | $83.5 \pm 0.0$ | $85.5 \pm 0.0$ | $82.1 \pm 0.0$ | $83.9 \pm 0.0$ |
| | VCD | $87.8 \pm 0.1$ | $88.7 \pm 0.1$ | $86.7 \pm 0.0$ | $87.6 \pm 0.0$ | $84.0 \pm 0.1$ | $84.5 \pm 0.1$ |
| | ICD | $83.5 \pm 0.0$ | $85.4 \pm 0.0$ | $83.4 \pm 0.0$ | $85.3 \pm 0.0$ | $81.8 \pm 0.0$ | $83.6 \pm 0.0$ |
| | M3ID | $85.4 \pm 0.0$ | $87.0 \pm 0.0$ | $84.8 \pm 0.0$ | $86.4 \pm 0.0$ | $83.2 \pm 0.0$ | $84.6 \pm 0.0$ |
| | AVISC | $83.8 \pm 0.0$ | $85.8 \pm 0.0$ | $83.5 \pm 0.0$ | $85.4 \pm 0.0$ | $82.1 \pm 0.0$ | $84.0 \pm 0.0$ |
| | AVCD | $\mathbf{88.0 \pm 0.1}$ | $\mathbf{88.9 \pm 0.1}$ | $\mathbf{86.9 \pm 0.1}$ | $\mathbf{87.7 \pm 0.1}$ | $\mathbf{84.5 \pm 0.1}$ | $\mathbf{85.0 \pm 0.1}$ |

**Computational Cost**   The computational cost of our method stems from the one-time generation of an adversarial example per prompt. This overhead is most pronounced in single-word answer tasks (e.g., POPE), leading to a modest but noticeable $1.22\times$ and $1.43\times$ increase over the VCD baseline for our 1-step and 2-step versions. Conversely, for longer tasks like captioning (e.g., CHAIR, which generates responses up to 64 tokens), this fixed cost is amortized over the generation process, resulting in a nearly negligible increase in inference time (approx. $1.03\times$ for 2-step AVCD). We characterize this overhead as context-dependent; while it presents a trade-off for short-latency tasks, we posit that it remains reasonable for general deployment, particularly as modern LVLMs trend toward generating longer, explanatory responses where the fixed cost becomes negligible.

## 4.2 EXPERIMENT RESULTS

**Results on POPE**   On the POPE object-existence benchmark, AVCD demonstrates superior performance. As shown in Table 1, AVCD achieves the highest F1 scores in the vast majority of settings for both LLaVA and Qwen-VL. For instance, on the LLaVA model's Random subset, AVCD reaches an F1 score of 89.1, outperforming both the greedy decoding (85.8) and the strong VCD baseline (87.9). This indicates that AVCD makes models more truthful when verifying object presence, effectively mitigating these assertion-based hallucinations. The directed signal from the adversarial perturbation likely provides a clearer contrast, enabling the model to reject non-existent objects.

**Results on CHAIR**   On the CHAIR benchmark, AVCD reduces the sentence-level hallucination rate, consistently outperforming both default decoding and the VCD baseline (Table 2). This consistent improvement is attributed to AVCD's use of adversarial perturbations, which more effectively target latent hallucination pathways than the undirected degradation from random noise.

**Results on MME**   Finally, on the MME benchmark, AVCD again achieves the highest overall score across all models (Table 3). AVCD's superior performance on this benchmark demonstrates that its benefits are not confined to specialized hallucination metrics. Instead, it enhances a model's overall reliability and accuracy across a diverse set of vision-language tasks.

**Summary of Results**   Across three benchmarks, AVCD consistently demonstrates a generalizable improvement in reducing LVLM hallucinations. While outperforming all baselines, the performance margin is more pronounced on LLaVA, an observation that sheds light on how architectural choices impact inference-time control. We posit that LLaVA's direct MLP projector offers higher feature-space fidelity, making it highly responsive to our directional steering, whereas Qwen-VL's compressive adapter may attenuate such fine-grained signals. This interaction represents a valuable finding for future research. Nonetheless, the consistent gains across diverse models underscore the fundamental benefit of leveraging directional adversarial signals for contrastive decoding.

Table 2: Evaluation on the MS COCO validation set. $C_S$ and $C_I$ correspond to CHAIR$_S$ and CHAIR$_I$, which represent the proportion of hallucinated captions and hallucinated objects within each caption, respectively. Lower values indicate fewer hallucinations. Default refers to the native decoding method. The best results in each setting are bolded.

| Method | Sampling | | | | Greedy | | | |
| | LLaVA | | Qwen-VL | | LLaVA | | Qwen-VL | |
| | $C_S \downarrow$ | $C_I \downarrow$ | $C_S \downarrow$ | $C_I \downarrow$ | $C_S \downarrow$ | $C_I \downarrow$ | $C_S \downarrow$ | $C_I \downarrow$ |
|---|---|---|---|---|---|---|---|---|
| Default | $28.1 \pm 1.3$ | $9.1 \pm 0.3$ | $26.8 \pm 1.5$ | $7.7 \pm 0.3$ | $26.9 \pm 1.3$ | $8.9 \pm 0.4$ | $25.8 \pm 1.1$ | $6.6 \pm 0.3$ |
| VCD | $26.4 \pm 1.3$ | $9.3 \pm 0.7$ | $27.4 \pm 1.4$ | $7.8 \pm 0.5$ | $25.2 \pm 2.2$ | $8.6 \pm 0.8$ | $26.2 \pm 1.4$ | $7.7 \pm 0.4$ |
| ICD | $27.0 \pm 1.6$ | $9.9 \pm 0.5$ | $25.6 \pm 1.5$ | $7.2 \pm 0.5$ | $26.1 \pm 1.5$ | $9.8 \pm 0.5$ | $25.3 \pm 0.2$ | $6.4 \pm 0.4$ |
| M3ID | $25.1 \pm 1.5$ | $8.2 \pm 0.6$ | $25.5 \pm 2.2$ | $\mathbf{7.1 \pm 0.5}$ | $25.8 \pm 2.2$ | $8.3 \pm 0.8$ | $24.0 \pm 1.1$ | $\mathbf{6.0 \pm 0.3}$ |
| AVISC | $26.6 \pm 1.7$ | $8.6 \pm 0.8$ | $26.4 \pm 1.7$ | $7.6 \pm 0.7$ | $26.1 \pm 0.8$ | $8.8 \pm 0.4$ | $\mathbf{23.9 \pm 1.3}$ | $6.3 \pm 0.5$ |
| AVCD | $\mathbf{24.0 \pm 0.9}$ | $\mathbf{8.0 \pm 0.4}$ | $\mathbf{25.4 \pm 1.4}$ | $8.0 \pm 0.6$ | $\mathbf{22.0 \pm 1.6}$ | $\mathbf{7.3 \pm 0.2}$ | $24.2 \pm 1.6$ | $7.1 \pm 0.6$ |

Table 3: Evaluation on the MME benchmark. The best performances within are bolded.

| Method | LLaVA | Qwen-VL |
|---|---|---|
| Sampling | $1356.1 \pm 35.6$ | $1256.8 \pm 10.7$ |
| Greedy | $1449.9 \pm 0.00$ | $1253.8 \pm 0.00$ |
| VCD | $1448.9 \pm 6.73$ | $1310.7 \pm 2.74$ |
| ICD | $1451.1 \pm 0.00$ | $1207.9 \pm 0.00$ |
| M3ID | $1417.5 \pm 0.00$ | $1263.1 \pm 0.00$ |
| AVISC | $1448.4 \pm 0.00$ | $1266.6 \pm 0.00$ |
| AVCD (ours) | $\mathbf{1498.0 \pm 17.0}$ | $\mathbf{1340.3 \pm 4.32}$ |

Table 4: Ablation of adversarial loss functions using the LLaVA model with greedy decoding. Performance is reported with F1 score on the POPE Random subset, CHAIR$_S$, and the MME score.

| Method | POPE $\uparrow$ | CHAIR $\downarrow$ | MME $\uparrow$ |
|---|---|---|---|
| Greedy | $85.8 \pm 0.0$ | $26.9 \pm 1.3$ | $1450 \pm 0.0$ |
| VCD | $87.9 \pm 0.1$ | $25.2 \pm 2.2$ | $1449 \pm 6.7$ |
| $L_{\cos}$ | $\mathbf{89.1 \pm 0.1}$ | $\mathbf{22.0 \pm 1.5}$ | $\mathbf{1498 \pm 17}$ |
| $L_{\text{norm}}$ | $88.9 \pm 0.1$ | $23.6 \pm 1.4$ | $1494 \pm 15$ |
| $L_{\text{comb}}$ | $\mathbf{89.1 \pm 0.0}$ | $23.0 \pm 1.2$ | $\mathbf{1498 \pm 8.6}$ |

### 4.3 ABLATION STUDY ON THE ADVERSARIAL LOSS

To investigate the impact of different adversarial loss functions, we conduct an ablation study comparing our primary cosine similarity loss (referred to as $L_{\cos}$ instead of $L_{\text{adv}}$) against two alternatives. The first, $L_{\text{norm}}$, maximizes feature separation by minimizing the negative L2 distance:

$$L_{\text{norm}}(I, I_{\text{adv}}) = -\sum_{j=1}^{N} ||\mathbf{f}_I^j - \mathbf{f}_{I_{\text{adv}}}^j||_2.$$

The second alternative is a combined loss, $L_{\text{comb}} = L_{\cos} + L_{\text{norm}}$. As shown in Table 4, all three adversarial configurations outperform the VCD baseline. Notably, while our primary $L_{\cos}$ loss is the most effective, all tested adversarial losses achieve significant gains over both the default and VCD baselines. These findings demonstrate the general robustness of our framework, suggesting its effectiveness hinges not on a single, finely-tuned loss, but on the broader strategy of leveraging any well-formulated adversarial loss to generate a potent contrastive signal. Additional experiment (e.g., qualitative results) and ablation studies (e.g., on $\epsilon$ and $T_{\text{adv}}$) can be found in Appendix F.

### 4.4 APPLICABILITY TO BLACK-BOX MODELS VIA ADVERSARIAL TRANSFER

A limitation of standard inference-time interventions, including the logit-based AVCD, is the requirement for internal access (gradients or logits), which restricts direct application to closed-source API models (e.g., GPT-4o, Gemini). To address this, we propose a "Transfer & Correct" strategy that leverages the adversarial transferability of our generated perturbations. Unlike random noise used in VCD, which degrades visual features stochastically, adversarial perturbations target semantic vulnerabilities shared across vision encoders (e.g., ViT-based CLIP Radford et al. (2021)).

We employ a surrogate open-source vision encoder (CLIP ViT-L/14) to generate the adversarial image $I_{adv}$ using AVCD. Since we cannot access the output logits of the black-box model, we utilize an indirect contrastive prompting pipeline consisting of two phases.

Table 5: Evaluation of Black-Box Applicability using the "Transfer & Correct" strategy. Draft indicates the intermediate output from the distorted image, and Default-* indicates the final corrected output. AVCD perturbations transfer effectively (high Draft error), leading to better final correction.

| Target Model | Method | $C_S \downarrow$ | $C_I \downarrow$ | Observation |
|---|---|---|---|---|
| GPT-4o-mini | Baseline | 13.4 | 6.8 | - |
| | Noise Draft | 100.0 | 9.0 | No Information |
| | VCD Draft | 18.6 | 14.3 | Low Transfer |
| | **AVCD Draft** | **32.0** | **50.1** | **High Transfer** |
| | Default-Noise | 13.4 | **6.4** | - |
| | Default-VCD | 14.2 | 7.4 | 6.0% Decrease |
| | **Default-AVCD** | **11.8** | **6.4** | **12.0% Increase** |
| Gemini-2.5-flash-lite | Baseline | 16.0 | 6.4 | - |
| | Noise Draft | 92.4 | 26.4 | No Information |
| | VCD Draft | 19.4 | 15.4 | Low Transfer |
| | **AVCD Draft** | **41.6** | **49.7** | **High Transfer** |
| | Default-Noise | 15.0 | **6.3** | 6.7% Increase |
| | Default-VCD | 14.8 | 7.2 | 7.5% Increase |
| | **Default-AVCD** | **13.8** | **6.3** | **13.7% Increase** |

1. **Transfer (Draft Generation):** The black-box model generates a caption based on $I_{adv}$. Due to adversarial transferability, this draft is expected to contain severe hallucinations.

2. **Correct (Refinement):** We prompt the model to generate a final description using the original image $I$, explicitly instructing it to use the hallucinated draft from step 1 (Transfer) as a negative constraint (i.e., "avoid errors found in the draft").

We evaluate this strategy on GPT-4o-mini and Gemini-2.5-flash-lite using 500 images from the CHAIR benchmark. The experimental settings are identical to those in Section 4.1, with $\epsilon = 64$ for AVCD and diffusion noise steps $t = 500$ for VCD. As shown in Table 5, the adversarial images generated by AVCD successfully transfer to these black-box models, inducing significantly higher hallucination rates in the draft phase compared to VCD (e.g., CHAIR$_S$ 41.6 vs. 19.4 on Gemini).

To rigorously investigate whether the correction gain stems merely from rejecting a highly hallucinatory draft regardless of its content, we introduced a "Noise Draft" baseline using pure Gaussian noise. While this baseline yields drafts with maximal hallucination rates (e.g., 100.0 CHAIR$_S$ on GPT-4o-mini) due to the complete absence of visual information, employing these drafts as negative constraints fails to yield meaningful improvements compared to the baseline. In contrast, "Default-AVCD" consistently achieves the lowest final hallucination rates, outperforming both the baseline and VCD-based correction. This confirms that AVCD's efficacy relies on inducing semantically misleading hallucinations that serve as potent negative constraints, unlike ungrounded noise. We present these findings to validate the unique potential of adversarial perturbations for transfer-based correction, demonstrating a capability structurally distinct from standard noise-based approaches.

## 5 CONCLUSION

In this work, we introduce Adversarial Visual Contrastive Decoding (AVCD), a simple, training-free inference strategy that effectively mitigates hallucinations in LVLMs. By replacing undirected Gaussian noise with adversarial perturbations, AVCD generates a more potent and informative contrastive signal. Our theoretical and empirical analyses confirm that these perturbations are not merely degradative but uniquely directional, actively steering the model toward hallucinatory states. Consequently, AVCD significantly outperforms existing baselines on standard benchmarks, achieving lower hallucination rates and higher accuracy scores without any model retraining. While AVCD introduces a modest, task-dependent overhead—most noticeable in single-word answer tasks but nearly negligible for longer, explanatory responses—this work underscores the broader potential of repurposing adversarial examples as constructive tools to enhance model reliability, opening future avenues for developing more robust and trustworthy AI systems.

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

# Appendix

## Appendix Contents

---

**Algorithm 1** Adversarial Visual Contrastive Decoding (AVCD)

---

1: **Input:** Original image $I$, input text token $\mathbf{x}$, vision encoder $E_V$, projector $P$, contrastive decoding hyperparameter $\alpha$, attack loss $L_{\text{adv}}$, attack margin $\epsilon$, attack step size $\eta$, attack step $T_{\text{adv}}$.
2: **Output:** Decoded output text tokens $\mathbf{y}$.

3: **function** ADV_GENERATE($I, E_V, L_{\text{adv}}, \epsilon, \eta, T_{\text{adv}}$)
4:     $\mathbf{f}_I \leftarrow E_V(I)$
5:     $I_{\text{adv}} \leftarrow I$
6:     **for** $t \leftarrow 1$ **to** $T_{\text{adv}}$ **do**
7:         $\mathbf{f}_{I_{\text{adv}}} \leftarrow E_V(I_{\text{adv}})$
8:         $I_{\text{adv}} \leftarrow I_{\text{adv}} - \eta \, \text{sign}\big(\nabla_{I_{\text{adv}}} L_{\text{adv}}(\mathbf{f}_I, \mathbf{f}_{I_{\text{adv}}})\big)$
9:         $I_{\text{adv}} \leftarrow \text{Clip}_{[I-\epsilon, \, I+\epsilon]}\big(I_{\text{adv}}\big)$
10:    **end for**
11:    **return** $I_{\text{adv}}$
12: **end function**

13: **function** AVCD($I, E_V, L_{\text{adv}}, \epsilon, \eta, T_{\text{adv}}$)
14:    $I_{\text{adv}} \leftarrow$ ADV_GENERATE($I, E_V, L_{\text{adv}}, \epsilon, \eta, T_{\text{adv}}$)
15:    $\mathbf{v} \leftarrow P(E_V(I)), \quad \mathbf{v}' \leftarrow P(E_V(I_{\text{adv}}))$
16:    $\mathbf{y} \leftarrow [\,]$
17:    **for** $k \leftarrow 1$ **to** $K$ **do**
18:        $\ell_{\text{orig}} \leftarrow \ell\big(\mathbf{y}_{<k}, \mathbf{v}, \mathbf{x}\big)$
19:        $\ell_{\text{dist}} \leftarrow \ell\big(\mathbf{y}_{<k}, \mathbf{v}', \mathbf{x}\big)$
20:        $y_k \leftarrow \arg\max S\big[(1+\alpha)\,\ell_{\text{orig}} - \alpha\,\ell_{\text{dist}}\big]$
21:        Append $y_k$ to $\mathbf{y}$
22:    **end for**
23:    **return** $\mathbf{y}$
24: **end function**

---

## A  THE ALGORITHMS OF AVCD

Algorithm 1 outlines the procedure for our proposed Adversarial Visual Contrastive Decoding (AVCD), which consists of two main stages: adversarial image generation (ADV_GENERATE) and contrastive decoding (AVCD). First, the ADV_GENERATE function creates a "hard negative" by iteratively applying a PGD attack to the original image. This attack is guided by an adversarial loss that minimizes the cosine similarity between the features of the original and perturbed images, effectively pushing their semantic representations apart.

The main AVCD function then performs autoregressive decoding. At each step, it computes logits from both the original image and the generated adversarial image. The final output is determined by contrasting these two signals—amplifying the original logits while subtracting the adversarial ones—which effectively suppresses hallucinatory tokens that are more likely to appear in the adversarial path.

## B  PROOF OF THE THEOREM

In this section, we provide proof for the theorem presented in our main paper.

**Theorem 1.** *Given an image $I$, its ground-truth text $x$ describing objects, and an arbitrary random text $x' \neq x$ describing random objects (chosen independently), then for a sufficiently large perturbation budget $\epsilon$, an untargeted adversarial perturbation $\delta$ with $||\delta|| \leq \epsilon$ satisfies the following:*

$$\mathbb{P}\left[\cos(\mathbf{f}(I+\delta), \mathbf{g}(x)) < \cos(\mathbf{f}(I+\delta), \mathbf{g}(x'))\right] \geq 1 - 2\exp(-c \cdot r) \tag{3}$$

*where $r$ is the rank (effective dimension) of the image embedding subspace and $c > 0$ is a constant determined by the concentration of measure.*

*Proof.* We begin by defining the structure of our embedding space. Let $R_I \in \mathbb{R}^d$ be the subspace of image embeddings, where $d$ is the dimensionality of the embedding space. The embedding of an image $I$ is a vector $\mathbf{f}(I) \in R_I$. Following prior works on the relationship between image and text embeddings (Merullo et al., 2023; Bhalla et al., 2024), we can approximate the text embedding $\mathbf{g}(x)$ as an affine transformation of the corresponding image embedding $\mathbf{f}(I)$. This can be expressed by decomposing $\mathbf{g}(x)$ into components parallel and orthogonal to $\mathbf{f}(I)$ and $R_I$:

$$\mathbf{g}(x) = \mathbf{g}(x)_{\|R_I} + \mathbf{g}(x)_{\perp R_I} \approx b\mathbf{f}(I) + \nu_I(x) + \mu, \tag{4}$$

where $b$ is a scalar coefficient, $\nu_I(x) \in R_I$ is a vector within the image embedding subspace that is orthogonal to $\mathbf{f}(I)$ (residual vector of the image embedding subspace), i.e., $\nu_I(x) \perp \mathbf{f}(I)$, and $\mu \in \mathbb{R}^d$ is a mean offset vector, representing the average displacement between text and image embeddings across the training dataset. We assume $\mu$ is orthogonal to the image embedding subspace, $\mu \perp R_I$. Similarly, the embedding for any other random text $x'$ can be expressed as:

$$\mathbf{g}(x') \approx c\mathbf{f}(I) + \nu_I(x') + \mu, \tag{5}$$

where $c$ is a scalar coefficient. For simplicity, all embedding vectors are assumed to be normalized, i.e., $||\mathbf{f}(I)|| = ||\mathbf{g}(x)|| = 1$. Using Proposition 1, we next establish the relationship between the scalars $b$ and $c$ based on the fundamental assumption that an image's embedding is inherently more similar to its true text description's embedding than to any random text's.

**Proposition 1.** *The embedding of an image $I$ is closer in cosine similarity to the embedding of the text $x$ describing its true objects than to that of any text $x'$ describing random objects.*

$$\cos(\mathbf{f}(I), \mathbf{g}(x)) > \cos(\mathbf{f}(I), \mathbf{g}(x')). \tag{6}$$

Let $\mathbf{f} = \mathbf{f}(I)$ and using our decomposition, the inequality becomes:

$$\cos(\mathbf{f}(I), \mathbf{g}(x)) > \cos(\mathbf{f}(I), \mathbf{g}(x')) \tag{7}$$

$$\rightarrow \mathbf{f} \cdot \mathbf{g}(x) > \mathbf{f} \cdot \mathbf{g}(x') \tag{8}$$

$$\rightarrow \mathbf{f} \cdot (b\mathbf{f} + \nu_I(x) + \mu) > \mathbf{f} \cdot (c\mathbf{f} + \nu_I(x') + \mu) \tag{9}$$

Given the orthogonality conditions ($\mathbf{f} \cdot \nu_I(x) = 0$ and $\mathbf{f} \cdot \mu = 0$) and that $||\mathbf{f}||^2 = 1$, the expression simplifies to:

$$b > c.$$

This confirms that the scalar coefficient $b$ for the true text-image pair is larger than the coefficient $c$ for a random pair, capturing the stronger alignment.

Next, we solve an optimization problem for the adversarial perturbation $\delta$. The optimization problem for the adversarial perturbation is as follows:

$$\min_{||\delta|| \leq \epsilon} \cos(\mathbf{f}(I), \mathbf{f}(I + \delta)) \quad \text{where} \quad ||\mathbf{f}(I + \delta)||^2 = 1. \tag{10}$$

Letting $\mathbf{f}(I + \delta) = h$, we can formulate the following Lagrangian equation for the optimization problem above.

$$\mathcal{L}(h, \lambda) = \mathbf{f}(I) \cdot h - \lambda(h^\top h - 1). \tag{11}$$

According to the KKT conditions, the partial derivative of the Lagrangian with respect to $h$ must be zero: $\frac{\partial \mathcal{L}}{\partial h} = \mathbf{f}(I) - 2\lambda h = 0$. This implies that $h = \frac{\mathbf{f}(I)}{2\lambda}$. In an ideal scenario, if we set $\lambda = -0.5$, then the cosine similarity, $\cos(\mathbf{f}(I), \mathbf{f}(I + \delta))$, would achieve its minimum value of $-1$. However, since the adversarial perturbation is constrained by its magnitude, $||\delta|| \leq \epsilon$, the resulting vector $h = \mathbf{f}(I + \delta)$ is not guaranteed to perfectly oppose $\mathbf{f}(I)$ ($a^2 < 1$). We therefore decompose it into a parallel component and orthogonal residual terms that ensure the unit-norm constraint ($||\mathbf{f}(I + \delta)||^2 = 1$) from our optimization is still met:

$$\mathbf{f}(I + \delta) = a\mathbf{f}(I) + \nu_I(\delta) + e_I(\delta), \tag{12}$$

where $a$ is a scalar coefficient which satisfies $-1 \leq a \leq 1$, $\nu_I(\delta) \in R_I$ is a residual error component in the image subspace orthogonal to $\mathbf{f}(I)$, and $e_I(\delta) \perp R_I$ is the residual error component, orthogonal to the image subspace. Then, we compare the cosine similarities in Theorem 1:

$$\cos(\mathbf{f}(I + \delta), \mathbf{g}(x)) = \mathbf{f}(I + \delta) \cdot \mathbf{g}(x) \tag{13}$$

$$= (a\mathbf{f} + \nu_I(\delta) + e_I(\delta)) \cdot (b\mathbf{f} + \nu_I(x) + \mu) \tag{14}$$

$$= ab + \nu_I(\delta) \cdot \nu_I(x) + e_I(\delta) \cdot \mu, \tag{15}$$

$$\cos(\mathbf{f}(I+\delta), \mathbf{g}(x')) = \mathbf{f}(I+\delta) \cdot \mathbf{g}(x') \tag{16}$$

$$= (a\mathbf{f} + \nu_I(\delta) + e_I(\delta)) \cdot (c\mathbf{f} + \nu_I(x') + \mu) \tag{17}$$

$$= ac + \nu_I(\delta) \cdot \nu_I(x') + e_I(\delta) \cdot \mu. \tag{18}$$

In high-dimensional spaces, the concentration of measure phenomenon asserts that two statistically independent vectors are nearly orthogonal with overwhelming probability. While the adversarial perturbation $\delta$ is directionally optimized relative to $\mathbf{f}(I)$, our method employs an untargeted attack using only the vision encoder. Crucially, the generation of $\delta$ is blind to the specific random text $x'$, ensuring that the perturbation residual $\nu_I(\delta)$ remains statistically independent of the random text residual $\nu_I(x')$. Applying the concentration inequality to these independent high-dimensional vectors, let $\hat{\nu}_I(\delta) = \frac{\nu_I(\delta)}{||\nu_I(\delta)||}$ and $\hat{\nu}_I(x') = \frac{\nu_I(x')}{||\nu_I(x')||}$. For any arbitrary $t' > 0$, the following holds:

$$\Pr\big[\big|\hat{\nu}_I(\delta) \cdot \hat{\nu}_I(x')\big| > t'\big] \leq 2\exp\big(-\frac{r(t')^2}{2}\big) \tag{19}$$

$$\rightarrow \big|\nu_I(\delta) \cdot \nu_I(x')\big| \leq (1 - a^2 - ||e_I(\delta)||^2)\sqrt{\frac{2\ln(2/\eta)}{r}} \tag{20}$$

$$\text{with prob.} \geq 1 - \eta \tag{21}$$

Here, $r$ denotes the rank of the image embedding space $R_I$. This implies that the probability of the cross-terms (including $\nu_I(\delta) \cdot \nu_I(x)$) being significant is exponentially low in high-dimensional spaces (by the rank $r$). This provides the theoretical basis for the probability bound $1 - 2\exp(-c \cdot r)$ stated in Theorem 1, where $c$ relates to the margin $(t')^2/2$. We thus assume these cross-terms are negligible, and the inequality is primarily driven by the main alignment terms. The theorem's condition $\cos(\mathbf{f}(I+\delta), \mathbf{g}(x)) < \cos(\mathbf{f}(I+\delta), \mathbf{g}(x'))$ therefore simplifies to:

$$ab < ac.$$

From Proposition 1, we know $b > c$. Therefore, the inequality $ab < ac$ can only be satisfied if $a < 0$. Our task is now to find the condition on $\epsilon$ that ensures $a$ becomes negative. We can approximate the perturbed embedding $\mathbf{f}(I+\delta)$ using a first-order Taylor expansion around $I$:

$$\mathbf{f}(I+\delta) \approx \mathbf{f}(I) + \nabla_I \mathbf{f}(I)\delta. \tag{22}$$

Now, we can approximate $a$:

$$a = \mathbf{f}(I) \cdot \mathbf{f}(I+\delta) \approx ||\mathbf{f}(I)||^2 + \mathbf{f}(I)^\top (\nabla_I \mathbf{f}(I)\delta). \tag{23}$$

To make $a$ negative, an attacker must minimize it. The term $\mathbf{f}(I)^\top (\nabla_I \mathbf{f}(I)\delta)$ is minimized when the perturbation $\delta$ is aligned with the negative direction of the gradient $\nabla_I \mathbf{f}(I)^\top \mathbf{f}(I)$. Given the constraint $||\delta|| \leq \epsilon$, the optimal perturbation is:

$$\delta = -\epsilon \frac{\nabla_I \mathbf{f}(I)^\top \mathbf{f}(I)}{||\nabla_I \mathbf{f}(I)^\top \mathbf{f}(I)||}. \tag{24}$$

Substituting this optimal $\delta$ gives the minimum approximate value for $a$:

$$a_{\min} \approx ||\mathbf{f}(I)||^2 - \epsilon \frac{\mathbf{f}(I)^\top \nabla_I \mathbf{f}(I) \nabla_I \mathbf{f}(I)^\top \mathbf{f}(I)}{||\nabla_I \mathbf{f}(I)^\top \mathbf{f}(I)||} \tag{25}$$

$$= ||\mathbf{f}(I)||^2 - \epsilon ||\nabla_I \mathbf{f}(I)^\top \mathbf{f}(I)||. \tag{26}$$

Provided that the gradient term $||\nabla_I \mathbf{f}(I)^\top \mathbf{f}(I)||$ is non-zero, $a_{\min}$ will become negative for a sufficiently large $\epsilon$. This leads to the approximate condition required to make $a_{\min} < 0$:

$$||\mathbf{f}(I)||^2 - \epsilon ||\nabla_I \mathbf{f}(I)^\top \mathbf{f}(I)|| < 0. \tag{27}$$

Rearranging for $\epsilon$ gives the final condition:

$$\epsilon > \frac{||\mathbf{f}(I)||^2}{||\nabla_I \mathbf{f}(I)^\top \mathbf{f}(I)||}. \tag{28}$$

This completes the proof. When the perturbation magnitude $\epsilon$ exceeds this approximate threshold, the alignment factor $a$ becomes negative, reversing the cosine similarity inequality with high probability (as determined by the dimension $r$) and causing the model to favor the incorrect object text $x'$. □

The threshold derived here is an estimate based on a first-order Taylor approximation. This approximation is most accurate when the embedding function has low curvature. In cases of high curvature or significant non-linearity, the actual $\epsilon$ required to invert the similarity may be larger than this theoretical threshold suggests. However, this does not limit our method's applicability. Since AVCD is not constrained to a specific $\epsilon$ value, our framework allows for the use of a sufficiently large budget (e.g. $\epsilon = 256$) to ensure the attack's success, as guided by this theoretical analysis. Indeed, our empirical analysis in Figure 3 validates this theory, showing that the relative similarity change becomes negative for $\epsilon \geq 64$, which confirms that the predicted inversion phenomenon occurs in practice.

**Existence Proof for Large $\epsilon$.** We acknowledge that the first-order Taylor approximation is a local estimate and may decrease in accuracy under large perturbation budgets ($\epsilon$). However, our method does not impose a strict upper bound on $\epsilon$. Therefore, rather than relying solely on the local approximation bound, we establish the validity of our theorem through a fundamental existence proof based on the continuity and codomain properties of the vision encoder. The vision encoder $f$ is a continuous function mapping from the pixel space to the embedding space, and crucially, its codomain is defined over $\mathbb{R}^d$.

Due to architectural components like Layer Normalization Ba et al. (2016) and linear projectors (which do not restrict outputs to non-negative values like ReLU), the encoder is theoretically capable of producing output vectors with negative cosine similarities relative to the original embedding $f(I)$. Since the codomain allows for vectors $v$ where $\cos(f(I), v) < 0$, and the function is continuous, by the Intermediate Value Theorem, there must exist an input perturbation $\delta$ within the feasible input space such that the output embedding creates a negative cosine similarity ($a < 0$). Once $a$ becomes negative (which is achievable with sufficiently large $\epsilon$ as proven above), the condition $ab < ac$ (derived from $b > c$) holds true. Consequently, the inequality reverses, proving that the perturbed image becomes more similar to the random text $x'$ than the ground truth $x$.

## C   RELATED WORK

**Hallucination Mitigation in LVLMs**   Researchers have investigated various approaches to address the hallucination problem in LVLMs. These include fine-tuning models on datasets specifically curated to exhibit hallucination phenomena (Zhao et al., 2023b; Yang et al., 2025b); using post-hoc revisor models to detect and correct hallucinations in generated text (Zhou et al., 2023; Yin et al., 2024); and employing Reinforcement Learning from Human Feedback (RLHF) with factually augmented data to penalize hallucinations (Yu et al., 2024a;b).

Another direction is to manipulate the model's inputs or decoding process at inference time to control hallucinations. VCD and other baselines (ICD, M3ID, and AVISC) are such inference-time method, and beyond VCD, other strategies like OPERA (Huang et al., 2024), HALC (Chen et al., 2024b), DOLA (Chuang et al., 2023), PAI (Liu et al., 2024), VTI (Liu et al., 2025), and Project-Away (Jiang et al., 2024a) have sought to reduce hallucinations by modifying the decoding strategy or using auxiliary information. Hallucinations themselves can take diverse forms (object, attribute, spatial relation), and a range of benchmarks has been proposed to measure these different types in LVLM outputs (Huang et al., 2024). Our work falls within this line of research, focusing on training-free decoding strategies—in particular, directly modifying and enhancing VCD via adversarial perturbations.

**Adversarial Attacks on LVLMs**   Adversarial attacks are techniques that introduce pixel perturbations to an input, thereby disrupting the model's visual feature extraction stage and inducing misclassifications or erroneous outputs. Early research in vision models reported numerous instances where methods like the Fast Gradient Sign Method (FGSM) (Goodfellow et al., 2014) or Projected Gradient Descent (PGD) (Madry et al., 2017) are used to subtly alter input images, leading to a decline in classification performance.

These vulnerabilities are often inherited by LVLMs, as they typically incorporate similar, pre-trained vision encoders. Consequently, similar strategies are applicable to LVLMs, causing a range of failures from simple object misidentification to the generation of confident but factually incorrect statements. For example, CroPA (Luo et al., 2023) generates a cross-prompt adversarial attack that optimizes both the image and text to mislead VLMs across hundreds of textual instructions. Simi-

larly, (Cui et al., 2024) show that many instruction-tuned multimodal models remain susceptible to PGD-style perturbations, highlighting the continued relevance of these classic attack methodologies.

**Constructive Use of Adversarial Attacks**   We leverage adversarial attack techniques for a constructive purpose rather than purely as something to defend against. This idea is beginning to emerge in other contexts as well, such as using adversarial examples to mitigate object hallucination (Zhang et al., 2025b), reduce dialog model hallucination (Park et al., 2024), or even enable more grounded reasoning in agents (Jalaian et al., 2025). Among these, the work by Zhang et al. (2025b) is particularly relevant as it also uses adversarial perturbations to mitigate LVLM hallucinations. However, their approach, VAP, is predicated on an objective function calculated from the LVLM's final, discrete text outputs. This non-differentiable objective necessitates a gradient estimation approach (Chen et al., 2017), which in practice is computationally demanding. Estimating the gradient requires multiple full LVLM inferences using varied inputs, generating responses from the original image (with and without a prompt) and from a noised image (with a prompt). This process must be repeated multiple times to estimate the gradient, making the overall cost of processing a single question exceptionally high, often requiring a double-digit number of full LVLM inferences. Moreover, the approach is inherently dependent on a pre-aligned text encoder (e.g., CLIP's) to measure the semantic similarity of text outputs, precluding its use on MLLMs built upon vision-only backbones.

In stark contrast, our method, AVCD, is designed to circumvent these limitations. Our adversarial objective is calculated exclusively on vision encoder features, making it independent of both the text prompt and any external text encoder. This allows us to generate a potent negative signal with only one or two efficient forward-backward passes through the vision encoder alone, avoiding the costly overhead of repeated full-model inferences. This makes AVCD a more direct and computationally targeted decoding strategy.

# D   ADDITIONAL ANALYSIS ON METHOD

This appendix provides additional analyses to supplement the findings in Section 3. We offer two further empirical investigations: (1) an analysis of the output probability distribution shift using Hellinger distance, and (2) a detailed, token-level visualization of the attention differences that underlie the main paper's DSVAR results.

## D.1   OUTPUT DISTRIBUTION SHIFT

To quantify the impact of visual distortions on the model's generative process, we measure the shift in the next token probability distribution. We employ the Hellinger distance (Hellinger, 1909), a symmetric metric that quantifies the similarity between two probability distributions, to compare the output distribution from the distorted image against that of the original. A larger distance signifies a more substantial change in the model's predictive behavior. The Hellinger distance is calculated as:

$$\frac{1}{\sqrt{2}} \sqrt{\sum \left( \sqrt{p(y_i|\mathbf{v}', \mathbf{x}, \mathbf{y}_{<i})} - \sqrt{p(y_i|\mathbf{v}, \mathbf{x}, \mathbf{y}_{<i})} \right)^2}$$

A higher Hellinger distance indicates a more significant divergence in the output probabilities, suggesting that the visual distortion has substantially altered the model's predictive certainty. However, a large shift alone does not distinguish between a targeted, misleading effect and a simple collapse of visual grounding. To ensure the distortion is not merely causing the model to ignore the visual input and revert to its language priors, we introduce a critical baseline: we also measure the Hellinger distance between the distribution from the distorted image and the distribution generated with no image at all ($p(y_i|\mathbf{x}, \mathbf{y}_{<i})$). This comparison allows us to verify if the model is being actively steered toward a different visual interpretation, rather than simply defaulting to a language-only generation mode.

Figure 5 presents these Hellinger distances as a function of distortion strength $\epsilon$. The results clearly show that adversarial perturbation induces a significantly larger Hellinger distance from the original image's output distribution compared to Gaussian noise. This indicates that adversarial perturbations push the model's output further away from the ground truth (as represented by its output for the original image), effectively amplifying the incorrectness or ungrounded nature of its predictions. Moreover, adversarial perturbations remain distant from the No Image baseline, confirming that

the adversarial perturbations are not merely causing the model to revert to a purely language-based mode. Instead, they are actively causing the model to generate outputs that are specifically ungrounded with respect to the visual input, which is the hallmark of visual hallucination that AVCD aims to mitigate.

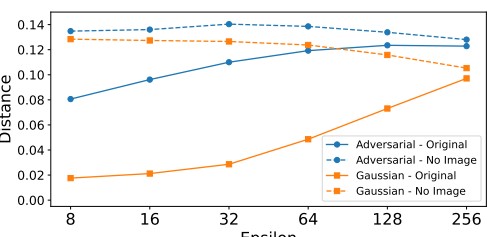

Figure 5: Hellinger distance between output distributions versus distortion strength ($\epsilon$). Adversarial perturbations cause a larger deviation from the original image's distribution.

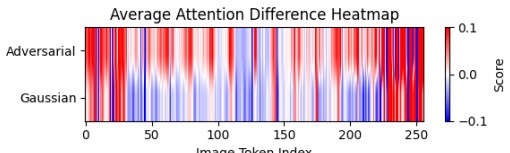

Figure 6: Average attention difference heatmap for each image token, conditioned on random captions. The x-axis represents the image token index of the LLaVA model (from 0 to 255). The color indicates the average attention difference: red for positive (increased attention) and blue for negative (decreased attention).

### D.2 DETAILED VISUALIZATION OF ATTENTION DIFFERENCES

To provide a more granular understanding of the aggregated DSVAR results presented in Section 3.4, we visualizes the token-level attention differences between the distorted and original images. Figure 6 presents a heatmap of the average attention difference for each individual visual token (not the summation as in the main paper), conditioned on random (non-GT) captions. This visualization breaks down the overall DSVAR score, showing how each visual token's attention weight contributes to the total sum.

The similar, strong patterns observed at the beginning and end of the token sequence are likely artifacts resulting from the padding used to standardize the input image size to a fixed resolution (e.g., 224x224). In the central, content-bearing region of the image, a clear dichotomy emerges. For Gaussian noise, the attention differences are predominantly negative (blue). This indicates that when prompted with a random caption, the Gaussian-noised image causes the model to pay less attention to its visual features compared to the original image. In stark contrast, for adversarial perturbation, the attention differences are predominantly positive (red). This demonstrates that the adversarial image does not simply degrade attention; it actively and broadly increases the model's attention to its visual tokens, effectively creating a false grounding for the non-present objects in the random caption.

This token-level visualization directly explains the aggregated DSVAR results reported in the main text. The sum of many small negative differences for Gaussian noise results in an overall negative or near-zero DSVAR. Conversely, the sum of the widespread positive differences for the adversarial case leads to a strongly positive DSVAR. This provides compelling visual evidence that adversarial perturbations create a targeted, misleading visual alignment, which is the core mechanism behind AVCD's effectiveness.

## E EXPERIMENT DETAILS

**Models and Baselines** We conduct experiments on representative LVLMs: LLaVA (Liu et al., 2023) and Qwen-VL (Bai et al., 2023). We experiments on four contrast-based methods as our primary baselines: visual contrastive decoding (VCD) (Leng et al., 2024), instructive contrastive decoding (ICD) (Wang et al., 2024), multi-modal mutual information decoding (M3ID) (Favero et al., 2024), and Attentional Vision Calibration (AVISC) (Woo et al., 2024). Like VCD, ICD applies contrastive decoding; however, instead of generating distorted images, it prepends a hallucination-inducing pre-prompt (e.g., "You are a confused detector") to the main prompt to produce hallucinations, and then uses them for contrastive decoding. M3ID also relies on contrastive decoding but compares against a no-image condition—correcting for the difference so that the model em-

phasizes visual information more strongly. AVISC introduces another perspective, identifying that LVLMs often over-rely on a few uninformative image tokens, which it calls blind tokens. It then performs contrastive decoding against the output conditioned only on these blind tokens to calibrate the model's attention. We use each method's default configuration.

**Implementation Details**   To ensure the reproducibility of our experiments, we provides a detailed breakdown of the hyperparameters and implementation settings used for both the baseline methods and our proposed AVCD.

- Contrastive Decoding Strength ($\alpha$): For all contrastive decoding methods, the strength hyperparameter $\alpha$ is consistently set to 0.5. While the $\alpha$ value is unified for a controlled comparison, other method-specific hyperparameters for the baseline models are set according to their respective original papers and implementations.

- Distortion Strength ($\epsilon$): The magnitude of the distortion, controlled by $\epsilon$, is carefully set for each dataset. We use $\epsilon = 64$ for the CHAIR and MME benchmarks and a larger budget of $\epsilon = 256$ for the POPE benchmark. To ensure a fair comparison, we adopt these values from the experimental setup of VCD. These values are selected to align with the noise levels produced by diffusion steps 500 and 999, respectively, in the VCD method, thereby allowing for a direct comparison between the effects of undirected random noise and our directional adversarial perturbations.

- AVCD-Specific Parameters: Our AVCD method utilizes a PGD-based attack, which requires setting the number of attack steps, $T_{\text{adv}}$, and the step size, $\eta$. For the main experiments, considering computational cost, we evaluate efficient configurations where $T_{\text{adv}} \in \{1, 2\}$. The step size $\eta$ is set based on a principled approach for exploring the perturbation space. For the single-step attack ($T_{\text{adv}} = 1$), we set the step size to the full perturbation budget, $\eta = \epsilon$. For multi-step attacks, we set the step size to ensure the total potential travel distance slightly exceeds the budget, following the rule $T_{\text{adv}} \times \eta = 1.25\epsilon$, which allows the attack to effectively explore the entire boundary of the $\epsilon$-ball. This multi-step rule applies to both our main experiment with $T_{\text{adv}} = 2$ (yielding $\eta = 0.625\epsilon$) and our deeper analysis experiment using $T_{\text{adv}} = 10$ (yielding $\eta = 0.125\epsilon$). Through this limited evaluation, the optimal number of steps was found to be $T_{\text{adv}} = 1$ for the POPE benchmark, while $T_{\text{adv}} = 2$ yielded the best results for CHAIR and MME.

- Decoding and Statistical Reliability: For sampling-based decoding, a temperature of 0.5 is used. To account for the inherent randomness, all experiments are run five times with different random seeds, with the exception of fully deterministic settings (e.g., greedy decoding with using full dataset), which are run once as their outputs are constant. We report the mean and standard deviation for all non-deterministic results. At the start of each run we fix all sources of randomness by setting the Python built-in RNG (random.seed(args.seed)), the PYTHONHASHSEED environment variable, NumPy's RNG (np.random.seed(args.seed)), and PyTorch's RNGs for both CPU (torch.manual_seed(args.seed)) and CUDA (torch.cuda.manual_seed(args.seed)) to the same seed value. We repeat experiment with five different seeds (1–5).

- All experiments are conducted on a dedicated Ubuntu 20.04.6 LTS (Focal Fossa) server equipped with an Intel Xeon Gold 6258R CPU running at 2.70GHz and a single NVIDIA A40 GPU with 46 GiB of on-card memory. The host system features 440 GiB of DDR4 RAM (with a 39GiB swap partition that remains unused during benchmarking). All code is written in Python 3.8.16, and model implementation and training are carried out using PyTorch 2.4.1 and the HuggingFace Transformers library version 4.33.2.

**Datasets and Evaluation Metrics**   We evaluate the methods using three standard hallucination benchmarks: POPE (Li et al., 2023), CHAIR (Rohrbach et al., 2018), and MME (Fu et al., 2023).

- POPE (polling-based object probing evaluation) adopts a binary question–answering format (for example, "Is there a [object] in the image?") to probe object hallucination. POPE constructs questions with a 50:50 ratio of objects that actually appear in the image versus objects that do not. For the non-existent objects, POPE select them according to one of three strategies: random selection, popular objects that frequently occur in the dataset, or

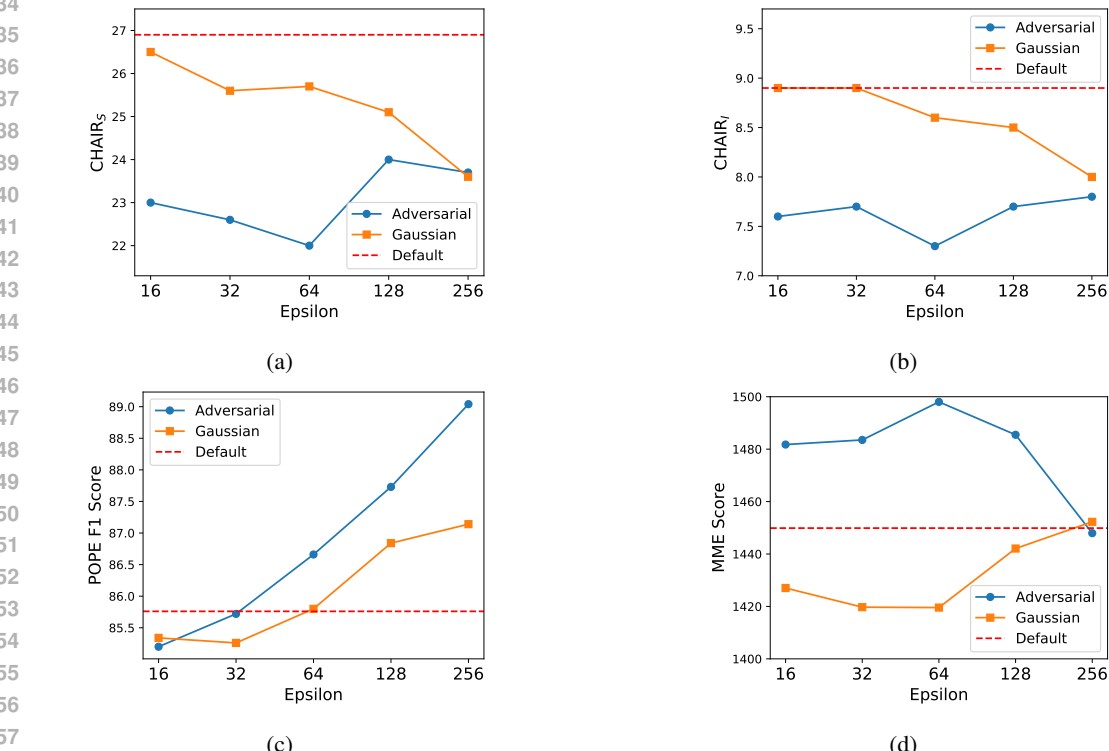

Figure 7: Ablation of adversarial noise strength. (a) CHAIR$_S$, (b) CHAIR$_I$, (c) F1 score for the POPE dataset, and (d) aggregate MME score plotted against noise magnitude $\epsilon$.

  adversarial objects that co-occur with objects present in the image (chosen at random from that co-occurrence set).

- For CHAIR (caption hallucination assessment with image relevance), we quantify how often the model generates incorrect objects (hallucinated objects) during image captioning. CHAIR provides two variants: CHAIR$_I$ (instance-level), which we compute by dividing the number of hallucinated objects by the total number of objects mentioned in a caption, and CHAIR$_S$, which represents the proportion of captions containing at least one hallucinated object. Lower CHAIR values correspond to fewer hallucinations. To perform this evaluation, we generate captions for 500 randomly selected images from the MS COCO (Lin et al., 2014) validation set using the prompt "Please describe this image in detail."

- MME (multi-modal LLM evaluation benchmark) comprises multiple sub-tasks in LVLMs. We use MME to detect object-level and attribute-level hallucinations through yes/no questions. Beyond simply verifying object presence, MME also probes visual details such as attributes, locations, and counts; this comprehensive approach reveals hallucinations that arise when the model relies primarily on linguistic priors. We follow the official MME evaluation protocol to compare performance.

# F ABLATION STUDIES AND METHODOLOGICAL DESIGN

## F.1 ABLATION STUDY ON ADVERSARIAL NOISE STRENGTH

To explore the impact of distortion strength ($\epsilon$), a crucial hyperparameter for our method, we ablate its magnitude from 16 to 256. The results, plotted in Figure 7, reveal a clear pattern related to the format of the downstream task. For the open-ended captioning of CHAIR and the complex verification task of MME, performance is optimal at a moderate perturbation of $\epsilon = 64$. In contrast, for the simple yes/no object existence questions of POPE, the F1 score peaks at the maximum tested value of $\epsilon = 256$. This suggests that binary verification benefits from the strongest possible contrastive

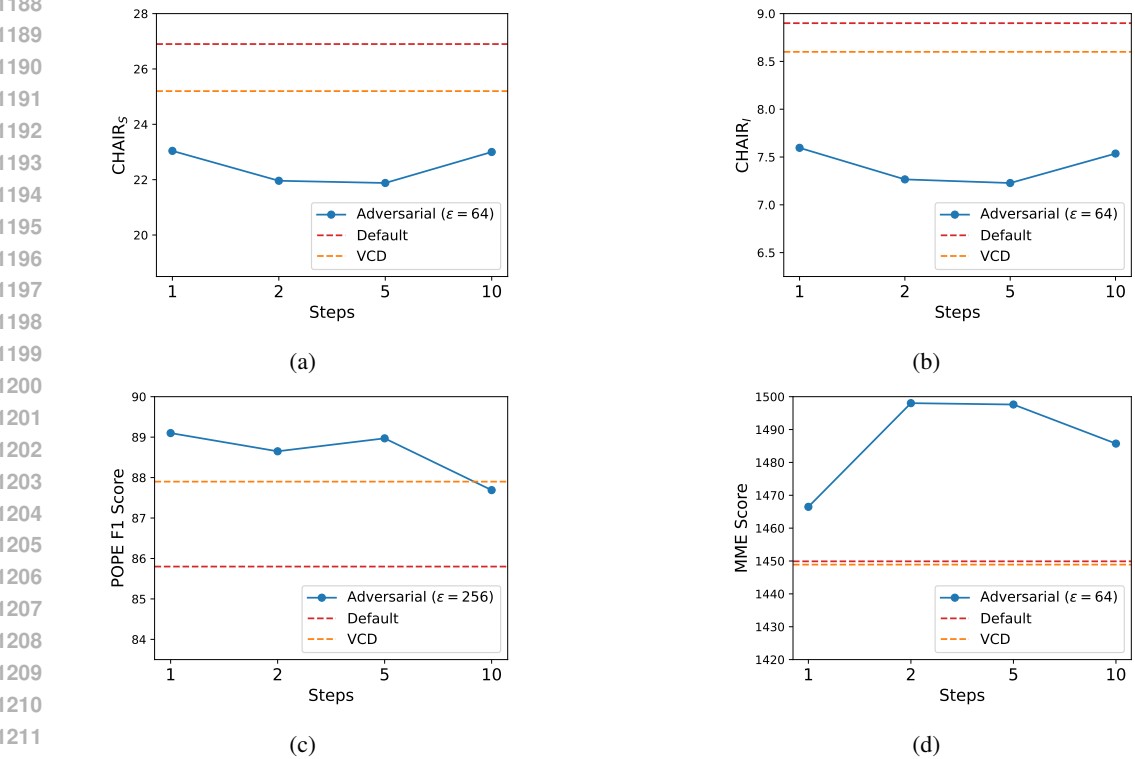

Figure 8: Ablation of the number of adversarial attack steps. (a) CHAIR$_S$, (b) CHAIR$_I$, (c) F1 score for the POPE dataset, and (d) aggregate MME score plotted against attack steps $T_{\text{adv}}$.

signal, while generative and complex tasks require a more nuanced perturbation to preserve output quality.

Table 6: Ablation study on distortion magnitude $\epsilon$ and attack steps $T_{\text{adv}}$. Bold indicates the best performance. (*) 2 steps for CHAIR and MME, and 1 step for POPE ($\epsilon \geq 64$).

| $\epsilon$ | $T_{\text{adv}}$ | $C_S\downarrow$ | $C_I\downarrow$ | MME↑ | POPE-R↑ | POPE-P↑ | POPE-A↑ |
|---|---|---|---|---|---|---|---|
| - | - | 26.9 | 8.9 | 1449.9 | 85.8 | 84.9 | 82.5 |
| 8 | 10 | 23.1 | 8.0 | 1484.2 | 88.0 | 85.5 | 80.0 |
| 16 | 10 | 23.5 | 7.8 | 1483.2 | 87.9 | 85.8 | 80.3 |
| 32 | 10 | 23.1 | 7.6 | 1490.2 | 87.8 | 85.6 | 80.2 |
| 64 | 2/1* | **22.0** | **7.3** | **1498.0** | 86.7 | 85.0 | 82.5 |
| 128 | 2/1* | 24.0 | 7.7 | 1485.5 | 87.7 | 86.5 | 82.8 |
| 256 | 2/1* | 23.7 | 7.8 | 1448.0 | **89.1** | **87.2** | **83.2** |

## F.2 ABLATION STUDY ON THE NUMBER OF ADVERSARIAL ATTACK STEPS

We also analyze the effect of the number of PGD steps ($T_{\text{adv}}$) to understand the trade-off between attack strength and performance. We evaluate $T_{\text{adv}} \in \{1, 2, 5, 10\}$, setting the attack step size ($\eta$) based on the number of steps: $\eta = \epsilon$ for a single-step attack ($T_{\text{adv}} = 1$), and $\eta = 1.25\epsilon/T_{\text{adv}}$ for multi-step attacks. The results, shown in Figure 8, reveal a significant finding: a small number of steps (1 or 2) is not only computationally efficient but also empirically optimal or near-optimal for all tested benchmarks. Specifically, performance peaks at a single step for the simple object verification task (POPE), while the more complex tasks (CHAIR and MME) benefit from 2 to 5 steps. This result is highly advantageous, as it demonstrates that AVCD does not require a costly, high-step attack to be effective, directly countering the notion of a heavy tuning burden.

Taken together, the preceding analyses of $\epsilon$ and $T_{\mathrm{adv}}$ highlight a key advantage of our adversarial approach. While methods like VCD are limited to controlling a single dimension (noise strength), AVCD offers a more powerful and flexible control mechanism. By jointly tuning the distortion magnitude ($\epsilon$), the number of attack steps ($T_{\mathrm{adv}}$), and the step size ($\eta$), our method enables a tailored, multi-dimensional optimization for diverse tasks. This allows for a level of precision in crafting the contrastive signal that is fundamentally unavailable in methods relying on simple Gaussian noise, leading to the superior performance shown in our experiments.

### F.3    MAGNITUDE VS. DEPTH

To rigorously justify our parameter design, we conduct a comparative study between the traditional adversarial setting (small $\epsilon$, high $T_{\mathrm{adv}}$) and our proposed AVCD setting (large $\epsilon$, low $T_{\mathrm{adv}}$). Historically, adversarial attacks operate under strict imperceptibility constraints (e.g., $\epsilon \approx 8/255$), necessitating small perturbations optimized over many steps (e.g., 10 steps). However, in AVCD, the adversarial image is an internal intermediate artifact never exposed to the user, liberating us from these visual constraints.

As shown in Table 6, while the traditional setting ($\epsilon = 8$, 10 steps) yields gains over the baseline, our proposed setting ($\epsilon \geq 64$, 1-2 steps) consistently achieves superior performance (e.g., CHAIR$_S$ 22.0 vs. 23.1). This confirms that a single, potent step with large magnitude provides a stronger contrastive signal than multiple small steps. Consequently, our large perturbation with small steps strategy achieves better hallucination suppression while maintaining significantly higher computational efficiency.

### F.4    COMPARISON WITH ALTERNATIVE ADVERSARIAL OBJECTIVES

While our method utilizes a vision-encoder-based loss ($L_{\cos}$) for computational efficiency, strictly output-level objectives (e.g., maximizing cross-entropy on LLM logits) are theoretically viable alternatives. However, they introduce severe scalability issues:

- **Computational Complexity ($O(N)$ vs $O(1)$):** Output-level attacks typically require backpropagating through the massive LLM for every token generation step ($O(N)$). In contrast, our method requires only a single, lightweight gradient pass through the vision encoder ($O(1)$).

- **Incompatibility with Caching:** Output-level attacks often necessitate updating the image dynamically, breaking standard KV-caching mechanisms. Our method generates a single static adversarial image, remaining fully compatible with efficient caching.

To empirically validate our choice, we compare AVCD against four alternative adversarial formulations on the CHAIR benchmark:

1. **Alternative 1 (Text-Image Align):** Maximizing the distance between image features and the generated caption embedding (Variant of MF-it Zhao et al. (2023a)).

2. **Alternative 2 (Hybrid):** Combining our original loss with Alternative 1.

3. **Alternative 3 (Black-box Latent):** Using Zeroth-Order optimization Chen et al. (2017) to push the image away from the text embedding (Variant of MF-tt Zhao et al. (2023a)).

4. **Alternative 4 (Logit Attack):** Direct maximization of Cross-Entropy loss on the LLM's output logits. (Restricted to the first token due to computational cost).

As shown in Table 7, all adversarial formulations outperform the baselines, validating the general effectiveness of the adversarial approach. However, AVCD achieves the best overall performance. We posit that since the target of our perturbation is the image space, optimizing the objective directly within the visual representation space (Vision Encoder) generates more robust gradients. Backpropagating errors through the heterogeneous modality gap (Text encoder and LLM), as done in Alternatives 1, 3, and 4, likely attenuates the gradient signal or introduces noise, making it less effective for manipulating visual features compared to our direct approach. Thus, our Vision Encoder Loss strikes the optimal balance between hallucination suppression and computational feasibility.

Table 7: Comparison of different adversarial objectives on CHAIR. Our vision encoder loss (AVCD) achieves the best performance while maintaining high efficiency.

| Method | Objective / Strategy | $C_S\downarrow$ | $C_I\downarrow$ |
|---|---|---|---|
| Baseline | Greedy | 26.9 | 8.9 |
| VCD | Diffusion Noise | 25.2 | 8.6 |
| Alternative 1 | Text-Image Align (MF-it variant) | 23.8 | 7.7 |
| Alternative 2 | Hybrid (Ours + MF-it) | 23.9 | 7.8 |
| Alternative 3 | Black-box Latent (MF-tt variant) | 23.9 | 8.1 |
| Alternative 4 | Logit Attack | 23.6 | 7.9 |
| AVCD | Vision Encoder Loss | **22.0** | **7.3** |

Table 8: Comparison with alternative contrast constructions. Generic dissimilarity (constant images or standard transformations) fails to achieve comparable performance to AVCD, and in some cases (e.g., Crop) performs worse than the baseline.

| Method | $C_S\downarrow$ | $C_I\downarrow$ | MME↑ | POPE-R↑ | POPE-P↑ | POPE-A↑ |
|---|---|---|---|---|---|---|
| Baseline | 26.9 | 8.9 | 1449.9 | 85.8 | 84.9 | 82.5 |
| AVCD | **22.0** | **7.3** | **1498.0** | **89.1** | **87.2** | **83.2** |
| Black Image | 25.2 | 8.4 | 1418.0 | 87.7 | 84.2 | 78.6 |
| White Image | 25.6 | 8.6 | 1409.5 | 87.2 | 83.2 | 77.8 |
| Average Image | 26.1 | 8.6 | 1389.7 | 87.2 | 83.4 | 77.9 |
| Blur | 25.0 | 8.6 | 1452.5 | 85.7 | 84.9 | 82.1 |
| Color Jitter | 27.0 | 9.1 | 1458.4 | 85.9 | 84.5 | 81.8 |
| Random Crop | 27.6 | 9.8 | 1422.9 | 84.6 | 84.0 | 81.1 |
| Random Flip | 25.6 | 8.9 | 1425.2 | 85.4 | 84.6 | 81.9 |
| Grayscale | 25.8 | 8.7 | 1463.1 | 85.3 | 84.5 | 82.0 |

## F.5    COMPARISON WITH ALTERNATIVE CONTRAST CONSTRUCTIONS

To verify whether the performance gains of AVCD stem specifically from adversarial directionality or simply from contrasting with generic dissimilarity, we expanded our evaluation to include a wide range of alternative contrastive baselines. We compare AVCD against two categories: (1) Constant Images (Black, White, Average) and (2) Standard Image Transformations (e.g., Blur, Color Jitter, Crop, Flip).

As summarized in Table 8, contrasting with generic constant images or standard transformations fails to achieve comparable performance to AVCD. In several cases, such as Random Crop (27.6) and Color Jitter (27.0), the hallucination rate ($C_S$) is even worse than the baseline (26.9). This confirms that generic dissimilarity merely introduces noise or irrelevant signals that do not effectively isolate hallucination modes. In contrast, AVCD consistently outperforms all alternative strategies, empirically confirming that specific adversarial directionality is crucial for providing a meaningful negative signal for contrastive decoding.

## F.6    EMPLOYING TARGETED ATTACK

To investigate whether a specific directional negative signal—steering the representation toward a distinct distractor—yields a more potent contrastive signal than unconstrained dissimilarity maximization, we implement a Push-Toward-Distractors strategy. We evaluate targeted adversarial attacks using three distinct selection criteria for the target embedding: the Nearest neighbor in the feature space, the Farthest image in the feature space, and a Random image selected from the dataset. The adversarial perturbation was optimized to minimize the feature distance to these respective targets.

As detailed in Table 9, the empirical results validate that targeted strategies significantly outperform the default baseline. Notably, steering toward a random target achieves a $CHAIR_S$ score of 21.8, slightly surpassing the standard untargeted AVCD. This suggests that attracting the embedding toward a mismatched neighbor can indeed create a highly effective contrastive signal. However, while targeted approaches prove effective, the proposed untargeted AVCD demonstrates superior stability and robustness across the full spectrum of benchmarks, preserving higher performance. This indicates that unconstrained maximization of dissimilarity captures a broader range of hallucination modes than pushing toward a specific fixed point.

From a deployment perspective, it is crucial to note that targeted methods generally introduce a dependency on an external reference dataset to retrieve neighbors or distractors, similar to retrieval-based methods like CICD (Zhao et al., 2025). In contrast, our standard AVCD approach remains entirely self-contained and data-free, operating solely on the single input image. Given that AVCD achieves top-tier performance without the logistical complexity of maintaining a reference corpus, we maintain it as the optimal default configuration for general-purpose hallucination mitigation.

Table 9: Comparison between Untargeted AVCD and Targeted strategies. While targeted attacks show promise, Untargeted AVCD demonstrates the best overall stability across diverse benchmarks without requiring an external reference dataset. Best results are bolded.

| Method | $C_S \downarrow$ | $C_I \downarrow$ | POPE-R $\uparrow$ | POPE-P $\uparrow$ | POPE-A $\uparrow$ | MME $\uparrow$ |
|---|---|---|---|---|---|---|
| Baseline | 26.9 | 8.9 | 85.8 | 84.9 | 82.5 | 1450 |
| AVCD (Untargeted) | 22.0 | **7.3** | **89.1** | 87.2 | **83.2** | **1498** |
| Targeted (Farthest) | 23.2 | 7.9 | **89.2** | 87.3 | **83.2** | 1481 |
| Targeted (Nearest) | 24.2 | 8.1 | 88.8 | **87.4** | 83.0 | 1446 |
| Targeted (Random) | **21.8** | 7.6 | 89.1 | **87.4** | 83.1 | 1476 |

# G EXTENDED EVALUATION AND GENERALIZABILITY

## G.1 ADDITIONAL RESULTS ON OTHER LVLM ARCHITECTURES

To further assess the generalizability of our proposed method, we conduct additional experiments on InstructBlip (Dai et al., 2023), MiniGPT4 (Zhu et al., 2023), and LLaVA-RLHF (Sun et al., 2024), which feature different architectural designs and training paradigms from the models used in our main experiments. The results are summarized in Table 10.

For InstructBlip, AVCD again demonstrates its effectiveness by achieving the best performance on the CHAIR and POPE benchmarks. While VCD shows a slightly higher score on the MME benchmark, AVCD still significantly outperforms the default greedy decoding. For MiniGPT4, AVCD consistently outperforms both baseline methods across all three benchmarks. Furthermore, the results on LLaVA-RLHF demonstrate that AVCD remains highly effective even for models aligned via Reinforcement Learning from Human Feedback (RLHF). As shown in the table, AVCD consistently surpasses both the default greedy decoding and VCD across all metrics, notably improving the MME score from 1199.8 to 1290.6. This indicates that our method is orthogonal to training-time alignment techniques and can further enhance the reliability of RLHF-aligned models. These additional findings suggest that the effectiveness of AVCD is not confined to a specific type of LVLM architecture. Instead, it serves as a robust and broadly applicable inference-time strategy for enhancing the reliability of various LVLMs.

Finally, we extend our evaluation to the state-of-the-art Qwen2.5-VL-7B (Bai et al., 2025) to assess AVCD's adaptability to advanced architectures employing dynamic resolution pipelines. Unlike standard fixed-resolution encoders, this model processes visual inputs at multiple scales, necessitating a tailored approach. We find that aligning the attack surface with task granularity is crucial: for detail-oriented tasks such as CHAIR and MME, we compute adversarial gradients through the dynamic processor to preserve high-fidelity features, whereas for coarse-grained verification in POPE, we target the raw resized input to induce a stronger global perturbation. As shwon in Table 11, despite the model's robust baseline, AVCD consistently yield superior performance, achieving a $CHAIR_S$ of 14.4 and an MME score of 1700. These results confirm that AVCD can be flexibly

Table 10: Performance comparison of AVCD against baselines on InstructBlip Dai et al. (2023), MiniGPT4 Zhu et al. (2023) and LLaVA-RLHF . Performance is reported with F1 score on the POPE Random subset, CHAIR$_S$, and the MME score.

| Model | Method | POPE ↑ | CHAIR ↓ | MME ↑ |
|---|---|---|---|---|
| InstructBlip | Greedy | 86.2 | 26.2 | 1354 |
| | VCD | 86.6 | 26.7 | **1375** |
| | AVCD | **88.5** | **25.0** | 1360 |
| MiniGPT4 | Greedy | 66.2 | 23.9 | 764.5 |
| | VCD | 61.5 | 22.7 | 719.1 |
| | AVCD | **69.8** | **20.7** | **775.5** |
| LLaVA-RLHF | Greedy | 77.8 | 22.3 | 1199.8 |
| | VCD | 76.9 | 23.6 | 1206.0 |
| | AVCD | **79.5** | **20.9** | **1290.6** |

adapted to sophisticated modern visual backbones, effectively suppressing hallucinations even in dynamic processing regimes.

Table 11: Performance comparison on Qwen2.5-VL-7B Bai et al. (2025) Despite the model's high baseline performance, AVCD achieves superior results across all benchmarks compared to default decoding and VCD. Best results are bolded.

| Method | $C_S$ ↓ | $C_I$ ↓ | POPE-R ↑ | POPE-P ↑ | POPE-A ↑ | MME ↑ |
|---|---|---|---|---|---|---|
| Baseline | 15.6 | 7.0 | 87.6 | 86.3 | 85.6 | 1686 |
| VCD | 16.6 | 7.6 | 87.5 | 86.0 | 85.5 | 1691 |
| AVCD | **14.4** | **6.6** | **90.5** | **89.1** | **87.4** | **1700** |

## G.2 ADDITIONAL COMPARISON WITH OTHER BASELINES

To situate AVCD within a broader landscape of hallucination mitigation techniques, we extend our evaluation to include five recent training-free steering and decoding interventions: DeGF (Zhang et al., 2025a), CICD (Zhao et al., 2025), OPERA (Huang et al., 2024), HALC (Chen et al., 2024b), and ProjectAway (Jiang et al., 2024a). All methods are evaluated under an identical experimental setting using LLaVA-1.5-7B with a $224 \times 224$ CLIP resolution across the CHAIR, POPE, and MME benchmarks. To rigorously assess computational efficiency and ensure consistency across all benchmarks, we conduct the evaluation using a fixed random seed, measuring the total inference time required to generate captions for the 500 samples in the CHAIR benchmark.

The results, summarized in Table 12, demonstrate that AVCD achieves the most stable and robust performance across diverse benchmarks. While CICD shows competitive performance on POPE, it suffers a significant performance drop on the complex tasks of MME (1314 compared to AVCD's 1498), indicating limited generalization. Similarly, while ProjectAway is computationally efficient, it is less effective in suppressing hallucinations, yielding a higher CHAIR$_S$ score of 24.6.

In addition to standard sampling-based evaluation, we conduct a supplementary comparison using greedy decoding to verify whether the performance gains are intrinsic to the method rather than artifacts of the decoding strategy. Compatible baselines were re-evaluated under greedy settings, excluding OPERA and HALC which rely on specific beam search mechanisms. As shown in the results, AVCD maintains superior performance in POPE and MME, outperforming all baselines. This consistency confirms that our method's robustness stems from intrinsic logit calibration. While CICD exhibits a marginal advantage in CHAIR under greedy decoding, it suffers a severe performance drop in MME, suggesting a trade-off with general perception capabilities. In contrast, AVCD effectively reduces hallucinations while preserving high-level visual reasoning. Furthermore, the effectiveness under greedy decoding highlights AVCD's unique capability to correct the Top-1 prediction, a property less prominent in standard noise-based interventions.

Finally, the analysis reveals a critical trade-off between hallucination suppression and computational cost. Although OPERA and HALC achieve lower hallucination rates on the CHAIR benchmark, they incur a prohibitive computational overhead. While their latency increase is moderate for short-answer tasks, it escalates drastically for sentence generation tasks like CHAIR. Specifically, inference becomes approximately $4\times$ slower with OPERA and nearly $12\times$ slower with HALC compared to AVCD. This suggests that the computational cost of these methods scales unfavorably with response length. In contrast, AVCD maintains a consistent and reasonable inference cost, offering a significantly more practical trade-off for real-world scenarios. Ultimately, AVCD is the only method that achieves top-tier results on both POPE and MME while remaining computationally efficient, confirming that our active adversarial steering approach provides a more generalized and scalable solution than heuristic decoding penalties or static feature steering.

Table 12: Experimental Results regarding extended comparative evaluation. Rows marked with (Greedy) denote performance using greedy decoding strategy.

| Method | $C_S\downarrow$ | $C_I\downarrow$ | POPE-R$\uparrow$ | POPE-P$\uparrow$ | POPE-A$\uparrow$ | MME$\uparrow$ | Time |
|---|---|---|---|---|---|---|---|
| AVCD | 22.1 | 7.3 | **89.2** | **87.0** | **83.0** | **1498** | 35m 8s |
| DeGF | 24.6 | 7.8 | 83.5 | 82.9 | 80.7 | 1391 | 1h 10m 26s |
| DeGF (Greedy) | 23.5 | 7.7 | 83.5 | 82.9 | 80.7 | 1373 | - |
| CICD | 23.4 | 7.4 | 88.3 | 85.9 | 81.4 | 1314 | 34m 4s |
| CICD (Greedy) | **21.8** | **7.2** | 88.3 | 86.3 | 81.6 | 1289 | - |
| OPERA | 19.2 | 6.1 | 84.6 | 83.6 | 81.4 | 1414 | 2h 11m 42s |
| HALC | **18.4** | **5.7** | 84.7 | 83.7 | 81.4 | 1426 | 7h 25m 47s |
| ProjectAway | 24.6 | 8.2 | 85.6 | 84.8 | 82.3 | 1444 | **24m 43s** |
| ProjectAway (Greedy) | 26.4 | 8.8 | 85.5 | 84.7 | 82.3 | 1450 | - |

### G.3 Impact of Input Resolution

We further investigate whether the efficacy of AVCD is sensitive to input resolution by conducting experiments with $336 \times 336$ resolution images on LLaVA-1.5. As presented in Table 13, AVCD maintains robust superiority over both greedy decoding and the VCD baseline across all benchmarks even with increased visual information. Notably, AVCD achieves the lowest hallucination rates on CHAIR ($C_S$ of 21.4 compared to 23.3 for greedy) and attains a remarkable score of 1536 on the MME benchmark. These results confirm that our method effectively scales to larger inputs, leveraging the richer visual details to generate potent contrastive signals without compromising the model's reasoning capabilities.

Table 13: Performance consistency with larger input images. Even with increased image resolution, AVCD maintains robust superiority over baselines across all benchmarks. Best results are bolded.

| Method | $C_S\downarrow$ | $C_I\downarrow$ | POPE-R$\uparrow$ | POPE-P$\uparrow$ | POPE-A$\uparrow$ | MME$\uparrow$ |
|---|---|---|---|---|---|---|
| Greedy | 23.3 | 7.6 | 89.7 | 86.8 | 81.7 | 1492 |
| VCD | 24.2 | 8.0 | 89.2 | 86.3 | 80.9 | 1496 |
| AVCD | **21.4** | **6.9** | **90.4** | **87.7** | **84.2** | **1536** |

### G.4 Evaluation on the MMHal Dataset

To further validate the robustness of AVCD on tasks requiring precise visual perception and complex reasoning, we conduct additional experiments on the MMHal-Bench (Sun et al., 2024). Unlike simple existence verification benchmarks, MMHal is specifically designed to evaluate detailed visual attributes (e.g., color, counting, spatial relationships) and includes adversarial questions intended to trigger hallucinations. As presented in Table 14, AVCD ($\epsilon = 64$) achieves an overall score of 2.14, significantly outperforming both the greedy decoding baseline (1.93) and VCD (2.01). This result offers two key insights: First, it addresses the concern that large adversarial perturbations

might inadvertently distort fine-grained visual cues; our findings indicate that the clean image branch effectively preserves essential details, while the adversarial branch serves to suppress language-driven hallucinations. Second, it demonstrates that the efficacy of AVCD extends beyond simple binary tasks to complex, open-ended generative tasks, confirming its broad generalizability.

Table 14: Performance comparison on MMHal-Bench Sun et al. (2024) on LLaVA. AVCD ($\epsilon = 64$) achieves the highest score, demonstrating efficacy in complex generative tasks. Results are averaged over five runs.

| Method | Greedy | VCD ($t = 500$) | VCD ($t = 999$) | AVCD ($\epsilon = 64$) | AVCD ($\epsilon = 256$) |
|---|---|---|---|---|---|
| Score | 1.93 | 2.01 | 2.01 | **2.14** | 2.07 |

## H  FURTHER ANALYSIS AND QUALITATIVE RESULTS

Table 15: Evaluation of cross-image adversarial transferability. Instance-specific optimization (AVCD) is essential for robust performance, whereas transferred perturbations often degrade performance on precision-critical tasks (e.g., MME).

| Method | $C_S\downarrow$ | $C_I\downarrow$ | MME↑ | POPE-R↑ | POPE-P↑ | POPE-A↑ |
|---|---|---|---|---|---|---|
| Baseline | 26.9 | 8.9 | 1449.9 | 85.8 | 84.9 | 82.5 |
| AVCD | **22.0** | **7.3** | **1498.0** | **89.1** | **87.2** | **83.2** |
| Random Image | 25.4 | 8.5 | 1463.2 | 88.1 | 86.7 | 83.1 |
| Nearest Neighbor | 24.3 | 8.2 | 1373.6 | 85.9 | 84.8 | 82.3 |
| Farthest Neighbor | 23.9 | 8.1 | 1369.2 | 86.0 | 85.2 | 82.7 |

### H.1  ANALYSIS OF CROSS-IMAGE ADVERSARIAL TRANSFERABILITY

To investigate whether the generated adversarial perturbations capture general feature sensitivities or are strictly instance-specific, we evaluate a cross-image transfer setting. We apply adversarial perturbations ($\delta = I_{adv} - I$) generated from a source image to a different target image. We test three source scenarios: (1) Random: $\delta$ from a randomly selected image; (2) Nearest: $\delta$ from the nearest neighbor in the vision encoder feature space (semantically similar); and (3) Farthest: $\delta$ from the farthest image.

As shown in Table 15, our instance-specific AVCD consistently outperforms all transfer baselines across all benchmarks. While transfer-based perturbations yield marginal improvements on CHAIR (e.g., Nearest 24.3 vs. Baseline 26.9) by acting as noise that disrupts language priors, they significantly degrade performance on precision-critical benchmarks like MME (dropping to 1370). This suggests that borrowed perturbations act as distractions that corrupt visual details without providing the correct steering signal. These results reinforce that per-image optimization is essential to simultaneously suppress hallucinations and maintain precise visual grounding.

### H.2  ANALYSIS OF CROSS-ARCHITECTURE ADVERSARIAL TRANSFERABILITY

To address the concern regarding black-box applicability, we conducted cross-architecture transfer experiments between LLaVA-1.5 and Qwen-VL. We evaluate the transferability by comparing the performance of transferred perturbations against the models' standard benchmarks provided in Table 16. This transfer performance outperforms the native LLaVA VCD baseline and surpass both the LLaVA Baseline. Similarly, the reverse direction demonstrated robust performance retention, which are comparable to or exceed the Qwen-VL VCD baseline. These findings confirm that our adversarial features capture semantic vulnerabilities that transfer across different architectures better than random noise.

Table 16: Evaluation of cross-architecture adversarial transferability. We compare the standard performance of LLaVA-1.5 and Qwen-VL (Baseline/VCD) against the performance when adversarial perturbations are transferred from a different source model (AVCD Transfer).

| Model | Method | $C_S\downarrow$ | $C_I\downarrow$ | POPE-R↑ | POPE-P↑ | POPE-A↑ | MME↑ |
|-------|--------|------|------|---------|---------|---------|------|
| LLaVA | Baseline | 26.9 | 8.9 | 85.8 | 84.9 | 82.5 | 1450.0 |
|       | VCD | 25.2 | 8.6 | 87.9 | **86.9** | 82.5 | 1448.9 |
|       | Transfer | **24.6** | **8.3** | **88.9** | **86.9** | **83.1** | **1462.4** |
| Qwen-VL | Baseline | 25.8 | **6.6** | 83.9 | 83.5 | 82.1 | 1253.8 |
|         | VCD | 26.2 | 7.7 | 87.8 | 86.7 | 84.0 | 1310.7 |
|         | Transfer | **24.5** | 7.3 | **88.0** | **86.8** | **84.1** | **1328.1** |

Table 17: Detailed breakdown of MME benchmark scores. We report the score for each sub-task and the total score.

| Method | Exist | Cnt | Pos | Clr | Post | Cel | Scn | Lnd | Art | OCR | Total |
|--------|-------|-----|-----|-----|------|-----|-----|-----|-----|-----|-------|
| LLaVA-1.5 | | | | | | | | | | | |
| Default | 185 | 148 | 135 | 160 | 128 | 120 | 158 | 162 | 117 | 138 | 1450 |
| VCD | 180 | 144 | 130 | 162 | 125 | 129 | 155 | 165 | 121 | 138 | 1449 |
| ICD | 185 | 137 | 140 | 165 | 124 | 116 | 156 | 162 | 115 | 148 | 1451 |
| M3ID | 185 | 138 | 130 | 160 | 122 | 113 | 158 | 163 | 115 | 133 | 1418 |
| AVISC | 180 | 148 | 135 | 160 | 128 | 121 | 158 | 164 | 118 | 138 | 1448 |
| DeGF | 195 | 148 | 112 | 153 | 118 | 107 | 153 | 151 | 115 | 140 | 1392 |
| CICD | 185 | 113 | 112 | 115 | 159 | 132 | 166 | 150 | 105 | 78 | 1315 |
| OPERA | 190 | 155 | 107 | 140 | 127 | 123 | 158 | 165 | 120 | 130 | 1414 |
| HALC | 190 | 153 | 117 | 140 | 125 | 122 | 159 | 164 | 120 | 138 | 1426 |
| ProjectAway | 185 | 143 | 147 | 160 | 128 | 122 | 159 | 161 | 115 | 125 | 1445 |
| DeGF (Greedy) | 190 | 147 | 118 | 145 | 109 | 102 | 155 | 151 | 118 | 138 | 1373 |
| CICD (Greedy) | 180 | 103 | 105 | 110 | 157 | 135 | 162 | 156 | 111 | 70 | 1289 |
| ProjectAway (Greedy) | 180 | 148 | 135 | 160 | 128 | 122 | 158 | 164 | 118 | 138 | 1450 |
| AVCD | 185 | 151 | 127 | 165 | 141 | 137 | 162 | 163 | 124 | 143 | 1498 |
| Qwen-VL | | | | | | | | | | | |
| Default | 170 | 145 | 98 | 180 | 164 | 86 | 155 | 96 | 110 | 50 | 1254 |
| VCD | 170 | 143 | 103 | 182 | 167 | 111 | 154 | 113 | 118 | 50 | 1311 |
| ICD | 161 | 143 | 100 | 167 | 155 | 85 | 150 | 89 | 108 | 50 | 1208 |
| M3ID | 165 | 150 | 98 | 180 | 168 | 88 | 156 | 99 | 110 | 50 | 1263 |
| AVISC | 170 | 150 | 98 | 180 | 164 | 87 | 158 | 99 | 110 | 50 | 1267 |
| AVCD | 171 | 150 | 113 | 182 | 174 | 104 | 155 | 118 | 123 | 50 | 1340 |

## H.3 DETAILED RESULTS OF MME

Table 17 provides a comprehensive breakdown of the MME benchmark performance. While some baselines may achieve marginally higher scores in specific sub-tasks depending on their optimization focus, AVCD demonstrates the most robust and balanced performance across the full spectrum of perception tasks. Unlike methods that exhibit high variance—excelling in one metric while degrading significantly in others—AVCD consistently maintains high performance across diverse categories without severe trade-offs. This consistency allows AVCD to achieve the highest total score among all evaluated methods. This validates that our approach effectively mitigates hallucinations while preserving and stabilizing the model's general perceptual capabilities.

## H.4 QUALITATIVE RESULTS

Figure 9 presents qualitative examples, using the LLaVA model for an image captioning task prompted with "Please describe this image in detail." The figure compares the output of standard greedy decoding on the original image against the output of our AVCD method, which contrasts the original image with its adversarially perturbed version.

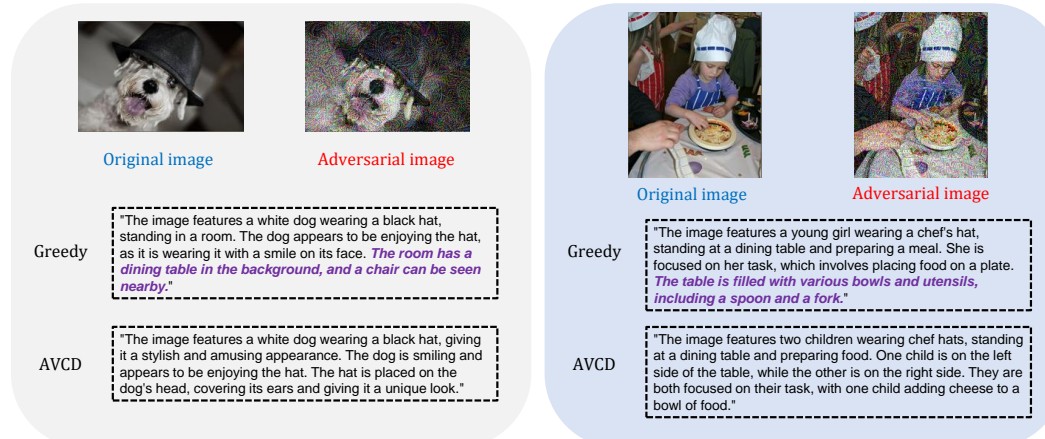

Figure 9: Qualitative comparison of AVCD and Greedy decoding. AVCD suppresses the adversarial-induced hallucinations seen in the Greedy outputs.

The top example shows a white dog wearing a hat. The standard greedy decoding of the original image hallucinates background objects, mentioning a "dining table in the background, and a chair" that are not clearly visible. AVCD, by using the adversarial image, generates a more focused and accurate description centered on the dog. In the bottom example, depicting a young girl with a chef's hat preparing food, the greedy baseline, when describing the original image, hallucinates an abundance of "various bowls and utensils, including a spoon and a fork" on the table. AVCD, leveraging the adversarial contrast, produces a more grounded caption describing "two children" and their actions with food, avoiding the unsubstantiated details. These examples highlight AVCD's ability to suppress the tendency to hallucinate extraneous details, leading to more reliable and factually consistent image descriptions.

### H.5 PROMPT DESIGN STRATEGY

To ensure a rigorous and fair evaluation on black-box models (GPT-4o and Gemini), we employ a carefully designed prompt strategy. First, to mitigate the evasive behavior of commercial APIs, which often decline to describe distorted images by claiming they are unclear or visually distorted, our prompts explicitly instruct the models to describe visual content regardless of image quality. Second, to prevent generating overly short or vague responses, we enforce a verbose output style (requiring at least 3–4 sentences). Crucially, these prompts are applied uniformly across all baselines and our method to ensure a fair comparison. The exact system prompts used for the two-stage "Transfer & Correct" pipeline are provided in Table 18 and Table 19.

## I LLM USAGE

During the manuscript preparation, we used Google's Gemini (https://gemini.google.com), a Large Language Model, to proofread our work. Our interaction with the LLM was iterative and focused exclusively on improving the quality of the writing. We affirm that the LLM served as an assistive tool and did not contribute to core research ideas, experimental design, analysis, and results presented in this paper. The final scientific content and all claims made in this paper are the sole responsibility of the authors.

1674
1675
1676
1677
1678
1679
1680
1681
1682
1683
1684
1685
1686
1687
1688
1689
1690
1691
1692
1693
1694
1695
1696
1697
1698
1699
1700
1701
1702
1703
1704
1705
1706
1707
1708
1709
1710
1711
1712
1713
1714
1715
1716
1717
1718
1719
1720
1721
1722
1723
1724
1725
1726
1727

Table 18: Stage 1 Prompt (Transfer): Used to generate the initial draft caption from the adversarial image (or noisy image for VCD). It enforces detailed description to expose hallucinations.

---

**System Prompt for Transfer**

You are a detailed image descriptive assistant. Your task is to generate a comprehensive and verbose caption describing the objects, entities, and scene within the provided image in great detail.

**[Style Guide Example]**
You are given a 'Style Guide Example' to understand the expected level of detail.
"The image features a red and green fire hydrant sitting in a garden near a sidewalk... (omitted for brevity)"

**[Your Task]**
Carefully examine the image provided. Your primary goal is to describe the objects and scene visually as much as possible to fill a detailed paragraph.

1. **Be Verbose & Detailed:** Do not write a short sentence. Write at least 3-4 full sentences. Describe the visual attributes (colors, shapes, textures), spatial relationships, and actions in detail.

2. **Prioritize Objects:** Identify the main objects using the [Style Guide Example] as a guide.

3. **Describe What You See:** Even if objects look abstract or stylized, describe their visual appearance thoroughly.

4. **DO NOT Analyze Quality:** Do not use words like "blurry," "distorted," or "unclear." Just describe the visual content.

5. **DO NOT Refuse:** Describe whatever visual patterns or objects are visible.

**[Good Example (Target Length & Style)]**
"The image depicts a cozy bedroom scene centered around a black metal bed frame... (omitted for brevity)"

**[Bad Example (Too Short)]**
"A bedroom with a bed and a laptop."

**[Bad Example (Refusal)]**
"The image appears to be visually distorted. Therefore, I cannot generate a caption."

Generate your final, detailed caption.

---

Table 19: Stage 2 Prompt (Correct): Used to refine the caption using the original image, explicitly using the draft from Stage 1 as a negative constraint.

---

**System Prompt for Correct**

You are an expert image verifier and captioner.  Your task is to
correct a potentially flawed draft caption based on the *actual*
image provided.
The draft caption was generated from a distorted version of the
image and may contain hallucinations or factual errors (e.g.,
objects that are not present).
Your goal is to produce a single, comprehensive, verbose, and
factually accurate description.

**Instructions:**

1. Carefully examine the provided image.  This is your only
   source of truth.

2. Read the draft caption.

3. Aggressively remove any statements, objects, or details from
   the draft that you cannot 100% visually confirm in the image.

4. Generate a new, final description based only on the visually
   confirmed facts.

5. Be Verbose & Detailed:  The final description must be rich
   and comprehensive.  Use the confirmed facts to write at
   least 3 full sentences.  Elaborate on the colors, materials,
   textures, and spatial positions of all confirmed objects.  Do
   not be concise.

---

**[Image Draft]**
Draft (from Distorted Image):
"{resp_adv}"

---

**[Your Task]**
Based only on the image provided, generate the final, detailed,
verbose, and factually accurate description.

