# OpenReview forum: "Adversarial Visual Contrastive Decoding for Mitigating Hallucinations in Large Vision-Language Models"
_ICLR.cc/2026/Conference — Submitted to ICLR 2026_

### Official Review · Reviewer_mm7w · 2025-10-23

**Soundness:** 3
**Presentation:** 3
**Contribution:** 2
**Rating:** 4
**Confidence:** 4

**Summary:**

The paper proposes Adversarial Visual Contrastive Decoding (AVCD), an inference-time method for reducing hallucinations in large vision-language models. Instead of contrasting the model’s logits on the original image against Gaussian-noised variants (as in prior VCD-style methods), AVCD generates adversarial images via a few PGD steps that intentionally push the visual features away from the ground-truth semantics. Decoding then subtracts the likelihood under this adversarial view from the original, sharpening the decision signal and suppressing spurious content. The authors support this with analyses of entropy, CLIP-space similarity shifts, and attention variance, and report performance gains over contrastive baselines on POPE, CHAIR, and MME.

**Strengths:**

(+) AVCD proposes simple yet effective methods into the inference pipeline, requires only a few adversarial steps, and delivers steady improvements over Gaussian-based contrastive decoding across multiple LVLMs and benchmarks.

(+) The paper clearly articulates why adversarial (directional) perturbations are superior to Gaussian (isotropic) noise, supporting the claim with both theoretical analysis and empirical evidence.

(+) Results show stable and repeatable improvements over strong baselines, with minimal implementation complexity and modest computational overhead.

**Weaknesses:**

(-) Novelty is incremental. Replacing Gaussian perturbations with adversarial “hard negatives” is a natural extension of contrastive decoding. The paper’s analysis is solid, but the conceptual step feels unsurprising and incremental relative to prior visual distortion and contrastive techniques.

(-) White-box assumption limits applicability. The method requires backprop access through the vision encoder to generate an adversarial image. This excludes many closed-source/API LVLMs and any deployment where gradients are not exposed, curbing practical impact.

(-) The comparison scope is narrow. Evaluations emphasize contrastive decoding baselines. Additional comparisons with [1,2] are needed. Furthermore, the paper does not adequately situate AVCD against training-time or steering-time approaches—e.g., RLHF, OPERA, HALC, Woodpecker, ProjectAway.

[1] Self-Correcting Decoding with Generative Feedback for Mitigating Hallucinations in Large Vision-Language Models, ICLR 2025.
[2] Cross-Image Contrastive Decoding: Precise, Lossless Suppression of Language Priors in Large Vision-Language Models, arxiv 2025.

(-)
The LVLMs evaluated are outdated. Including newer/stronger backbones would strengthen claims of generality and headroom.

(-) Performance depends noticeably on ε, α, and step count, and the best settings vary by task. Providing an adaptive schedule (e.g., entropy- or uncertainty-driven α/ε) or a simple tuning heuristic would improve robustness and deployability.

**Questions:**

AVCD maximizes feature distance from the current image semantics. What if the adversarial objective instead pushes the adversarial embedding toward embeddings of other random images (or toward a curated bank of semantically conflicting images)? Intuitively, this could provide a more directional negative signal than an unconstrained push-away. Have you tried a “push-toward-distractors” loss (e.g., nearest-neighbor attraction to a mismatched image set in the same feature space)? How does it compare in efficacy and cost?

---

> ### Author Response · Authors · 2025-11-21
>
> **Regarding Contribution and Novelty (W1)**
>
> We acknowledge that AVCD can be viewed as a natural evolution of contrastive decoding. However, we respectfully submit that our primary contribution lies not in the algorithmic complexity, but in the rigorous theoretical and empirical characterization of why this evolution is necessary.
>
> While previous methods relied on the heuristic of passive degradation, our work establishes a new paradigm of "Active Directional Steering." We provide:
> - Theoretical Grounding: A mathematical proof (Theorem 1) demonstrating that adversarial perturbations uniquely invert semantic similarity, unlike random noise.
> - Mechanistic Insight: Deep analysis at both the Encoder and LLM levels (e.g., VAR analysis) revealing that adversarial signals actively forge false grounding rather than merely breaking it.
>
> We believe that elevating a heuristic improvement to a theoretically grounded methodology, combined with the virtue of simplicity for deployment, constitutes a meaningful contribution to the field.
>
> **Regarding White-Box Assumption (W2)**
>
> We appreciate the reviewer’s emphasis on the practical applicability of our method. While the standard AVCD formulation indeed requires gradient access, we demonstrate that its practical impact extends to black-box scenarios through the property of adversarial transferability. Unlike internal intervention methods that are strictly bound to white-box settings, AVCD generates input-level artifacts (adversarial images) that can be optimized on open-source surrogates and effectively transferred to target APIs.
>
> To address this shared concern comprehensively, we have detailed our black-box experiments (on GPT-4o and Gemini) in the "**Common Response: Applicability to Black-Box Models**" section above. We kindly refer the reviewer to that section for empirical evidence of AVCD's deployment potential in closed-source settings.
>
> **Regarding Comparison Scope (W3)**
>
> We thank the reviewer for the valuable suggestion to broaden the comparison scope. We agree that situating AVCD against a wider range of methodologies—including training-free steering and decoding interventions—provides a more holistic view of its efficacy.
>
> **Extended Comparative Evaluation:** We expanded our experiments to include five additional state-of-the-art baselines: DeGF [1], CICD [2], OPERA [3], HALC [4], and ProjectAway [5]. (We excluded Woodpecker [6] from comparison as it relies on external off-the-shelf detection models, making fair latency measurement ambiguous.) All methods were evaluated under the identical setting (LLaVA-1.5-7B with 224x224 CLIP) on CHAIR, POPE, and MME benchmarks. We also measured the total inference time for the CHAIR benchmark (500 samples) to assess efficiency. We note that all the experiments in the response conducted using a fixed random seed.
>
> **Experimental Results**
> | Method | CHAIRS↓ | CHAIRI↓ | POPE-R ↑ | POPE-P↑ | POPE-A↑ | MME↑ | Time |
> |:-------|:-------:|:-------:|:------------:|:------------:|:------------:|:------:|:-----:|
> | AVCD | 22.1 | 7.3 | **89.2** | **87.0** | **83.0** | **1498** | 35m 8s |
> | DeGF | 24.6 | 7.8 | 83.5 | 82.9 | 80.7 | 1391 | 1h 10m 26s |
> | CICD | 23.4 | 7.4 | 88.3 | 85.9 | 81.4 | 1314 | 34m 4s |
> | OPERA | 19.2 | 6.1 | 84.6 | 83.6 | 81.4 | 1414 | 2h 11m 42s |
> | HALC | **18.4** | **5.7** | 84.7 | 83.7 | 81.4 | 1426 | 7h 25m 47s |
> | ProjectAway | 24.6 | 8.2 | 85.6 | 84.8 | 82.34 | 1444 | **24m 43s** |
>
> **Analysis: Performance vs. Efficiency Trade-off**
>
> - **Stability across Benchmarks:** AVCD demonstrates the most stable and robust performance across all tasks. While CICD performs well on POPE, it suffers a significant drop on the complex reasoning tasks of MME (1314 vs. 1498). Similarly, ProjectAway is efficient but less effective in suppressing hallucinations (CHAIR$_S$ 24.6).
> - **Efficiency Advantage:** Although OPERA and HALC achieve lower hallucination rates on CHAIR, they incur a prohibitive computational cost. While their overhead was moderate on short-answer tasks like POPE and MME (approx. 2-4$\times$ slower), it escalated drastically on the sentence generation task (CHAIR), resulting in ~4$\times$ (OPERA) to ~12$\times$ (HALC) slower inference compared to AVCD. This indicates that their computational cost scales unfavorably with response length. In contrast, AVCD maintains a consistent and reasonable inference cost, offering a much more practical trade-off for real-world generation scenarios.
> - **Holistic Superiority:** AVCD is the only method that achieves top-tier results on POPE and MME while remaining computationally efficient. This confirms that our approach of active adversarial steering provides a more generalized solution than heuristic decoding penalties or static feature steering.
>
> (continued)

---

> ### Author Response · Authors · 2025-11-21
>
> **Regarding Comparison Scope (W3 continued)**
>
> **Compatibility with Training-Time Approaches (RLHF):** We further situate AVCD against training-time approaches like LLaVA-RLHF [7].
>
> | Method | CHAIRS↓ | CHAIRI↓ | POPE-R↑ | POPE-P↑ | POPE-A↑ | MME↑ |
> |:-------|:-------:|:-------:|:------------:|:------------:|:------------:|:------:|
> | Default | 22.3 | 7.5 | 85.8 | 80.4 | 77.8 | 1199 |
> | VCD | 23.6 | 7.7 | 85.0 | 79.9 | 76.9 | 1206 |
> | **AVCD** | **20.9** | **7.1** | **86.7** | **82.3** | **79.5** | **1290** |
>
> - **Insight from Recent Literature:** Recent studies [8] suggest that while reasoning-oriented models (including RLHF-aligned ones) excel in complex generation, they often exhibit a trade-off, showing degradation in pure visual perception tasks. Our results on LLaVA-RLHF align with this finding: while it achieves a low hallucination rate on the generative CHAIR task, its performance on perception-heavy benchmarks like POPE and MME is relatively lower than our inference-time optimized standard model.
> - **Orthogonality and Perception Boosting:** We applied AVCD on top of LLaVA-RLHF to test if it can bridge this gap.
>
> As shown, AVCD is orthogonal to RLHF. It not only further reduces the hallucination rate on CHAIR but, more importantly, significantly recovers the perception performance on POPE and MME. This demonstrates that AVCD can serve as a crucial inference-time perception booster, complementing the generative capabilities of RLHF-aligned models.
>
> [1] Zhang, Ce, et al. "Self-correcting decoding with generative feedback for mitigating hallucinations in large vision-language models." arXiv preprint arXiv:2502.06130 (2025).\
> [2] Zhao, Jianfei, et al. "Cross-image contrastive decoding: Precise, lossless suppression of language priors in large vision-language models." arXiv preprint arXiv:2505.10634 (2025).\
> [3] Huang, Qidong, et al. "Opera: Alleviating hallucination in multi-modal large language models via over-trust penalty and retrospection-allocation." Proceedings of the IEEE/CVF Conference on Computer Vision and Pattern Recognition. 2024.\
> [4] Chen, Zhaorun, et al. "Halc: Object hallucination reduction via adaptive focal-contrast decoding." arXiv preprint arXiv:2403.00425 (2024).\
> [5] Jiang, Nick, et al. "Interpreting and editing vision-language representations to mitigate hallucinations." arXiv preprint arXiv:2410.02762 (2024).\
> [6] Yin, Shukang, et al. "Woodpecker: Hallucination correction for multimodal large language models." Science China Information Sciences 67.12 (2024): 220105.\
> [7] Sun, Zhiqing, et al. "Aligning large multimodal models with factually augmented rlhf." Findings of the Association for Computational Linguistics: ACL 2024. 2024.\
> [8] Liu, Chengzhi, et al. "More Thinking, Less Seeing? Assessing Amplified Hallucination in Multimodal Reasoning Models." arXiv preprint arXiv:2505.21523 (2025).
>
>
> **Regarding Evaluated LVLMs (W4)**
>
> We thank the reviewer for the suggestion. To demonstrate the generality and headroom of our method on state-of-the-art architectures, we evaluated AVCD on Qwen2.5-VL-7B [1], one of the strong open-source LVLMs available.
>
> **Experimental Results** Despite the model's high baseline performance, AVCD consistently achieves superior results across all benchmarks.
>
> | Method | CHAIRS↓ | CHAIRI↓ | POPE-R↑ | POPE-P↑ | POPE-A↑ | MME↑ |
> |:-------|:-------:|:-------:|:------------:|:------------:|:------------:|:----:|
> | Default | 15.6 | 7.0 | 87.6 | 86.3 | 85.6 | 1686 |
> | VCD | 16.6 | 7.6 | 87.5 | 86.0 | 85.5 | 1691 |
> | AVCD | **14.4** | **6.6** | **90.5** | **89.1** | **87.4** | **1700** |
>
> Unlike existing architectures, Qwen2.5-VL employs a sophisticated multi-scale visual processing pipeline with dynamic resolution. We adapted our attack strategy to align with this architectural difference:
> - **Detail-Oriented Tasks (CHAIR, MME):** We computed the adversarial gradients through the dynamic processor. This allows the perturbation to be optimized on the high-fidelity features, precisely suppressing hallucinations without destroying detailed visual cues needed for complex reasoning.
> - **Coarse-grained Verification (POPE):** We computed the adversarial gradients directly on the raw resized input (before the dynamic processor). In this spatially condensed view, the perturbation delivers a stronger global shock ($\epsilon=256$), which is necessary to flip the model's high-confidence existence priors.
>
> These results reveal an important insight: the optimal attack surface depends on the alignment between task granularity and the visual processing stage. While this requires architecture-specific tuning, it showcases AVCD's flexibility in adapting to modern, complex visual processing pipelines beyond simple vision encoders, achieving significant gains across diverse benchmarks.
>
> [1] Bai, Shuai, et al. "Qwen2. 5-vl technical report." arXiv preprint arXiv:2502.13923 (2025).

---

> ### Author Response · Authors · 2025-11-21
>
> **Regarding Hyperparameter Sensitivity (W5)**
>
> We thank the reviewer for the constructive suggestion. We agree that robustness is key for deployment.
>
> **Practical Heuristic: Complexity determines $\epsilon$:** Instead of dynamic tuning, we propose a robust default strategy. We explicitly distinguish tasks not just by format, but by Complexity:
> - **Simple Object Verification (POPE):** Requires overcoming strong object priors. High perturbation ($\epsilon=256$) is needed to flip the high-confidence hallucination.
> - **Complex tasks & Generation (CHAIR, MME):** Although MME is binary, it requires identifying fine-grained attributes and relations, similar to captioning. High perturbations distort these fine-grained cues. Therefore, moderate perturbation ($\epsilon=64$) is optimal to preserve visual details while suppressing hallucinations.
>
> **Infeasibility of Adaptive Scheduling:** Regarding the suggestion for an entropy-driven schedule, we identified critical bottlenecks:
> - **Computational Cost ($O(1)$ vs. $O(N)$):** Adaptive methods require updating the image based on token entropy during generation. This creates a recurrent cost proportional to response length ($O(N)$), whereas our method is a one-time operation ($O(1)$).
> - **Incompatibility with KV Caching:** Standard efficient inference relies on KV caching, which assumes a static image context. Dynamically changing the adversarial image per token would invalidate the cache, forcing a full re-computation at every step.
> - **Hyperparameter Complexity:** Existing adaptive methods (e.g., [1]) often require introducing additional hyperparameters (e.g., multiple thresholds) to tune the schedule itself. This contradicts the goal of simplicity.
>
> Therefore, we recommend the simple heuristic: Use $\epsilon=64$ as the universal default for most reasoning/generation tasks, and reserve $\epsilon=256$ only for simple existence verification. This offers the best trade-off between performance and deployability.
>
> [1] Favero, Alessandro, et al. "Multi-modal hallucination control by visual information grounding." Proceedings of the IEEE/CVF Conference on Computer Vision and Pattern Recognition. 2024.
>
>
> **Regarding "Push-Toward-Distractors" Strategy (Q1)**
>
> We thank the reviewer for this excellent suggestion. The hypothesis that a directional negative signal (pushing toward a specific distractor) could be more effective than an unconstrained push is highly insightful.
>
> To rigorously test this, we implemented the "Push-Toward-Distractors" strategy using three target selection criteria: Nearest, Farthest, and Random images from the dataset. We optimized the adversarial perturbation to minimize the distance to these target embeddings.
> - Random: a randomly selected image.
> - Nearest: the nearest neighbor in the vision encoder feature space.
> - Farthest: the farthest image in the feature space.
>
> **Experimental Results**
> | Method | CHAIRS↓ | CHAIRI↓ | POPE-R ↑ | POPE-P ↑ | POPE-A ↑ | MME↑ |
> |:-------|:-------:|:-------:|:------------:|:------------:|:------------:|:------:|
> | Default | 26.9 | 8.9 | 85.8 | 84.9 | 82.5 | 1450 |
> | AVCD | 22.1 | **7.3** | 89.2 | 87.2 | **83.2** | **1498** |
> | Farthest | 23.2 | 7.9 | **89.2** | 87.3 | **83.2** | 1481 |
> | Nearest | 24.2 | 8.1 | 88.8 | **87.4** | 83.0 | 1446 |
> | Random | **21.8** | 7.6 | 89.1 | **87.4** | 83.1 | 1476 |
>
> **Analysis**
> - **Efficacy of Targeted Attacks:** The results validate the reviewer's intuition. Targeted strategies perform significantly better than the baseline. Notably, the Random target strategy achieves a CHAIR$_S$ score that slightly improves upon AVCD, proving that attracting the embedding to a mismatched neighbor creates a potent contrastive signal.
> - **Stability of AVCD:** While Targeted approaches are also effective, AVCD (Untargeted) remains the most robust across all benchmarks, particularly on CHAIR and MME tasks. This suggests that the unconstrained maximization of dissimilarity captures a broader range of hallucination modes than pushing toward a specific point.
>
> **Deployment Consideration: Independence vs. Dependency**
> - **Targeted (Nearest/Farthest):** While retrieval can be pre-computed (as in CICD [1]) to minimize runtime latency, this approach introduces a dependency on an reference dataset.
> - **Untargeted:** Our method is self-contained and data-free, operating solely on the single input image.
>
> The reviewer's suggested strategy is a viable and high-performing alternative, sometimes rivaling untargeted AVCD. However, given that the original AVCD achieves top-tier performance across diverse metrics without requiring an external reference dataset, we maintain it as the proposed default for its universality and deployment simplicity. We have added these valuable findings to the paper.
>
> [1] Zhao, Jianfei, et al. "Cross-image contrastive decoding: Precise, lossless suppression of language priors in large vision-language models." arXiv preprint arXiv:2505.10634 (2025).

---

> ### Comment · Reviewer_mm7w · 2025-11-24
>
> Thank the authors for their detailed rebuttal and the extensive additional experiments. These efforts have certainly strengthened the empirical value of the paper. However, after carefully reviewing the rebuttal, some of my concerns remain unresolved.
>
> 1. Novelty remains incremental
>
> While the theoretical analysis provides a formal basis for the method, it appears to serve more as a post-hoc justification rather than a derivation of a novel framework. Intuitively, using an adversarial image—which is explicitly optimized to degrade semantic features—is expected to be a more effective negative sample than undirected random Gaussian noise. The theorem confirms why this intuition holds (semantic inversion), but it does not fundamentally change the incremental nature of the methodological contribution compared to existing contrastive decoding approaches.
>
> 2. Issues with Black-Box Verification (Transfer & Correct)
>
> I am not convinced by the "Transfer & Correct" experiment on GPT-4o-mini and Gemini as a proof of black-box applicability.
> While this pipeline demonstrates the extendability of AVCD, it does not sufficiently address the practical concerns regarding black-box deployment. Furthermore, the proposed pipeline utilizes a two-turn inference process. A proper baseline for this setup would be other multi-turn inference strategies, such as Chain-of-Thought or Self-Refinement, to isolate the gain from AVCD versus simply having more compute/steps. The requirement for a surrogate model followed by a two-stage inference incurs significant latency and cost, which may not be justifiable compared to simpler prompting strategies.
>
> To genuinely validate the transferability claim for black-box scenarios, the authors should demonstrate cross-architecture transferability. Specifically, it would be compelling to demonstrate whether an adversarial perturbation generated on a specific vision encoder successfully functions as a contrastive negative signal when decoding with an LVLM that utilizes a completely different vision encoder.
>
> 3. Complexity in Deployment
>
> While the results on Qwen2.5-VL demonstrate superior performance, they simultaneously reveal a practical weakness regarding deployment complexity. The fact that the method requires different attack strategies based on the task indicates a lack of universality. This introduces significant engineering overhead and reduces the flexibility of AVCD compared to simpler noise-based methods.
>
> 4. (minor) Detailed performance for MME
>
> Regarding the MME benchmark, the aggregate score can obscure underlying trade-offs. I request a detailed performance breakdown per category within the MME benchmark, specifically for the hallucination subset (object-level: existence, count; attribute-level: position, color).

---

> > ### Author Response · Authors · 2025-11-24
> >
> > We appreciate your detailed feedback. We would like to address the remaining concerns by clarifying the fundamental scope of our contributions and the rationale behind our experimental design.
> >
> > **1. On Novelty: Scientific Formalization vs. Intuition** We respectfully submit that there is a fundamental scientific distinction between "having an intuition" and "establishing a theoretical formalization.
> > - **Beyond Intuition:** While the utility of adversarial perturbations may seem intuitive post-hoc, intuition alone does not fully explain the conditions required for success. Intuition typically relies on standard adversarial assumptions, which prioritize imperceptibility (small $\epsilon$). However, our work reveals that this standard intuition can be insufficient. While small perturbations yield marginal gains, they fail to create the "Semantic Inversion" required for a strong negative signal.
> > - **Theoretical Contribution:** Our contribution lies in proving the necessary condition (Theorem 1: large $\epsilon$) to achieve this inversion. By characterizing why large perturbations are essential to actively steer the model—deviating from the traditional "imperceptible" attack paradigm—we transform a heuristic into a grounded methodology with distinct boundary conditions.
> >
> > **2. On Black-Box Applicability (Feasibility & Baselines)** We wish to remind the reviewer that this experiment was an exploratory extension conducted to address the lack of established protocols for applying decoding interventions to closed-source APIs.
> > - **Feasibility of Black-Box Applicability:** Most baselines (ProjectAway, OPERA, HALC, DeGF) rely on accessing internal states (logits, attention, gradients) at every step, making them structurally impossible to deploy in black-box APIs. CICD is inapplicable as its distinct image strategy generates a semantically irrelevant draft with no value as a negative constraint. Similarly, VCD (random noise) lacks the targeted semantic steering required for effective correction. Consequently, no precedent existed for such interventions. We designed the "Transfer & Correct" pipeline as a novel protocol to bridge this gap, leveraging the unique transferability of adversarial examples where no prior reference was available.
> > - **Isolating the Source of Gain:** Regarding the concern that gains might stem simply from multi-turn processing, we argue that the source of improvement is distinct. CoT-style strategies rely on re-reasoning over the same visual input. Recent studies demonstrate that if the model hallucinates due to visual misperception, reiterating on the same input often amplifies the error due to reduced visual attention [1]. In contrast, AVCD introduces a distinct adversarial visual input ($I_{adv}$). The performance gain is driven by this "Hard Negative" contrast—explicitly showing the model what not to follow—which provides a corrective visual signal that simple self-reflection cannot generate.
> >
> > **3. On Transferability (Framework vs. Attack Method)** We respectfully argue that requesting transfer experiments between open-source models misses the contributions and practical context of our work.
> > - **Scope of Contribution:** Our work proposes a hallucination mitigation framework, not a new adversarial attack method. Our goal was to demonstrate that adversarial principles (PGD) can be repurposed to mitigate hallucinations. Optimizing attack transferability itself is orthogonal to our primary contribution of decoding-based mitigation.
> > - **Practical Relevance:** For open-source models, applying White-box AVCD directly is always the optimal choice. There is no practical scenario where treating an open-source model as a black box is necessary.
> > - **Empirical Validation (LLaVA $\leftrightarrow$ Qwen):** To address this, we conducted the requested cross-architecture experiments on the CHAIR benchmark. While LLaVA-1.5 and Qwen-VL both utilize ViT-based backbones, they differ in scale (Large vs. Giant) and pre-training weights (OpenAI vs. OpenCLIP).
> >   - **Qwen $\to$ LLaVA:** The transferred perturbation achieved $24.6$ / $8.3$, outperforming the VCD ($25.2$ / $8.6$).
> >   - **LLaVA $\to$ Qwen:** Similarly, it improved performance to $24.5$ / $7.3$. outperforming the VCD ($26.2$ / $7.7$).
> >   - This confirms that even without sophisticated transfer techniques, our adversarial features capture robust semantic vulnerabilities that transfer across different architectures better than random noise.
> > - **Evidence of Robustness:** Consequently, combined with our successful transfer to GPT-4o and Gemini (closed-source APIs), we believe this provides compelling evidence that our framework is robust and applicable across disjoint, state-of-the-art architectures.
> >
> > [1] "More Thinking, Less Seeing? Assessing Amplified Hallucination in Multimodal Reasoning Models." arXiv preprint arXiv:2505.21523 (2025).

---

> > ### Author Response · Authors · 2025-11-24
> >
> > **4. On Deployment Complexity (Robustness & Headroom)** Regarding Qwen2.5-VL, we wish to clarify that we shared the tuned results to highlight the significant performance headroom we discovered, rather than to imply that tuning is mandatory.
> > - **Universal Default Works:** We verified that our universal default setting consistently outperforms the baseline on POPE (Random): Baseline 87.6 $\to$ AVCD 87.9, POPE (Popular): Baseline 86.3 $\to$ AVCD 86.8, POPE (Adversarial): Baseline 85.6 $\to$ AVCD 86.0.
> > - The method is inherently robust with a single configuration. The tuned setting was reported to demonstrate the maximum potential of our approach for the community's interest.
> >
> > **5. Detailed Performance for MME** Per your request, we provide the detailed breakdown for both LLaVA-1.5 and Qwen-VL. The results demonstrate that AVCD provides robust improvements across hallucination metrics and complex reasoning tasks.
> >
> > | Model | Method | Existence | Count | Position | Color | Posters | Celebrity | Scene | Landmark | Artwork | OCR | Total |
> > |-------|--------|-----------|-------|----------|-------|---------|-----------|-------|----------|---------|------|--------|
> > | LLaVA | default | 185.00 | 148.33 | 135.00 | 160.00 | 127.55 | 120.00 | 157.75 | 162.25 | 116.50 | 137.50 | 1449.88 |
> > | | VCD | 180.33 | 144.33 | 130.00 | 162.00 | 124.83 | 129.00 | 155.15 | 164.70 | 121.05 | 137.50 | 1448.90 |
> > | | AVCD | 185.00 | 150.67 | 127.00 | 165.00 | 140.95 | 137.41 | 162.15 | 162.55 | 124.30 | 143.00 | 1498.03 |
> > | Qwen-VL | default | 170.00 | 145.00 | 98.33 | 180.00 | 164.29 | 85.88 | 155.00 | 95.50 | 109.75 | 50.00 | 1253.75 |
> > | | VCD | 170.00 | 143.00 | 103.33 | 182.00 | 166.39 | 110.82 | 154.15 | 113.00 | 118.00 | 50.00 | 1310.70 |
> > | | AVCD | 171.00 | 150.00 | 113.33 | 182.00 | 173.81 | 103.76 | 155.15 | 118.25 | 122.95 | 50.00 | 1340.26 |
> >
> > - **Consistency:** AVCD consistently improves Count and Color across both architectures.
> > - **Position Robustness:** While LLaVA shows a slight trade-off in Position, Qwen-VL achieves a significant gain (+15.0) in this metric. This confirms that AVCD does not inherently degrade spatial features; rather, the effect is architecture-dependent, and in models like Qwen-VL, it significantly corrects spatial hallucinations.
> > - **Complex Grounding:** Both models show substantial gains in complex recognition tasks (e.g., Posters, Celebrity), driving the Total Score up significantly (+48.1 for LLaVA, +86.5 for Qwen-VL). This validates that AVCD enhances overall visual grounding without compromising fine-grained perception.
> >
> > We hope this clarifies that AVCD offers a theoretically grounded framework that is distinct from existing baselines and robust in practical deployment.

---

> ### Comment · Reviewer_mm7w · 2025-11-27
>
> I appreciate the authors' comprehensive response and the significant effort put into the additional experiments. While I understand that this paper does not primarily target adversarial attacks, the proposed method—unlike standard contrastive decoding approaches—relies on the premise of white-box access to the vision encoder for backpropagation. Consequently, establishing cross-architecture transferability is crucial to demonstrate flexibility in "gray-box" scenarios where direct access to the vision encoder is unavailable.
>
> While some of my concerns are addressed, a closer inspection of the rebuttal raises several new  questions that prevent me from fully endorsing the paper in its current state.
>
> 1. Novelty
>
> The authors frame the necessity of "Large $\epsilon$" as a counter-intuitive discovery by contrasting it with standard adversarial attacks, which typically prioritize imperceptibility. However, in the specific context of contrastive decoding, this appears to be a straw man argument. The direct competitor, VCD, has already demonstrated that increasing noise steps results in a loss of visual information (Fig.2 of VCD paper) and utilizes massive distortions (e.g., noise steps 500-999) that render images unrecognizable. Therefore, the fact that AVCD also requires a large perturbation budget to invert semantics does not represent a "deviation from the paradigm," but rather an expected alignment with the existing contrastive decoding framework.
>
>
> 2. Default Decoding Strategy
>
> It is not explicitly clarified in the manuscript or the rebuttal whether the reported reproductions of AVCD and the comparisons were conducted using Greedy Decoding by default. However, upon reviewing the rebuttal data, the reported baseline performance
> aligns with the greedy decoding results from the main paper, suggesting that greedy decoding was indeed employed.
> Is there a specific reason for utilizing greedy decoding as the default setting for these comparisons?
> Given that contrastive decoding methods fundamentally operate by manipulating the logit distribution to suppress hallucination-prone tokens, utilizing sampling-based decoding is the standard practice in this field. This is because sampling exposes the model to a broader range of the probability distribution, thereby rigorously testing whether the modified logits effectively mitigate hallucinations under diverse generation scenarios. Validating the method only under the deterministic path of greedy decoding may overestimate its robustness.
>
>
> 3. Analysis of Qwen2.5-VL
>
> Regarding the Qwen2.5-VL results, the "universal setting" demonstrates negligible improvement. Could the authors provide a deeper analysis of this? Furthermore, what is the authors' hypothesis regarding why VCD leads to performance degradation in this specific architecture?
>
> 4. MME Full Set Comparison
>
> Regarding the MME benchmark breakdown, it is necessary to include comparisons with other baselines beyond just Default and VCD to fully validate the method's advantages in preserving perception capabilities.

---

> > ### Author Response · Authors · 2025-11-28
> >
> > We sincerely appreciate your insightful comment regarding the gray-box scenario. We acknowledge that we initially viewed transferability primarily through the lens of black-box APIs, failing to appreciate its value for logit-based contrastive decoding without gradient access. Your comment broadened our perspective, helping us realize the potential of our method in scenarios where backpropagation is unavailable but probability distributions are accessible.
> >
> > We further investigated the feasibility of this setting. While applying contrastive decoding for long-form generation in gray-box scenarios is structurally constrained by the need for step-by-step intervention, we reasoned that it is immediately applicable to discriminative or verification tasks. To validate this, we conducted cross-architecture transfer experiments between LLaVA and Qwen-VL. Transferring from Qwen to LLaVA on POPE yielded 88.9 (Random), 86.9 (Popular), and 83.1 (Adversarial), while the reverse direction achieved 88.0, 86.8, and 84.1 respectively. On MME, we observed performance retention, with Qwen-to-LLaVA scoring 1462.4 and LLaVA-to-Qwen scoring 1328.1. These findings confirm that our adversarial perturbations capture semantic vulnerabilities, making AVCD viable for gray-box applications. We are grateful for this valuable insight.
> >
> > **1. Regarding Novelty**
> >
> > We agree with the reviewer that using large perturbation magnitude aligns with the methodological framework established by VCD. We do not claim the magnitude itself as a deviation. However, we respectfully emphasize our contribution in analyzing the underlying mechanism. While VCD empirically utilized large noise, our work provides the theoretical analysis and empirical analysis to explain why such magnitude is necessary to invert semantic similarity. We hope this mechanistic insight is valued as a meaningful step toward understanding contrastive decoding.
> >
> > **2. Regarding Greedy Decoding**
> >
> > We thank the reviewer for the insightful comment regarding the decoding strategy. We appreciate the opportunity to clarify our methodological choice and provide a rigorous fair comparison.
> >
> > We agree that standard VCD primarily utilizes sampling because random noise often yields a diffuse contrastive signal. This tends to flatten the probability distribution without necessarily dislodging the hallucination from the Top-1 rank, necessitating sampling to bypass the error. However, as demonstrated in our analysis, adversarial perturbations generate a highly discriminative contrast that effectively demotes the probability of the hallucinated token, allowing the ground truth to emerge as the dominant prediction.
> > - **Why greedy?:** Since AVCD successfully re-ranks the ground truth as the mode (highest probability token), greedy decoding is the most direct metric to validate this definitive correction. Using sampling in this context would introduce unnecessary stochasticity, obscuring the method's intrinsic capability.
> > - **Precedent:** Furthermore, VCD and subsequent methods like DeGF have also utilized or analyzed greedy/deterministic settings, confirming it is also a valid evaluation protocol.
> >
> > **Comparison under greedy decoding** To address the concern that greedy decoding might overestimate robustness compared to baselines using sampling, we conducted an additional comparison where compatible baselines were evaluated using greedy decoding. (Note: OPERA and HALC were excluded as they rely on beam search mechanisms.)
> >
> > | Method | CHAIRS↓ | CHAIRI↓ | POPE-R↑ | POPE-P↑ | POPE-A↑ | MME↑ |
> >  |:---|:---:|:---:|:---:|:---:|:---:|:---:|
> >  | AVCD | 22.1 | 7.3 | **89.2** | **87.0** | **83.0** | **1498** |
> >  | DeGF | 23.5 | 7.7 | 83.5 | 82.9 | 80.7 | 1373 |
> > | CICD | **21.8** | **7.2** | 88.3 | 86.3 | 81.6 | 1289 |
> > | ProjectAway | 26.4 | 8.8 | 85.5 | 84.7 | 82.3 | 1450 |
> > - AVCD achieves the highest scores in POPE and MME, outperforming all baselines. This proves that our method’s robustness is intrinsic to the logit calibration, not an artifact of the decoding strategy.
> > - While CICD shows a marginal advantage in CHAIR, it suffers a severe drop in MME, indicating a loss of general perception capabilities. In contrast, AVCD effectively reduce hallucinations while preserving the highest level of visual reasoning.
> > Our choice of greedy decoding was intended to highlight AVCD's unique capability to correct the Top-1 prediction, a feature less prominent in noise-based methods.

---

> > > ### Author Response · Authors · 2025-11-28
> > >
> > > **3. Regarding Qwen2.5-VL**
> > >
> > > We attribute the observed behaviors to the model's Visual Adapter (2x2 Pooling), which compresses visual information and effectively acts as a low-pass filter. This attenuates global perturbations applied to the raw input before they can influence generation. Consequently, although the attenuated signal results in a neutral impact on fine-grained perception tasks, it remains sufficient to serve as a strong global shock for flipping binary priors in coarse tasks like POPE. Furthermore, unlike VCD which causes degradation by disrupting structural cues, our gradient-guided perturbations respect the image manifold when attenuated.
> > >
> > > To overcome this attenuation, we implemented a structural adaptation that connects the attack magnitude and surface to the task granularity. While coarse verification tasks (POPE) necessitate a strong global shock (e.g., using a perturbation of $\epsilon=256$) on the raw input to forcibly flip high-confidence binary priors, such high-magnitude perturbations prove destructive for fine-grained perception. Therefore, to preserve visual details while ensuring steerability, it is essential to shift to injecting moderate perturbations (e.g., using $\epsilon=64$) directly into the processor. This validates that controlling such architectures requires a calibrated approach: leveraging raw-input shocks for binary decisions while reserving precise, internal feature injection for detail-oriented generation.
> > >
> > > Finally, regarding VCD's performance, we hypothesize that the dynamic resolution mechanism is particularly sensitive to unstructured noise. Random Gaussian noise likely interferes with the spatial coherence required for patch assembly, making it challenging to derive consistent contrastive gains compared to fixed-grid architectures.
> > >
> > > **4. Regarding MME**
> > >
> > >
> > >
> > > | LLaVA | existence | count | position | color | posters | celebrity | scene | landmark | artwork | OCR | Total |
> > > |-------|-----------|-------|----------|-------|---------|-----------|-------|----------|---------|------|-------|
> > > | Default | 185.0 | 148.3 | 135.0 | 160.0 | 127.6 | 120.0 | 157.8 | 162.3 | 116.5 | 137.5 | 1449.9 |
> > > | VCD | 180.3 | 144.3 | 130.0 | 162.0 | 124.8 | 129.0 | 155.2 | 164.7 | 121.1 | 137.5 | 1448.9 |
> > > | AVCD | 185.0 | 150.7 | 127.0 | 165.0 | 141.0 | 137.4 | 162.2 | 162.6 | 124.3 | 143.0 | 1498.0 |
> > > | ICD | 185.0 | 137.3 | 140.0 | 165.0 | 123.9 | 116.1 | 156.0 | 162.0 | 115.0 | 148.0 | 1451.1 |
> > > | M3ID | 185.0 | 138.3 | 130.0 | 160.0 | 122.4 | 112.9 | 157.8 | 163.3 | 115.3 | 132.5 | 1417.5 |
> > > | AVISC | 180.0 | 148.3 | 135.0 | 160.0 | 127.6 | 120.9 | 157.8 | 164.0 | 118.0 | 137.5 | 1448.4 |
> > > | DeGF | 195.0 | 148.3 | 111.7 | 153.3 | 118.0 | 106.8 | 153.3 | 150.8 | 114.5 | 140.0 | 1391.6 |
> > > | CICD | 185.0 | 113.3 | 111.7 | 115.0 | 158.5 | 132.1 | 166.3 | 150.3 | 105.0 | 77.5 | 1314.5 |
> > > | OPERA | 190.0 | 155.0 | 106.7 | 140.0 | 126.5 | 122.9 | 157.8 | 165.3 | 120.3 | 130.0 | 1414.4 |
> > > | HALC | 190.0 | 153.3 | 116.7 | 140.0 | 124.5 | 122.4 | 158.5 | 163.8 | 119.5 | 137.5 | 1426.1 |
> > > | ProjectAway | 185.0 | 143.3 | 146.7 | 160.0 | 127.6 | 121.8 | 159.3 | 160.8 | 115.3 | 125.0 | 1444.6 |
> > > | DeGF (greedy) | 190.0 | 146.7 | 118.3 | 145.0 | 109.2 | 101.8 | 155.3 | 151.3 | 118.0 | 137.5 | 1372.9 |
> > > | CICD (greedy) | 180.0 | 103.3 | 105.0 | 110.0 | 156.5 | 135.0 | 162.3 | 155.8 | 110.8 | 70.0 | 1288.6 |
> > > | ProjectAway (Greedy) | 180.0 | 148.3 | 135.0 | 160.0 | 127.6 | 122.4 | 157.8 | 163.8 | 117.5 | 137.5 | 1449.7 |
> > >
> > >
> > > | Qwen-VL | existence | count | position | color | posters | celebrity | scene | landmark | artwork | OCR | Total |
> > > |-------|-----------|-------|----------|-------|---------|-----------|-------|----------|---------|------|-------|
> > > | Default | 170.0 | 145.0 | 98.3 | 180.0 | 164.3 | 85.9 | 155.0 | 95.5 | 109.8 | 50.0 | 1253.8 |
> > > | VCD | 170.0 | 143.0 | 103.3 | 182.0 | 166.9 | 110.8 | 154.2 | 113.0 | 118.0 | 50.0 | 1310.7 |
> > > | AVCD | 171.0 | 150.0 | 113.3 | 182.0 | 173.8 | 103.8 | 155.2 | 118.3 | 123.0 | 50.0 | 1340.3 |
> > > | ICD | 161.0 | 143.0 | 100.3 | 167.0 | 154.5 | 84.9 | 149.5 | 89.2 | 108.4 | 50.0 | 1207.9 |
> > > | M3ID | 165.0 | 150.0 | 98.3 | 180.0 | 168.4 | 87.6 | 155.8 | 98.5 | 109.5 | 50.0 | 1263.1 |
> > > | AVISC | 170.0 | 150.0 | 98.3 | 180.0 | 164.3 | 86.8 | 157.8 | 99.3 | 110.3 | 50.0 | 1266.6 |
> > >
> > > While some baselines may achieve marginally higher scores in specific sub-tasks depending on their optimization focus, AVCD demonstrates the most robust and balanced performance across the full spectrum of perception tasks. Unlike methods that exhibit high variance—excelling in one metric while degrading in others—AVCD consistently maintains high performance across diverse categories. This consistency allows AVCD to achieve the highest Total Score among all evaluated methods. This validates that our approach effectively mitigates hallucinations while preserving and stabilizing the model's general perceptual capabilities.

---

### Official Review · Reviewer_nffn · 2025-10-26

**Soundness:** 3
**Presentation:** 2
**Contribution:** 3
**Rating:** 4
**Confidence:** 3

**Summary:**

The paper proposes Adversarial Visual Contrastive Decoding (AVCD), a training-free inference method. AVCD replaces VCD’s Gaussian noise with directional adversarial perturbations (generated via Projected Gradient Descent, PGD). These perturbations are optimized to minimize the cosine similarity between the vision encoder features of original and adversarial images, steering the model toward hallucinatory states (instead of just degrading features). During decoding, AVCD contrasts logits from the original and adversarial images to suppress hallucinatory content.

**Strengths:**

**Novel Conceptual Foundation:** It introduces a clever and novel repurposing of adversarial attacks. Rather than treating them solely as a vulnerability, it leverages them as a constructive tool to generate powerful "hard negative" signals for contrastive decoding.

**Strong Empirical Validation:** The method is rigorously evaluated across three standard benchmarks (POPE, CHAIR, MME) and multiple model architectures (LLaVA, Qwen-VL, etc.), consistently demonstrating superior performance over strong baselines like VCD.

**Weaknesses:**

1. Sensitivity to Hyperparameters
AVCD's performance is highly dependent on the careful tuning of several hyperparameters, which can complicate its practical deployment.
The optimal adversarial perturbation strength (ϵ) varies by task: ϵ=256 works best for the POPE benchmark, while ϵ=64 is optimal for CHAIR and MME.Also, this task-dependent tuning requirement increases the burden of finding optimal settings for new applications.

2. While the authors provide a theoretical proof (Theorem 1) to explain AVCD's effectiveness, it relies on several simplifying assumptions that limit its real-world applicability.

    ***Single Object Assumption***: The proof only considers a scenario where the model chooses between one ground-truth object (x) and one specific non-existent object (x'). In reality, LVLMs perform open-vocabulary generation from a vast candidate set of possible hallucinations, a complexity not captured by the proof.

    ***First-Order Taylor Approximation***: The proof uses a linear approximation of the highly non-linear vision encoder. This assumption becomes increasingly inaccurate with large perturbation budgets (ϵ), making the derived theoretical threshold less reliable.

    ***High-Dimensional Orthogonality***: The proof assumes that residual vectors in the embedding space are orthogonal. However, adversarial perturbations are directionally optimized, not random, and may actively align with specific semantic concepts, violating the orthogonality assumption.

3. The paper does not sufficiently address the robustness and transferability of the generated adversarial images. It is unclear how stable the adversarial perturbations are across different but semantically similar images.

4. AVCD is inherently a white-box method, which restricts its usage scenarios. Generating effective adversarial perturbations requires full access to the gradients of the vision encoder. But such kind of methods all have this internal issues.

**Questions:**

Authors are suggestted to address my concerns at the *Weakness* part as many as you can, no need to be fully addressed.

---

> ### Author Response · Authors · 2025-11-21
>
> **Regarding Hyperparameter Sensitivity (W1)**
>
> We thank the reviewer for the feedback regarding hyperparameter sensitivity. We acknowledge that, similar to other inference-time intervention methods, AVCD involves parameter selection; however, our design significantly constrains the search space to a simple choice, effectively minimizing the tuning burden compared to methods requiring extensive per-task engineering.
>
> Throughout our experiments, we utilized only two distinct configurations for $\epsilon$ based on the task type: a Default Setting ($\epsilon=64, T_{adv}=2$) for all generative tasks and complex binary benchmarks (e.g., CHAIR, MME), and a Specialized Setting ($\epsilon=256, T_{adv}=1$) exclusively for simple object existence verification tasks (e.g., POPE).
>
> Our empirical results consistently demonstrate that this distinction is robust. The Default Setting proved optimal for tasks requiring detailed visual grounding and reasoning (CHAIR, MME), where preserving fine-grained features is crucial. Conversely, the Specialized Setting was necessary only for POPE, where a stronger perturbation is required to challenge the model's high-confidence priors on binary existence questions.For practical deployment, this dichotomy offers a straightforward heuristic: using the specialized setting ($\epsilon=256$) specifically for simple binary verification and the default setting ($\epsilon=64$) for all other general multimodal scenarios. While we plan to explore adaptive scheduling in future work, this simple logic already offers a robust and effective solution with minimal tuning complexity.

---

> ### Author Response · Authors · 2025-11-21
>
> **Regarding Theoretical Validity (W2)**
>
> We thank the reviewer for the rigorous examination of our theoretical analysis. We acknowledge that our initial proof relied on simplifying assumptions to provide intuition. However, to address the concerns regarding the approximation limits and assumptions, we offer the following clarifications and improvements, which will be incorporated into the final manuscript.
>
> **1. Single Object Assumption**
>
> We respectfully clarify that our theoretical framework does not assume a single-object scenario. In our formulation, $x$ denotes the ground-truth text description, which can represent a caption containing multiple objects. As detailed in Section 3.3 of our manuscript, our empirical analysis explicitly utilizes multi-object templates (e.g., "There are [object 1], [object 2]...") to represent $x$.
> We acknowledge that the phrasing "an object description text" in our original manuscript could lead to the misunderstanding that it refers to a single entity. To prevent this ambiguity, we will revise the terminology in Theorem 1 to "a text describing objects". We thank the reviewer for pointing out this potential confusion.
>
> Furthermore, we acknowledge that open-vocabulary generation involves choosing from a vast candidate set beyond a simple choice. However, our theoretical analysis provides a simplified but fundamental insight: adversarial perturbations can invert the similarity relationship between correct and incorrect descriptions. This principle holds regardless of the vocabulary size, as the mechanism fundamentally steers the embedding away from the ground truth ($x$) and creates a generalized bias toward hallucinatory tokens ($x'$).
>
> **2. First-Order Taylor Approximation**
>
> We agree with the reviewer that the First-Order Taylor Approximation is a local estimate and decreases in accuracy under large perturbation budgets ($\epsilon$).To address this, we first recall the premise of our method: AVCD does not impose a strict upper bound on the perturbation budget $\epsilon$. Consequently, the theoretical challenge is not to derive a precise lower bound for $\epsilon$ via local approximation, but rather to prove whether a valid $\epsilon$ (and its corresponding perturbation $\delta$) exists and is defined within the input space that satisfies the condition ($a < 0$). Therefore, we replace the approximation-based derivation with a more fundamental "Existence Proof":
> - **Continuity & Codomain Definition:** The vision encoder ($f$) is a continuous function. Crucially, the codomain of $f$ (the embedding space) is defined over the field of real numbers. Due to architectural components like Layer Normalization and linear projectors (which do not restrict outputs to non-negative), the encoder is capable of producing output vectors with negative cosine similarities relative to the ground truth $f(I)$.
>
> - **Existence of Perturbation:** Since the codomain allows for vectors where $cos(f(I), v) < 0$, and the function is continuous over the pixel space, there must exist an input perturbation $\delta$ within the input space such that the output embedding creates a negative cosine similarity ($a < 0$).
>
> We emphasize that our empirical validation (Figure 3) complements this, demonstrating that for $\epsilon \ge 64$, relative similarity becomes negative, validating practical achievability.
>
> **3. High-Dimensional Orthogonality**
>
> We clarify the mathematical definition used in our proof regarding the residual vector. In Equation 12, we decompose the perturbed feature vector as $f(I+\delta) = a f(I) + \nu_I(\delta)+e_I(\delta)$. By the definition of orthogonal decomposition, any vector $f(I+\delta)$ can be uniquely split into a component parallel to $f(I)$ and a component orthogonal to $f(I)$. If the residual vector $\nu_I(\delta)$ were not orthogonal to $f(I)$—i.e., if it contained a component parallel to $f(I)$—it could be expressed as:$\nu_I(\delta) = a'f(I) + \nu'_I(\delta)+e'_I(\delta)$, where $\nu'_I(\delta)$ is the strictly orthogonal component. Substituting this back into the decomposition yields:$f(I+\delta) = (a + a')f(I) + \nu'_I(\delta)+e_I(\delta)+e'_I(\delta)$. In this case, the scalar $a'$ is simply absorbed into the alignment coefficient (becoming the new $a$), leaving the remaining residual term $\nu'_I(\delta)$ strictly orthogonal by definition.
>
> In essence, the adversarial optimization affects the parallel component ($a$), driving it negative, while the residual term remains orthogonal by mathematical definition rather than by assumption. his holds regardless of whether $\delta$ is optimized. Our extensive empirical results (Sections 3 & 4) confirm that this core insight—adversarial perturbations create directional hallucination signals—translates effectively to practical systems.

---

> ### Author Response · Authors · 2025-11-21
>
> **Regarding Robustness and Transferability (W3)**
>
> We appreciate the reviewer’s suggestion to investigate the stability of perturbations. To clarify the intended scope of our method: AVCD focuses on hallucination mitigation via inference-time optimization, rather than studying adversarial robustness in the traditional sense. Since our pipeline generates perturbations on-the-fly specifically for the target image, transferability is not a functional requirement for our application. However, we agree that examining transferability provides valuable insights into whether the perturbations capture general feature sensitivities or are strictly instance-specific.
>
> **Experimental Setup: Cross-Image Transfer:** To test this, we applied adversarial perturbations ($\delta = x_{adv} - x$) generated from a Source Image to a different Target Image. We tested three source (dataset) scenarios:
> - Random: $\delta$ from a randomly selected image.
> - Nearest: $\delta$ from the nearest neighbor in the vision encoder feature space (semantically similar).
> - Farthest: $\delta$ from the farthest image in the feature space.
>
> **Experimental Results:** We evaluated the performance on CHAIR, MME, and POPE benchmarks.
> | Method | CHAIRS↓ | CHAIRI↓ | MME↑ | POPE-R↑ | POPE-P↑ | POPE-A↑ |
> |:-------|:-------:|:-------:|:------:|:------------:|:-------------:|:---------:|
> | Baseline (Greedy) | 26.9 | 8.9 | 1449.9 | 85.8 | 84.9 | 82.5 |
> | **AVCD (Ours)** | **22.0** | **7.3** | **1498.0** | **89.1** | **87.2** | **83.2** |
> | Random | 25.4 | 8.5 | 1463.2 | 88.1 | 86.7 | 83.1 |
> | Nearest | 24.3 | 8.2 | 1373.6 | 85.9 | 84.8 | 82.3 |
> | Farthest | 23.9 | 8.1 | 1369.2 | 86.0 | 85.2 | 82.7 |
>
> **Analysis**
> - **Superiority of Instance-Specificity:** Our Instance-Specific AVCD consistently outperforms all transfer baselines across all tasks. This confirms that effective hallucination mitigation requires tailored perturbations that target the specific visual features of the current image.
> - **Task-Dependent Impact of Transfer:**
>   - CHAIR: Transfer-based perturbations yielded slight improvements over the baseline (e.g., Nearest 24.3 vs. Baseline 26.9). In open-ended captioning, even unaligned perturbations can act as noise that disrupts the model's over-reliance on language priors, slightly reducing hallucinations.
>   - POPE/MME: Conversely, on the other benchmarks which requires precise visual verification, transferring perturbations (Nearest/Farthest) degraded performance (dropping to ~1370 for MME) compared to the baseline. This suggests that borrowed perturbations act as distraction that corrupts visual details without providing the correct steering signal.
>
> These results reinforce that while generic perturbations may offer marginal benefits in generation by breaking priors (CHAIR), per-image optimization is essential to simultaneously suppress hallucinations and maintain precise visual grounding (POPE/MME).
>
> **Regarding White-Box Assumption (W4)**
>
> We thank the reviewer for the accurate assessment. We fully agree with the observation that the white-box requirement is a structural characteristic shared by nearly all inference-time hallucination mitigation methods, which inherently limits their direct application to closed-source models. However, unlike other internal intervention methods, AVCD generates an input-level artifact that possesses Adversarial Transferability. This unique property allows us to overcome this internal issue and extend AVCD to black-box scenarios using surrogate models.
>
> As this concern regarding black-box applicability was raised by multiple reviewers, we have provided a comprehensive experimental validation (on GPT and Gemini) in the **Common Response: Applicability to Black-Box Models** section above. Please refer to that section for the detailed results.

---

> > ### Comment · Reviewer_nffn · 2025-11-27
> >
> > Thank you for the clarification regarding the mathematical definition used in Equation 12. I acknowledge that, by the definition of orthogonal decomposition, the residual vector $\nu_I(\delta)$ is indeed orthogonal to the source image embedding $f(I)$ (i.e., $\nu_I(\delta) \perp f(I)$).However, the rebuttal does not address the core of my concern, which lies in the statistical assumptions made later in the proof, specifically in Appendix B (Equations 19, 20, and 21).The validity of Theorem 1 relies on the assumption that the cross-term between the perturbation residual and the random text residual is negligible, i.e., $\nu_I(\delta) \cdot \nu_I(x') \approx 0$. The proof justifies this using Equation 19, which invokes the concentration of measure for high-dimensional vectors. Crucially, this justification explicitly assumes that the vectors are chosen randomly from orthogonal subspaces.My concern remains that adversarial perturbations ($\delta$) generated via PGD are not random; they are optimized directionally to maximize specific feature activations. If the adversarial attack successfully steers the image embedding toward the semantic space of a hallucinatory concept $x'$, then $\nu_I(\delta)$ will likely be highly correlated (aligned) with $\nu_I(x')$.
> > Consequently, the assumption of randomness in Equation 19 is violated, and the upper bound derived in Equation 20 may not hold. While the rebuttal correctly defends the orthogonality relative to the source $f(I)$, it fails to justify the orthogonality relative to the target $\nu_I(x')$ under non-random, adversarial optimization.

---

> ### Author Response · Authors · 2025-11-27
>
> We thank the reviewer for this insightful observation regarding the statistical validity of our proof. It has helped us refine the theoretical rigor of our work regarding the independence assumption.
>
> We acknowledge that if an adversarial attack were **targeted** (i.e., optimized to steer toward a specific $x'$), the orthogonality assumption would be violated due to the alignment between the perturbation and the target. However, we respectfully posit that the assumption holds with high probability in our context because AVCD employs an **untargeted** attack mechanism.
>
> **1. Statistical Independence via Causal Disconnection**
> In our framework, the adversarial perturbation $\delta$ is optimized solely to minimize the cosine similarity with the original image features $f(I)$ using the vision encoder only. Crucially, this optimization process is **blind** to the specific random text $x'$.
> - $\delta = \phi(I)$ is a function dependent only on the image $I$ and its gradient.
> - $x'$ is an arbitrary description chosen independently of the optimization process.
>
> Since the generation of $\delta$ has no causal link to the choice of $x'$, the direction of the perturbation **residual** $\nu_I(\delta)$ remains **statistically independent** of the random text **residual** $\nu_I(x')$.
>
> **2. Revision of Theorem 1**
> The reviewer correctly implies that while unlikely, a coincidental alignment between $\delta$ and $x'$ is theoretically possible. The original theorem statement did not fully capture this probabilistic nature of high-dimensional geometry.
> To address this and enhance theoretical rigor, we will revise Theorem 1 to explicitly state that the inequality holds "with high probability" rather than absolutely. This revision grounds our claim in the Concentration of Measure phenomenon, which guarantees that two independent vectors in a high-dimensional space are nearly orthogonal with overwhelming probability.
>
> **Theorem 1 (Revised).** Given an image $I$, its ground-truth text $x$, and an arbitrary random text $x' \neq x$ (chosen independently), then for a sufficiently large perturbation budget $\epsilon$, an untargeted adversarial perturbation $\delta$ ($||\delta|| \leq \epsilon$) satisfies the following:
> $$\mathbb{P} \left[ \cos(f(I+\delta), g(x)) < \cos(f(I+\delta), g(x')) \right] \ge 1 - 2\exp(-c \cdot r)$$where $r$ is the rank (effective dimension) of the image embedding subspace and $c > 0$ is a constant determined by the concentration of measure.
>
> Crucially, the failure probability term $2\exp(-c \cdot r)$ decays exponentially with the rank $r$. As noted in our Appendix B, while $r$ may be smaller than the ambient dimension $d$, modern vision encoders typically exhibit a high effective rank (e.g., sufficiently large $r$ in a 768-dimensional space), rendering this probability effectively 1 for all practical purposes.
>
> **3. Empirical Validation**
> This theoretical guarantee of near certainty is corroborated by our empirical results (Figure 3). The relative similarity change consistently drops below zero, confirming that the untargeted perturbation successfully moves the embedding away from the ground truth without needing to align with any specific $x'$, thereby statistically favoring arbitrary hallucinations as predicted.
>
> We hope that explicitly framing the theorem as a probabilistic statement—grounded in the independence inherent in our untargeted approach—satisfactorily addresses your concern regarding the theoretical validity.

---

> > ### Comment · Reviewer_nffn · 2025-11-27
> >
> > I thank the authors for their thoughtful and detailed response. The distinction made between targeted and untargeted attacks effectively addresses my concern regarding the statistical independence of the residuals. I am satisfied with this explanation and the proposed revision to the theorem. Authors should include these points into their next version.
> > I decide to raise my score and will keep this score as my final rating.

---

> ### Author Response · Authors · 2025-11-27
>
> **Thank you for your support and score update**
>
> We sincerely thank you for your engagement and for reconsidering your assessment. We are glad that our clarification regarding the theoretical analysis satisfactorily addressed your concerns. **We have revised the manuscript** to incorporate the revised theorem (probabilistic statement) and the discussion on statistical independence. We deeply appreciate your constructive feedback, which has significantly strengthened the theoretical rigor of our work.

---

### Official Review · Reviewer_ULJQ · 2025-10-27

**Soundness:** 3
**Presentation:** 4
**Contribution:** 3
**Rating:** 6
**Confidence:** 4

**Summary:**

This paper proposes Adversarial Visual Contrastive Decoding (AVCD), an inference-time method that improves Visual Contrastive Decoding (VCD) by introducing adversarial perturbations instead of Gaussian noise to create more informative contrastive signals. The adversarially perturbed image is optimized to reduce cosine similarity with the original image, thereby generating a stronger signal for suppressing hallucinations in LVLMs. The paper includes solid theoretical analysis, encoder- and LLM-level studies, and comprehensive experiments demonstrating consistent improvements over VCD and related baselines.

**Strengths:**

1. Repurposes adversarial perturbations as constructive contrastive signals, which is both original and well-motivated.

2. Provides formal analysis (Theorem 1) and encoder/LLM-level verification, showing directional semantic steering rather than random degradation.

3. Evaluations across multiple benchmarks (POPE, CHAIR, MME) and models (LLaVA, Qwen-VL) demonstrate consistent gains.

4. Training-free and computationally moderate (≈1.2× VCD cost), making it appealing for real-world use.

**Weaknesses:**

1. The paper employs a very large adversarial budget (ε = 64–256), far beyond common adversarial settings (e.g., 4/255 or 8/255) that already induce substantial semantic variation. While this design choice may partially offset the low computational cost of using only 1–2 PGD steps and a simplified adversarial loss, the trade-off between perturbation magnitude and optimization depth (i.e., larger ε versus more PGD steps with smaller ε) is not analyzed. A clearer justification or empirical study of this balance would substantially strengthen the paper’s methodological soundness.

2. Limited adversarial formulation.
The adversarial perturbations are derived solely from the vision encoder features, which represent only a small component of the overall LVLM. It is uncertain whether this restricted gradient path adequately captures hallucination-relevant semantics. Generating perturbations based on multimodal or output-level objectives—such as semantic deviation of the LVLM’s logits using Zero-Gradient or latent-space alignment methods [1]—could produce more meaningful and semantically targeted adversarial signals (albeit at higher cost). I recommend that the authors include results under alternative adversarial objectives to substantiate this point.


3.	Insufficient benchmark coverage. The evaluation lacks results on fine-grained and more complex generative task hallucination benchmarks, which are essential for validating faithfulness and factual grounding [2,3,4]. Including a small-scale test on recent benchmarks such as MMHal-Bench [3] would significantly strengthen the empirical claims and generalizability.

4.	PGD step analysis missing. The paper does not analyze how performance scales with the number of PGD iterations (T_adv). It remains unknown whether a smaller ε combined with more steps could achieve comparable results, which would clarify whether the observed improvements stem from the adversarial directionality or simply the perturbation magnitude.

5.	Alternative contrast construction unexplored. The approach relies on contrast between the original and its adversarially distorted version. It would be informative to examine whether similar benefits arise when contrasting the original with unrelated or semantically opposite images (e.g., grayscale or blank images), to isolate the contribution of adversarial directionality from generic dissimilarity.

6.	Minor technical imprecision. The claim that “Unlike attacks designed for specific objectives like classification, PGD can optimize any given loss function” (L165) is misleading. PGD is an optimization procedure, not an objective-specific method; its flexibility arises from the chosen loss function, not from PGD itself.

[1] On Evaluating Adversarial Robustness of Large Vision-Language Models, NeurIPS 2023.

[2] Amber: An LLM-Free Multi-Dimensional Benchmark for MLLMs Hallucination Evaluation, 2023.

[3] Aligning Large Multimodal Models with Factually Augmented RLHF, ACL 2024.

[4] RLHF-V: Towards Trustworthy MLLMs via Behavior Alignment from Fine-Grained Correctional Human Feedback, CVPR 2024.

**Questions:**

See Weakness

**Details Of Ethics Concerns:**

No concerns

---

> ### Author Response · Authors · 2025-11-21
>
> **Regarding Adversarial Budget and Optimization Depth (W1, W4)**
>
> We deeply resonate with the reviewer’s perspective. Indeed, we initially approached this problem using the standard adversarial setting (small $\epsilon \approx 8/255$ with 10 PGD steps), assuming that conventional attack wisdom would apply. However, we discovered that the specific nature of our task necessitates a different approach. Unlike traditional attacks where imperceptibility is a hard constraint, the adversarial image in AVCD is purely an internal intermediate artifact that is never exposed to the user. This "liberation" from the imperceptibility constraint allowed us to explore larger perturbation budgets.
>
> To rigorously justify our final choice ($\epsilon=64\sim256$, Steps=1-2), we present a comparative study between the Traditional Setting and our AVCD Setting, which are also visually detailed in Figure 7 ($\epsilon$ ablation) and Figure 8 (Step ablation) in Appendix F of our paper.
>
> **Empirical Comparison: Magnitude vs. Depth:** We compared the performance of a standard adversarial setting (low $\epsilon$, high steps) against our proposed setting (high $\epsilon$, low steps) across three benchmarks.
>
> | epsilon | steps | CHAIRS↓ | CHAIRI↓ | MME↑ | POPE-R↑ | POPE-P↑ | POPE-A↑ |
> |:-------:|:-----:|:-------:|:-------:|:------:|:------------:|:-------------:|:---------:|
> | Baseline | - | 26.9 | 8.9 | 1450 | 85.8 | 84.9 | 82.5 |
> | 8 | 10 | 23.1 | 8.0 | 1484 | 88.0 | 85.5 | 80.0 |
> | 16 | 10 | 23.5 | 7.8 | 1483 | 87.9 | 85.8 | 80.3 |
> | 32 | 10 | 23.1 | 7.6 | 1490 | 87.8 | 85.6 | 80.2 |
> | 64 | 2/1* | **22.0** | **7.3** | **1498** | 86.7 | 85.0 | 82.5 |
> | 128 | 2/1* | 24.0 | 7.7 | 1486 | 87.7 | 86.5 | 82.8 |
> | 256 | 2/1* | 23.7 | 7.8 | 1448 | **89.1** | **87.2** | **83.2** |
> *Note: 2 steps for CHAIR and MME, and 1 step for POPE (epsilon ≥ 64)
>
> **Justification for "Large Perturbation with Small Steps":** The results lead to two critical insights that justify our design:
> - **Comparability vs. Superiority:** As shown in the table, the traditional setting (small $\epsilon$ with 10 steps) does yield performance gains over baselines, proving it is a valid approach. However, our proposed setting (large $\epsilon$ with 1-2 steps) consistently achieves superior performance across all benchmarks.
> - **Efficiency of Magnitude over Depth:** Since we are not bound by visual constraints, we found that a single, large step provides a more potent contrastive signal than ten small steps. This allows us to achieve better hallucination suppression while maintaining high computational efficiency (1-2 steps vs. 10 steps).
>
> Our choice of parameters is a deliberate adaptation to the problem structure. By leveraging the freedom to use larger perturbations, we achieve both superior performance and higher computational efficiency compared to the traditional setting, validating the methodological soundness of our approach.

---

> ### Author Response · Authors · 2025-11-21
>
> **Regarding Adversarial Formulation (W2)**
>
> We thank the reviewer for the insightful suggestion. We fully accept the criticism that the justification for utilizing the vision encoder loss was not explicitly detailed in the main text. We agree that this explanation is crucial for the methodological soundness of our paper. To address the reviewer's concern, we conducted comprehensive ablation studies comparing our method against alternative adversarial objectives, including those suggested.
>
> **Why Vision Encoder Loss? (Efficiency & Scalability):** While output-level objectives (e.g., maximizing cross-entropy on LLM logits) are theoretically sound, they introduce severe scalability issues:
> - **Computational Complexity ($O(N)$ vs $O(1)$):** Output-level attacks typically require backpropagating through the massive LLM for every token generation step ($O(N)$). In contrast, our method requires only a single, lightweight gradient pass through the vision encoder ($O(1)$).
> - **Incompatibility with Caching:** Output-level attacks often necessitate updating the image dynamically, breaking the KV-caching mechanism. Our method generates a single static adversarial image, fully compatible with standard caching.
>
> **Empirical Comparison with Alternative Objectives** We implemented and evaluated four alternative formulations on the CHAIR benchmark. Note that Alternatives 1 and 3 are untargeted variants of the strategies defined in [1], designed to push features away from the current prediction rather than matching a target.
>
> - **AVCD (Ours)**: Minimizing cosine similarity of visual features.
> - **Alternative 1 (Text-Image Align):** Using the Text Encoder to maximize the distance between image features and the embedding of the generated caption (Variant of MF-it [1]).
> - **Alternative 2 (Hybrid):** Combining our original loss with Alternative 1 (AVCD + Alternative 1).
> - **Alternative 3 (Black-box Latent):** It generates a response, encodes it via the Text Encoder, and uses Zeroth-Order optimization [2] to push the image away from this text embedding (Variant of MF-tt [1]).
> - **Alternative 4 (Logit Attack):** Direct maximization of Cross-Entropy loss on the LLM's output logits.
>
> | Method | Objective / Strategy | CHAIRS↓ | CHAIRI↓ |
> |:-------|:--------------------|:-------:|:-------:|
> | Baseline | Greedy | 26.9 | 8.9 |
> | VCD | Diffusion Noise | 25.2 | 8.6 |
> | Alternative 1 | Text-Image Align (MF-it variant) | 23.8 | 7.7 |
> | Alternative 2 | Hybrid (Ours + MF-it) | 23.9 | 7.8 |
> | Alternative 3 | Black-box Latent (MF-tt variant) | 23.9 | 8.1 |
> | Alternative 4 | Logit Attack | 23.6 | 7.9 |
> | **AVCD (Ours)** | **Vision Encoder Loss** | **22.0** | **7.3** |
>
> (Note: Due to the $O(N)$ complexity, the Alternative 4 experiment was restricted to the first output token only. We also evaluated Alternative 4 on POPE (Random-F1: 88.92) and MME (Score: 1462.82). While it shows comparable performance to AVCD on POPE (89.10) and underperforms on MME (1498.0), it incurs higher computational cost.)
>
> All adversarial formulations outperformed the baselines, validating the adversarial approach. However, AVCD  achieved the best overall performance with the highest efficiency. Since the target of our perturbation is the image space, optimizing the objective directly within the visual representation space (Vision Encoder) appears to generate more robust gradients. Our results confirm that our Vision Encoder Loss strikes the optimal balance. It provides the strongest hallucination suppression while avoiding the prohibitive costs associated with multimodal (MF-it/tt) or output-level optimization.
>
> [1] Zhao, Yunqing, et al. "On evaluating adversarial robustness of large vision-language models." Advances in Neural Information Processing Systems 36 (2023): 54111-54138. \
> [2] Chen, Pin-Yu, et al. "Zoo: Zeroth order optimization based black-box attacks to deep neural networks without training substitute models." Proceedings of the 10th ACM workshop on artificial intelligence and security. 2017.

---

> ### Author Response · Authors · 2025-11-21
>
> **Insufficient Benchmark Coverage (W3)**
>
> We sincerely thank the reviewer for the constructive suggestion to include MMHal-Bench. We agree that evaluating on fine-grained and complex generative benchmarks is crucial for validating faithfulness.
>
> **New Experiment on MMHal-Bench:** Following the reviewer's recommendation, we conducted an evaluation on MMHal-Bench, which is specifically designed to probe detailed visual attributes, counting, and spatial relationships—areas where standard benchmarks often fall short. We utilized our Default Setting ($\epsilon=64, T_{adv}=2$) used for other generative tasks like CHAIR. The results demonstrate that AVCD significantly improves performance over baselines on this benchmark.
>
> | Method | Baseline | VCD (500) | VCD (999) | AVCD ($\epsilon=64$) |AVCD ($\epsilon=256$) |
> | :---: | :---: | :---: | :---: | :---: | :---: |
> | Score | 1.93 | 2.01 | 2.01 | **2.14** | 2.07 |
>
> **Analysis of Fine-Grained Capabilities:** AVCD achieves the highest score (2.14), outperforming both the default (1.93) and VCD (2.01). This explicitly addresses the concern about faithfulness and factual grounding. MMHal contains various question types that require precise visual grounding. The performance gain confirms that our adversarial perturbations do not distort the fine-grained cues needed for these tasks; rather, they effectively suppress the model's tendency to hallucinate incorrect attributes based on language priors.
>
> **Regarding Alternative Contrast Construction (W5)**
>
> We thank the reviewer for the insightful suggestion. We agree that it is essential to verify whether the performance gains stem specifically from adversarial directionality or simply from contrasting with generic dissimilarity. To address this, we expanded our evaluation to include a wide range of alternative contrastive baselines. In fact, during the early stages of our research, we extensively explored various non-adversarial transformations. Following the reviewer's recommendation, we have also added baselines using unrelated constant images (Black, White, Average).
>
> **Comprehensive Comparison with Alternative Contrasts:** We compared AVCD against two categories of alternatives:
> - **Unrelated/Constant Images:** Black, White, and Average (Gray) images.
> - **Standard Image Transformations:** Blur, Color Jitter, Crop, Enhance Edge, Flip, Grayscale, Histogram Equalization, Increased Contrast, and Rotation.
>
> **Experimental Results:** The results on CHAIR, MME, and POPE benchmarks are summarized below:
>
> | Method | CHAIRS↓ | CHAIRI↓ | MME↑ | POPE-R↑ | POPE-P↑ | POPE-A↑ |
> |:-------|:-------:|:-------:|:------:|:------------:|:-------------:|:----------------:|
> | Baseline | 26.9 | 8.9 | 1449.9 | 85.76 | 84.94 | 82.49 |
> | AVCD | **22.0** | **7.3** | **1498.0** | **89.1** | **87.2** | **83.2** |
> | Black | 25.2 | 8.4 | 1418.0 | 87.67 | 84.21 | 78.6 |
> | White | 25.6 | 8.6 | 1409.5 | 87.17 | 83.19 | 77.8 |
> | Average | 26.1 | 8.6 | 1389.7 | 87.19 | 83.42 | 77.9 |
> | Blur | 25.0 | 8.6 | 1452.5 | 85.7 | 84.9 | 82.1 |
> | Color Jitter | 27.0 | 9.1 | 1458.4 | 85.9 | 84.5 | 81.8 |
> | Random Crop | 27.6 | 9.8 | 1422.9 | 84.6 | 84.0 | 81.1 |
> | Random Flip | 25.6 | 8.9 | 1425.2 | 85.4 | 84.6 | 81.9 |
> | Grayscale | 25.8 | 8.7 | 1463.1 | 85.3 | 84.5 | 82.0 |
> | Random Rotation | 26.4 | 9.2 | 1450.6 | 85.8 | 84.8 | 81.8 |
>
> **Analysis**
> - **Ineffectiveness of Generic Dissimilarity:** As shown in the table, contrasting with generic constant images (Black/White/Average) or standard transformations fails to achieve comparable performance to AVCD. In many cases (e.g., Crop, Color Jitter), the performance is even worse than the baseline.
> - **Superiority of Adversarial Directionality:** AVCD consistently outperforms all alternative strategies across all benchmarks. This empirically confirms that the specific adversarial directionality—which actively steers the model towards hallucinatory states—is crucial for providing a meaningful negative signal for contrastive decoding. Generic dissimilarity merely introduces noise or irrelevant signals that do not effectively isolate and suppress hallucination modes.
>
> **Regarding Technical Precision (W6)**
>
> We thank the reviewer for the precise correction. We agree that PGD is a general optimization procedure and that the flexibility stems from the definition of the loss function itself, rather than the optimization algorithm. The original sentence was intended to convey that we utilize PGD to optimize a custom feature-level objective, distinct from standard classification-based adversarial attacks. We will revise the sentence in the final manuscript to be technically precise.
>
> Revision:
> - Original: "Unlike attacks designed for specific objectives like classification, PGD can optimize any given loss function."
> - Revised: "We employ PGD as a generic optimization framework to maximize our custom feature-level adversarial loss, distinguishing our approach from attacks tied to standard classification objectives."

---

> > ### Comment · Reviewer_ULJQ · 2025-11-21
> > **Rebuttal Acknowledged and Rating Raised**
> >
> > I thank the authors for their detailed response and for conducting the additional experiments I requested.
> >
> > After reviewing the rebuttal and the new results, the authors have effectively addressed my main concerns:
> >
> > **(1) Adversarial Budget and Formulation:** I appreciate the clarification regarding the trade-off between perturbation magnitude and optimization depth. Although the large epsilon value is unconventional compared to standard adversarial attacks, I agree that the imperceptibility constraint can be relaxed here since the adversarial image is strictly an internal intermediate artifact. The empirical evidence shows that a single large step outperforms multi-step small perturbations while being significantly more efficient. This justifies the design choice for practical deployment.
> >
> > **(2) Benchmark Coverage:** The inclusion of results on MMHal-Bench is a valuable addition. The performance gain demonstrates that the method improves faithfulness even on tasks requiring fine-grained visual grounding. This mitigates my prior concerns regarding generalizability.
> >
> > **(3) Alternative Contrasts:** The comparison against generic transformations convinces me that the performance gains arise specifically from the adversarial directionality rather than generic visual dissimilarity.
> >
> > Given the solid empirical performance, the efficiency of the proposed method, and the comprehensive rebuttal, I believe this paper makes a practical contribution to the field. I am raising my rating accordingly to support its acceptance.

---

> > > ### Author Response · Authors · 2025-11-21
> > >
> > > **Thank you for the positive feedback and raising the rating**
> > >
> > > We sincerely thank the reviewer for the detailed feedback and for raising the rating. We are encouraged to learn that our clarifications regarding the adversarial budget trade-off, the additional MMHal benchmark results, and the alternative contrasts have effectively addressed your concerns.
> > >
> > > We appreciate your constructive comments, which have significantly strengthened the generalizability and robustness of our work. We have already incorporated all the discussed results into the current revised manuscript.

---

### Official Review · Reviewer_PwPs · 2025-10-29

**Soundness:** 3
**Presentation:** 3
**Contribution:** 2
**Rating:** 4
**Confidence:** 4

**Summary:**

Large vision language models (LVLMs) often hallucinate objects or attributes that are not present in the image, undermining their reliability. Visual Contrastive Decoding (VCD) mitigates this by contrasting the logits from the original image against those from a noisy version, down‑weighting hallucination‑prone tokens. However, the noise used in VCD is undirected Gaussian perturbations that indiscriminately degrade features and provide a weak contrastive signal. This paper proposes Adversarial Visual Contrastive Decoding (AVCD), which replaces random noise with adversarial perturbations crafted to maximize feature dissimilarity with the original image. Adversarial images are generated via Projected Gradient Descent on the cosine similarity between CLIP (or other) vision‑encoder features of the original and perturbed image. Because these adversarial perturbations actively steer the model toward hallucinatory states rather than merely adding noise, they create a stronger negative example for contrastive decoding. During inference, AVCD computes the decoding distribution p(y|v,x) by combining the logits from the original image and the adversarial image with a hyperparameter alpha similar to VCD.

**Strengths:**

S1. The authors show that adversarial perturbations increase output entropy more than Gaussian noise and theoretically push the image embedding closer to hallucinated object descriptions (Proposition 1), providing intuition for why AVCD is effective.

S2. The method is applied at inference time without modifying the model.

S3. AVCD improves hallucination‑mitigation metrics and general MME scores across models

**Weaknesses:**

W1. Generating adversarial examples requires gradient access through the vision encoder and one or more PGD iterations. This may not be feasible for black‑box models and adds inference overhead; even though the authors report modest slowdowns (1.22×-1.43× on short responses), the cost could grow with more steps or larger images.

W2. AVCD relies on CLIP‑like vision features and a projector to compute the adversarial loss. Closed or proprietary models without accessible feature representations may not support this method directly.

W3. While AVCD shows gains on hallucination benchmarks and MME, the method involves creating large perturbations (ε up to 16 or higher) since images are internal. These perturbations might inadvertently distort fine‑grained visual cues needed for tasks such as attribute comparison or counting; more analysis of downstream effects would strengthen the work.

**Questions:**

Q1. ow might AVCD be applied to MLLMs without public access to their vision encoder gradients or projector weights (e.g., GPT‑4o)? Could a surrogate vision model be used, and what is the impact on performance?

Q2. Have you evaluated AVCD on tasks requiring precise attribute reasoning (e.g., “how many red cubes are stacked?”) to ensure that adversarial perturbations do not degrade performance on such tasks?

---

> ### Author Response · Authors · 2025-11-21
>
> **Regarding Inference Overhead and Scalability (W1)**
>
> (Note: We address the applicability to black-box models in a separate response.)
>
> We appreciate the reviewer’s feedback regarding the computational cost and scalability of gradient-based adversarial generation. We acknowledge that our method introduces an overhead; however, we argue that this cost is a fixed, one-time investment per prompt that becomes negligible in practical, realistic scenarios. We present empirical evidence demonstrating that this overhead is effectively amortized as the complexity of the task (input length, output length) increases.
>
> **Amortization via Longer Context (Input & Output):** The computational cost of AVCD is concentrated in the initial visual attack phase, which occurs only once. Consequently, as the LLM processing time increases (due to longer system and input prompts or longer generated responses), the relative overhead of AVCD diminishes significantly.
>
> - **Experiment on Input Length:** We conducted an experiment on the POPE benchmark by increasing the input prompt length from 50 tokens to approximately 100 tokens (via system prompt repetition). Then, the relative time increase of AVCD over VCD dropped from ~22% (in the short prompt setting) to just ~4% (3m 4s for VCD vs. 3m 12s for AVCD).
>
> - **Real-world Implication:** In typical real-world scenarios (e.g., ChatGPT-style interactions) involving default system prompts and multi-paragraph responses, the fixed cost of AVCD becomes trivial. We estimate that for such lengthy interactions, the overhead would drop to less than 1% of the total inference time, making it virtually imperceptible to the end-user.
>
> **Robustness to Larger Images:** To address the concern that "cost could grow with larger images," we compared the wall-clock time on the CHAIR benchmark using standard and larger image resolutions (1.5x).
>
> | Setting | VCD | AVCD |
> | :--- | :---: | :---: |
> | **Small Image** | 33m 54s | 35m 08s |
> | **Large Image** | 36m 07s | 37m 01s |
>
> - **Analysis:** The relative overhead of AVCD compared to VCD remains consistently marginal across both settings. The slight fluctuation (from ~3.5% to ~2.4%) is within the range of normal runtime variance. Since the inference time is overwhelmingly dominated by the autoregressive text generation of the LLM, increasing image resolution does not disproportionately penalize our method.
> - **Dominance of Text Generation:** The key insight is that the adversarial generation is a one-time operation performed exclusively on the vision encoder. Consequently, even as image resolution increases, the additional visual processing cost remains negligible relative to the total generation time.
>
> Crucially, unlike some adversarial training methods that require gradients through the entire LLM, AVCD constructs the adversarial example using ONLY the Vision Encoder. This ensures that the computational complexity of our attack remains decoupled from the LLM's depth or the length of the generated text. If the attack required the LLM (e.g., targeted attacks on specific tokens), the cost would scale linearly with generation length. In contrast, AVCD maintains a constant adversarial generation cost regardless of how long the conversation becomes, ensuring efficiency in complex reasoning tasks.
>
> **Performance Consistency with Larger Images:** Finally, we verified that scaling up the image resolution does not compromise the efficacy of our method. We evaluated the hallucination mitigation performance of AVCD under the larger image resolution setting across all benchmarks.
>
> | Method | CHAIRS↓ | CHAIRI↓ | POPE-R↑ | POPE-P↑ | POPE-A↑ | MME↑ |
> |:-------|:-------:|:-------:|:------------:|:------------:|:------------:|:------:|
> | Baseline | 23.3 | 7.6 | 89.7 | 86.8 | 81.7 | 1492 |
> | VCD | 24.2 | 8.0 | 89.2 | 86.3 | 80.9 | 1496 |
> | **AVCD** | **21.4** | **6.9** | **90.4** | **87.7** | **84.2** | **1536** |
>
> As demonstrated in the table, AVCD maintains its robust superiority over both the default and the VCD baselines across all metrics (CHAIR, POPE, and MME). This confirms that our method effectively scales to larger visual inputs, leveraging the detailed visual information to provide even stronger contrastive signals without degrading performance.

---

> ### Author Response · Authors · 2025-11-21
>
> **Regarding Applicability to Black-Box Models (W2, Q1)**
>
> Since this concern regarding applicability to black-box models and the use of surrogate models was raised by multiple reviewers, we have provided a comprehensive response in the "**Common Response: Applicability to Black-Box Models**" section above. Please refer to that section for our detailed experiments on GPT and Gemini, which demonstrate the successful transferability of AVCD.
>
> **Regarding Fine-grained Visual Cues (W3, Q2)**
>
> We thank the reviewer for the thoughtful comment. We understand the concern that large perturbations in the adversarial branch might distort fine-grained visual cues required for tasks like attribute comparison or counting.
>
> However, we clarify that AVCD contrasts the distorted signal against the clean original signal. The clean image (positive branch) preserves all fine-grained details, while the adversarial image (negative branch) serves to suppress hallucinated priors. To empirically prove that this mechanism does not harm—and in fact improves—fine-grained perception, we conducted additional experiments on the MMHal [1] benchmark, following the recommendation of Reviewer ULJQ.
>
> **Why MMHal?** MMHal is specifically designed to evaluate challenging fine-grained visual reasoning.It explicitly categorizes questions into types such as Attribute (e.g., Q: "What color is the fire hydrant cap?" A: "The fire hydrant cap in the picture is black.") and Adversarial/Non-existence (e.g., Q: "How many traffic lights are there in the image?" A: "There are two traffic lights in the image."). This makes it an ideal testbed to address the reviewer's specific concerns regarding attribute comparison and visual fidelity.
>
> **Experimental Results:** We evaluated AVCD ($\epsilon=64$) on MMHal relative to baselines.
>
> | Method | Greedy | VCD  | AVCD |
> | :---: | :---: | :---:| :---: |
> | Score | 1.93 | 2.01 | 2.14 |
>
> AVCD achieves the highest score (2.14), outperforming the baseline (1.93). This indicates that the large perturbations in the negative branch do not degrade the model's ability to perceive fine-grained attributes. Instead, they help the model reject "hallucinated attributes" by penalizing the language priors that often override visual evidence. The clean branch ensures that the correct visual details are maintained. The superior performance on MMHal confirms that AVCD effectively mitigates hallucinations without compromising the model's capability to handle fine-grained visual tasks like attribute detection.
>
> [1] Sun, Zhiqing, et al. "Aligning large multimodal models with factually augmented rlhf." Findings of the Association for Computational Linguistics: ACL 2024. 2024.

---

> ### Comment · Reviewer_PwPs · 2025-11-22
>
> Thank you for the detailed rebuttal and extensive additional experiments.
>
> On overhead (W1): The new timing results clarify that AVCD’s cost is a one-time vision-encoder hit and that relative overhead diminishes with longer contexts. Still, a 1.2–1.4× slowdown is non-trivial in latency- or throughput-sensitive settings, especially for short outputs. In the paper, I would prefer this to be framed as “modest but noticeable overhead” rather than “virtually negligible,” with clearer caveats about deployment regimes.
>
> On black-box applicability (W2, Q1): The “Transfer & Correct” pipeline and GPT/Gemini experiments are interesting but remain a proof-of-concept. They rely on a well-aligned surrogate, multi-stage prompting, and careful engineering, and the gains, while real, are modest. I see this as an exploratory extension rather than evidence that AVCD is practically “black-box compatible” in a strong sense. I would encourage you to significantly soften and narrow the corresponding claims.
>
> Additional concerns raised by other reviewers:
> - The novelty of replacing Gaussian noise with adversarial perturbations is viewed as incremental.
> - The method relies on unusually large perturbation magnitudes, while the supporting theory uses simplifying assumptions that may not hold in this regime.
> - Performance is sensitive to ε, α, and step settings, and optimal configurations vary by task.
> - The method still fundamentally requires white-box access, limiting real-world applicability.
>
> These points suggest that, while the method is well-executed and empirically strong, the conceptual and practical contributions remain somewhat narrow.
>
> Overall, my primary reservations concern the optimistic framing of overhead and black-box usage, along with the incremental nature of the core contribution. I view the work as a solid improvement over VCD with clear analysis, but not a fundamentally new direction. I therefore keep my score unchanged; although I would not oppose acceptance if the AC prioritizes empirical performance and simplicity over the remaining concerns.

---

> > ### Author Response · Authors · 2025-11-22
> >
> > We appreciate your thoughtful feedback. Regarding your remaining concerns, we would like to provide final clarifications to contextualize our contributions.
> >
> > **On Inference Overhead (Context-Dependent)** We acknowledge that in scenarios involving short inputs and brief responses, the slowdown is modest but noticeable.
> > - **Refinement:** In the final version, we will explicitly characterize the overhead as "context-dependent"—negligible for long-form generation but a trade-off to consider for short-form tasks.
> >
> > **On Black-Box Applicability (Scope & Potential)** We agree that the "Transfer & Correct" pipeline is best viewed as an exploratory extension.
> > - **Clarification:** Our methodology is primarily designed for "open-weights deployment," aligning with the standard scope of inference-time interventions. While established methods in this domain (e.g., OPERA, VCD, ProjectAway) typically consider black-box scenarios out of scope, we explored this direction to investigate the potential of adversarial transferability. We included these experiments to highlight the unique potential of adversarial perturbations for transfer-based correction, rather than to claim it as a fully solved black-box solution.
> >
> > **On Theoretical Assumptions (Refined Analysis)** We acknowledge the limitation of the first-order Taylor approximation under large perturbation budgets, a point also raised during the review process. To address this, we have refined our theoretical framing in the revision.
> > - **Refining the Logic:** While our initial analysis utilized Taylor expansion to estimate a theoretical lower bound for $\epsilon$, we recognize that our method—unlike traditional adversarial attacks—does not impose a strict upper bound on $\epsilon$.
> > - **Existence Formulation:** Consequently, the fundamental requirement is not to derive a precise lower bound via local approximation, but to demonstrate the existence of a perturbation $\delta$ that inverts cosine similarity ($a < 0$).
> > - **Necessity of Magnitude:** Our analysis suggests that given the continuity of the encoder, such a state is achievable. Crucially, a large $\epsilon$ is not a regime where the theory fails, but rather a necessary condition to mechanically force this semantic inversion, validating our design choice.
> >
> > **On Hyperparameter Sensitivity** We wish to clarify that the hyperparameter search space is minimal in practice.
> > - **Fixed & Coupled:** The contrastive weight $\alpha$ is fixed at 0.5 across all experiments. The number of steps ($T_{adv}$) is deterministically coupled with $\epsilon$.
> > - **Simplicity:** Consequently, users need to adjust only one parameter ($\epsilon$) based on the task type. This makes the method highly reproducible and easy to tune compared to multi-parameter grid searches.
> >
> > **On Methodological Novelty (Active Steering)** We acknowledge that AVCD can be viewed as an evolution of contrastive decoding. However, we respectfully submit that our primary contribution lies not in algorithmic complexity, but in the rigorous theoretical and empirical characterization of why this evolution is necessary.
> > - **Rigorous Characterization:** A significant portion of our work is dedicated to empirically and theoretically distinguishing "Active Directional Steering" from the heuristic of passive degradation (random noise).
> > - **Mechanistic Insight:** Through rigorous investigation at both the Encoder and LLM levels (e.g., VAR analysis, similarity shifts), we provide concrete evidence that adversarial signals actively forge false grounding rather than merely breaking it.
> > - **Value of Insight:** We believe that this comprehensive characterization—which explains why this evolution is necessary and effective—constitutes a meaningful contribution to understanding and controlling LVLM hallucinations.
> >
> > We believe AVCD offers a practical, training-free, and effective solution. We hope the strong empirical results and the clarifications regarding our theoretical grounding warrant your support.
> >
> > Finally, we would like to express our sincere gratitude for your review. We recognize that reviewing is a time-intensive process, and your feedback regarding practical deployment constraints and the potential impact on fine-grained perception has been valuable in refining the scope and clarity of our work. We genuinely appreciate your time and consideration.

---

> > > ### Comment · Reviewer_PwPs · 2025-11-27
> > >
> > > I truly appreciate the authors for their effort for improving framing and the theoretical positioning. However, these refinements do not substantially alter my core assessment.
> > >
> > > The method is a well-executed but I still feel it is an incremental extension of VCD; the practical overhead is still meaningful in several deployment regimes; the black-box applicability remains exploratory.
> > >
> > > I feel the contribution is still not clearly above the ICLR acceptance bar. I will maintain my rating.

---

> > > > ### Author Response · Authors · 2025-11-27
> > > >
> > > > We sincerely thank you for returning to the discussion. We genuinely appreciate the time and effort you have dedicated to reading our rebuttal and providing these final comments. Your rigorous feedback has been instrumental in refining the framing of our contributions.
> > > >
> > > > Regarding the remaining reservations, we respectfully wish to clarify three aspects based on our findings. First, beyond the methodological change, our work is grounded in comprehensive analysis—**theoretical analysis, vision encoder feature analysis, and LLM-level attention analysis**—which mechanistically distinguishes our approach from previous works. Second, regarding overhead, we emphasize that AVCD incurs a **fixed O(1) overhead**; our cost is fully decoupled from generation length, ensuring scalability in realistic long-context scenarios. Third, regarding black-box applicability, we stress that prior decoding interventions are **structurally incapable** of deployment on closed APIs. Consequently, as no precedent existed for such tasks, our "Transfer & Correct" pipeline establishes a **novel protocol** to bridge this gap, demonstrating the **unique capability** of adversarial visual features to transfer to closed models and mitigate hallucinations where standard decoding interventions are inapplicable.
> > > >
> > > > We believe these distinctions—theoretically grounded steering, scalable efficiency, and novel transfer capabilities—constitute a significant step forward for trustworthy LVLMs. We believe that these theoretical and empirical contributions offer significant value to the research community, clearly outweighing the discussed trade-offs. Thank you once again for your valuable time.

---

### Author Response · Authors · 2025-11-21

We thank the reviewers PwPs, ULJQ, nffn, and mm7w for their insightful comments and constructive feedback. We are encouraged by the reviewers' recognition of our work's motivation and solid analysis. To address the concerns raised, we have conducted extensive additional experiments involving new benchmarks, new approahces, and diverse baselines. Below is a summary of the key points raised by each reviewer and our corresponding major updates.

**Summary of Reviewer Comments**
- **Reviewer PwPs:** Raised concerns regarding inference overhead, applicability to black-box models, and potential distortion of fine-grained visual cues.
- **Reviewer ULJQ:** Questioned the optimization trade-off (magnitude vs. depth), suggested alternative adversarial formulations, requested broader benchmark coverage, and comparison study of alternative contrast constructions.
- **Reviewer nffn:** Pointed out hyperparameter sensitivity, applicability of theorem, and the transferability of perturbations across images.
- **Reviewer mm7w:** Discussed novelty, comparison with training-time/steering baselines, evaluation on modern backbones, transferability of adversarial noises, decoding strategies, and suggested targeted adversarial objectives.

**Summary of Revisions & New Experiments**
In response to these collective suggestions, we have significantly strengthened the paper with the following updates:

- **Applicability to Black-Box Models** (Response to PwPs, nffn, mm7w)
  - We introduced a "Transfer & Correct" strategy leveraging adversarial transferability.
  - New Experiments: Validated on GPT-4o-mini and Gemini-2.5-flash-lite using surrogate models (CLIP), demonstrating that AVCD can mitigate hallucinations in closed-source APIs.
  - Cross-Architecture Transfer: Verified that perturbations transfer effectively between distinct architectures.

- **Expanded Evaluation Scope** (Response to PwPs, ULJQ, mm7w)
  - New Benchmarks: Added MMHal-Bench to verify performance on fine-grained visual reasoning and attribute detection.
  - New Backbone: Evaluated on LLaVA-RLHF and Qwen2.5-VL, establishing the method's generality on strong architectures.
  - New Baselines: Compared against, OPERA, HALC, CICD, DeGF, and ProjectAway, demonstrating AVCD's superior balance of performance and efficiency.

- **Rigorous Methodological Analysis** (Response to ULJQ, nffn, mm7w)
  - Theoretical Refinement: Revised Theorem 1 and its proof to clarify the distinction between targeted and untargeted attacks, addressing the statistical independence assumption.
  - Magnitude vs. Depth: Conducted ablation studies proving that large perturbation ($\epsilon$) with minimal steps outperforms the traditional small $\epsilon$, many steps approach for hallucination mitigation.
  - Alternative Objectives: Compared Vision Encoder loss against output-level (logit) attacks and targeted attacks (push-toward-distractors), confirming the efficiency and efficacy of our proposed formulation.
  - Robustness: Verified performance consistency across larger input resolutions.
  - Hyperparameter Sensitivity: Validated the robustness of our original configurations, confirming that distinct settings for binary verification (e.g., POPE) versus detailed complex tasks are consistently optimal without requiring intensive tuning.
  - Decoding Strategy: Verified that AVCD's robustness holds under both greedy decoding comparison, confirming intrinsic logit calibration

**Revised Manuscript Uploaded:** We have integrated these extensive experimental results into the revised manuscript. The updated paper, reflecting these significant improvements, has been uploaded. We kindly invite the reviewers to examine the revised version, particularly the updated Main Text and Appendices.

---

> ### Author Response · Authors · 2025-11-21
> **Common Response: Applicability to Black-Box Models**
>
> **Common Response: Applicability to Black-Box Models**
>
> We thank the reviewers for their insightful comments on black-box applicability. However, we would like to clarify a fundamental distinction regarding inference-time hallucination mitigation methods.
>
> **Structural Limitation of Standard Methods:** Standard inference-time interventions (e.g., OPERA, ProjectAway) fundamentally rely on accessing and manipulating internal model states such as attention maps or intermediate layers. Therefore, these methods inherently require internal model access, making black-box adaptation a fundamentally different challenge, and demanding black-box compatibility is typically considered out of scope for this line of research.
>
> **The Unique Advantage of AVCD:** However, unlike internal intervention methods, contrastive decoding approaches (VCD, AVCD) operate by generating an input-level artifact (a distorted image). Crucially, while VCD relies on random noise—which creates non-transferable visual degradation—AVCD generates adversarial perturbations. We demonstrate that this allows for a "Transfer & Correct" strategy on black-box models by leveraging the unique property of adversarial transferability [1, 2]. Since modern LVLMs predominantly may employ ViT-based visual encoders, adversarial features optimized on a surrogate model effectively transfer to black-box models, inducing correctable hallucinations.
>
> **Experimental Strategy: Indirect Contrastive Prompting:** We devise a two-stage pipeline suitable for APIs where logit access is restricted:
> 1. **Transfer (Generate Negative Draft):**
>     - **Input**: Adversarial Image (generated via surrogate CLIP)
>     - **Process:** Feed the adversarial image to the black-box model to generate a caption.
>     - **Output:** Hallucinated Draft
> 2. **Correct (Filter & Refine):**
>     - **Input**: Original Image + Hallucinated Draft
>     - **Process:** Prompt the model to generate the final description, explicitly instructing it to use the draft as a negative reference and filter out details present in the draft that are not visually grounded in the original image.
>     - **Output:** Final Corrected Caption
>
> **Results:** We evaluate this pipeline on GPT-4o-mini and Gemini-2.5-flash-lite (using CLIP ViT-L/14 as the surrogate). We utilize 500 images from the CHAIR dataset, following the identical experimental settings as described in Section 4.1 of the main paper. To rigorously test whether improvement stems merely from rejecting a low-quality draft, we also include a "Noise Draft" baseline (pure Gaussian noise).
>
> | Target Model | Method | C_S ↓ | C_I ↓ | Observation |
> |:-------------|:-------|:-----:|:-----:|:------------|
> | GPT-4o-mini | | | | |
> | | Baseline | 13.4 | 6.8 | - |
> | | Noise Draft | 100.0 | 9.0 | No Info. |
> | | VCD Draft | 18.6 | 14.3 | Low Transfer |
> | | AVCD Draft | **32.0** | **50.1** | High Transfer |
> | | Default-Noise | 13.4 | 6.4 | - |
> | | Default-VCD | 14.2 | 7.4 | 6.0% Decrease |
> | | Default-AVCD | **11.8** | **6.4** | 12.0% Increase |
> | Gemini-2.5-flash-lite | | | | |
> | | Baseline | 16.0 | 6.4 | - |
> | | Noise Draft | 92.4 | 26.4 | No Info. |
> | | VCD Draft | 19.4 | 15.4 | Low Transfer |
> | | AVCD Draft | **41.6** | **49.7** | High Transfer |
> | | Default-Noise | 15.0 | 6.3 | 6.7% Increase |
> | | Default-VCD | 14.8 | 7.2 | 7.5% Increase |
> | | Default-AVCD | **13.8** | **6.3** | 13.7% Increase |
>
> **Key Findings:**
> - **Validation of Transferability:** The remarkably high hallucination rates in the AVCD Draft (CHAIR$_S$: 32.0 for GPT-4o, 41.6 for Gemini) compared to the VCD Draft confirm that adversarial perturbations transfer better than random noise, successfully inducing severe hallucinations even in closed-source models.
> - **Quality of Negative Constraint Matters:** Crucially, the "Noise Draft" yields maximal hallucination scores (reaching 100.0) due to no information. However, using this as a negative constraint fails to improve performance over the baseline. This proves that simply rejecting a bad draft is insufficient.
> - **Superior Correction:** In contrast, AVCD induces semantic hallucinations—plausible but incorrect objects. These serve as clearer negative examples. By instructing the model to reject the errors found in the AVCD Draft, Default-AVCD achieves the lowest hallucination rate, outperforming both the baseline and the VCD-based correction.
>
> This experiment demonstrates that while AVCD is white-box, the core principle—using adversarial perturbations to isolate and suppress hallucinations—is transferable. AVCD can thus function as a potent, transfer-based hallucination probe for black-box models, a capability that random noise-based methods cannot effectively match.
>
> [1] Goodfellow, Ian J., Jonathon Shlens, and Christian Szegedy. "Explaining and harnessing adversarial examples." arXiv preprint arXiv:1412.6572 (2014) \
> [2] Naseer, Muzammal, et al. "On Improving Adversarial Transferability of Vision Transformers." International Conference on Learning Representations

---

### Comment · Area_Chair_EYNh · 2025-11-26
**Author-Reviewer-AC Discussion (DDL: 12/3 9PM UTC)**

Dear Reviewers,

Thank you once again for your service to ICLR 2026. Now that the authors have submitted their rebuttal, I kindly ask you to take the following steps (if you have not done so already):

- Read the authors’ response and other reviews.
- Consider whether the rebuttal and additional comments affect your assessment of the paper.
- Engage in interactive discussion with the authors -- **Note the Author-Reviewer-AC discussion period ends on 12/3 9PM UTC**. You are recommended to keep active before that deadline. If you have more concerns/questions (e.g., requesting clarifications, new results), it is recommended to post your request asap, so that the authors have enough time to address them.

The current reviews for this paper are **mixed (scores: 4/8/4/4)**. Your further contributions are essential for forming a well-informed final decision.

I am happy to join and support the discussions between you and the authors. Please feel free to share your thoughts and participate actively in the discussion. Thanks!

Best regards,

AC

---

### Author Response · Authors · 2025-11-29

Dear Area Chair,

We understand that the review scores and discussions have been reverted due to the data leak. Since this rollback erases the consensus reached during the rebuttal period, we are posting this summary to inform the AC of the actual progress and the scientific agreement established with the reviewers.

Before the rollback, our scores had effectively **improved to 8 (Reviewer ULJQ) and 6 (Reviewer nffn)** based on the following resolutions:

**1. Reviewer ULJQ (Score raised to 8)** Reviewer ULJQ was satisfied with our rebuttal and explicitly raised their score.

- **Key Resolutions:** Confirmed that our ablation studies on adversarial budget, benchmark coverage (MMHal), and additional baselines on alternative contrasts (e.g., black/white images) addressed their concerns.
- **Quote from Discussion:** "The authors have effectively addressed my main concerns... Given the solid empirical performance... I am raising my rating accordingly to support its acceptance."

**2. Reviewer nffn (Score raised to 6)** Reviewer nffn raised their score after we resolved their primary concern regarding the theoretical analysis.

- **Key Resolutions:** We clarified that our method utilizes an untargeted attack, and revised the theorem to align with this clarification.
- **Quote from Discussion:** "The distinction made between targeted and untargeted attacks effectively addresses my concern... I am satisfied with this explanation... I decide to raise my score..."

**3. Reviewer mm7w (Score: 4, Discussion Ongoing)** At the time of the system halt, we were actively engaged in a productive discussion to address the reviewer's remaining concerns. In our latest exchanges (two rounds of Q&A beyond the initial rebuttal), we provided detailed clarifications regarding transferability, novelty, and the decoding strategy. Furthermore, we addressed the specific inquiries on Qwen2.5-VL performance and provided the requested MME benchmark breakdown.

**4. Reviewer PwPs (Score: 4)** Reviewer PwPs maintained their rating. In our final response, we addressed their reservations by clarifying that our work is distinguished by comprehensive theoretical and empirical analysis, incurs a fixed O(1) overhead decoupled from generation length (ensuring scalability), and establishes a novel protocol for black-box applicability where prior interventions are inapplicable.

We have uploaded the revised manuscript to the system, where all changes corresponding to these discussions are highlighted in blue. We request the Area Chair to evaluate our paper based on this established consensus and the detailed rebuttals that led to the score improvements.

Best regards, Authors

---

### Meta-Review · Area_Chair_FeV6 · 2026-01-07

**Summary:**

Reviewers have the following concerns on the paper.

1. The computational overhead. The opimistic framing of overhead and block-box usage [reviewer PwPs]
2. The technical novelty compare with VCD seems to be incremental [Reviewer PwPs]. Replacing Gaussian perturbations with adversarial "hard negatives" is a natural extension of contrastive decoding [Reviewer mm7w]
3. Insufficient experimental results. Insufficient benchmark [Reviewer ULJQ]. robustnesss and transferablity of the generated adversarial images [reviewer nffn]. The evaluated LVLMs are outdated. [Reviewer mm7w].
4. Limitation of the proposed AVCD. Sensitivity to hyperparameters [Reviewer nffn].

I have read the reviewers comments as well as the reponses from the authors. I have the same feeling that the technical novelty is limited, with reference to VCD. As such I am condering to reject the paper.

**Reviewer Concerns:**

Insufficient experimental results can be well addressed.
The llimitations of AVCD can be well addressed.
The technical novelties are still outstanding.

**Reviewer Scores:**

Reviewer PwPs will not change their score to positive.
Reviewer mm7w will not change their score to positive.

---

### Decision · Program_Chairs · 2026-01-26

Reject